# Genetic associations with educational fields

Rosa Cheesman [1,2] ✉, Ville Anapaz[3], Sjoerd van Alten [4],
Abdel Abdellaoui [5], Ralph Porneso[1], Joakim C. Ebeltoft[1], Ziada Ayorech[1],
Perline A. Demange[1], Espen Moen Eilertsen[1], Agnes Fauske[6,7],
Alexandra Havdahl [1,2,8], Hannu Lahtinen[9], Torkild Hovde Lyngstad [6],
Qi Qin [1], FinnGen[*], Andrea Ganna [3,10] & Eivind Ystrom [1,2,11]

Educational field choices shape careers, wellbeing and the societal skill distribution, yet genetic influences on what people study remain poorly understood. Here we show that genetic factors are associated with educational field specializations using genome-wide association studies (GWASs) across 463,134 individuals from Finland, Norway and the Netherlands (effective *n* between 40,072 and 317,209). We identified 17 independent genome-wide significant variants linked to 7 of 10 educational fields, with average heritability of 7%. The genetic signal is specific to field choice rather than educational level, persisting after controlling for years of schooling and confounding factors. By examining genetic clustering across specializations, we uncovered two key dimensions: technical versus social and practical versus abstract. We performed GWASs of these components and demonstrated distinct genetic correlations with personality, behavior and socioeconomic status. Our findings demonstrate that genomic research can illuminate 'horizontal' stratification, revealing insights into vocational interests and social sorting beyond traditional attainment measures.

Education is fundamental to the economies, cultures and social stratification systems of modern societies[1–3]. Extensive research has linked education to a myriad outcomes, including occupational[4], culture[5] and health[6] domains.

Although we often focus on years of schooling, the type of education matters just as much. Fields of education, ranging from fine arts to finance, are extremely diverse and involve varying degrees of cultural, economic, technical and communicative skills[7]. Engineering students typically earn more than humanities graduates, even with similar educational levels[8], and field specializations influence everything from

attitudes and fertility to social networks and marriage markets across generations[9–11]. This 'horizontal' stratification—differences in educational pathways despite similar years of schooling—tells a crucial part of the story that vertical measures miss. The mass expansion of educational systems has made field choice increasingly salient[12,13]. The importance of one's field of study has increased as the signaling value of educational level has weakened[8].

Educational choices follow social patterns. Women are overrepresented in caring fields such as nursing and social work, whereas men are overrepresented in technical fields such as engineering and

[1]PROMENTA Centre, Department of Psychology, University of Oslo, Oslo, Norway. [2]PsychGen Centre for Genetic Epidemiology and Mental Health, Division of Public Health and Prevention, Norwegian Institute of Public Health, Oslo, Norway. [3]Institute for Molecular Medicine Finland, HiLIFE, University of Helsinki, Helsinki, Finland. [4]School of Business and Economics, Vrije Universiteit Amsterdam, Amsterdam, the Netherlands. [5]Department of Psychiatry, Amsterdam UMC, University of Amsterdam, Amsterdam, the Netherlands. [6]Department of Sociology & Human Geography, University of Oslo, Oslo, Norway. [7]KIFO Institute for Church, Religion and Worldview Research, Oslo, Norway. [8]Nic Waals Institute, Lovisenberg Diaconal Hospital, Oslo, Norway. [9]Helsinki Institute for Demography and Population Health, University of Helsinki, Helsinki, Finland. [10]Analytical and Translational Genetics Unit, Massachusetts General Hospital, Harvard Medical School, Boston, MA, USA. [11]CREATE Centre for Research on Equality in Education, Faculty of Educational Sciences, University of Oslo, Oslo, Norway. [*]A list of authors and their affiliations appears at the end of the paper. ✉e-mail: r.c.g.cheesman@psykologi.uio.no

finance[14,15]. Generally, male-dominated occupations provide higher wages[16–18]. It has been suggested that these patterns are maintained due to socialization, priming, status, gendered norms and cultural stereotypes[19–21]. Parents' educational backgrounds strongly predict their children's field choices[22,23], with students from more educated backgrounds choosing more financially risky careers[24]. Geographical factors such as urban–rural disparities also affect choice norms and access to certain fields[25].

Beyond social factors, individual psychology shapes sorting into fields, through systematic behavioral tendencies, vocational interests and beliefs about future prospects[26,27]. For example, more extraverted people sort into fields that provide opportunities for social contact like healthcare and higher levels of openness to experience are observed among students of the arts, humanities and psychology[28].

Previous studies have tried to identify choice mechanisms by measuring preferences for intrinsic versus extrinsic rewards or entrepreneurial versus bureaucratic characteristics[29]. However, these preference measures are typically available only in small sample sizes, explain little variance and rarely include actual field choices. A holistic data-driven approach to understand the structure of educational fields would be to apply dimension-reducing multivariate techniques to actual field choices. However, this is difficult because an individual usually studies only one field.

Given that various traits linked to educational fields are heritable[30], fields of study likely are too. Genetic variants may be associated with field choices in several ways. There may be active gene–environment correlations (rGEs), where people choose their experiences in line with their heritable traits[31]. Evocative rGEs arise when individuals are encouraged into certain fields due to their heritable traits. Genetic influence has been demonstrated in twin studies of vocational interests and choices, such as in creative professions[32,33], and of school subject choices, where heritability estimates are around 50% for humanities and 60% for science, technology, engineering and mathematics (STEM)[34]. However, population-level genetic associations with diverse educational fields remain unstudied using modern genomic methods.

Genomic approaches offer unique advantages for studying educational fields. First, they can identify common dimensions underlying field choices by estimating genetic covariance structures using genome-wide association study (GWAS) summary statistics. It can be done even when study samples for each outcome do not overlap, such as when individuals are observed in only one field[35]. This allows identification of a smaller number of components explaining the covariance structure of educational field choices. Second, genomic data are valuable for causal inference when studied within the social context. Naive associations between genetic variants and educational outcomes include not only direct genetic effects (effects of an individual's own DNA on their field choice, operating via active and evocative rGEs) but also confounding due to correlations with environmental influences (passive rGEs)[31]. Possible confounders include indirect genetic effects of relatives' genomes on the focal individual's field choice, geographical and social stratification, for example, due to regional educational policies and population stratification (as a result of allele frequency differences across subpopulations)[36–38]. By placing individuals' genetic data within their family and geographical contexts, these mechanisms can be disentangled[38–40]. The relative contributions of direct versus nondirect genetic associations with field choices have yet to be established.

In addition, the study of educational fields can enrich genomic research on social stratification. Genomic research on conventional vertical position in educational attainment, income and occupational status[41–43] ignores the diversity of interests and skills that educational pathways entail and the important constraints on and consequences of these pathways. Genetic associations with various fields of study are unlikely to be completely accounted for by the known genetic correlates of socioeconomic status, and thus may lead to new insights on how individual and contextual factors combine to influence life chances.

Here we studied genetic associations with ten broad fields of education using population-wide data from Finland, Norway and the Netherlands. First, we explored whether genetic factors were associated with fields of education independent of level of education. Second, we separated direct genetic associations from confounding ones using within-family and geographical data. Third, we provided an empirical description of genetically associated clustering across educational fields and summarized key patterns (principal components (PCs)) of sorting into fields. Fourth, we expanded the scope of research on the role of educational fields within the social and life sciences through phenome-wide genetic correlation analyses.

Educational qualifications are complex outcomes influenced not only by individual traits, interests and skills, but also by numerous social barriers and supports. Our analyses focus on Nordic countries where education is free and social security high[44]. Results therefore likely reflect individual interests and preferences rather than family resources or monetary constraints, although social barriers persist even in these egalitarian settings.

## Results

### Genetic associations with ten fields of education
We analyzed population-wide administrative data from Norwegian and Finnish education registers for adults aged 25+ years, capturing ten broad fields defined by the International Standard Classification of Education (ISCED). We extracted data on individuals' highest completed qualification by the year 2018, including qualifications at all levels. After linking the register data to genotype data in the Norwegian Mother Father and Child Cohort Study (MoBa)[45,46] and FinnGen[47] and performing GWASs, we performed sample-size-weighted meta-analyses with METAL[48]. The sum of the effective sample sizes was 317,209 for engineering, manufacturing and construction, 292,929 for health and welfare, 261,182 for business, administration and law, 168,157 for services (including transport, security and personal services), 102,970 for education, 97,262 for arts and humanities, 69,123 for social sciences, journalism and information, 63,834 for agriculture, forestry, fisheries and veterinary, 50,819 for information and communication technologies (ICTs) and 40,072 for natural sciences, mathematics and statistics. Population and cohort sample sizes, plus effective sample sizes for GWASs, are shown in Supplementary Table 1.

We identified 17 independent genome-wide significant SNPs across 7 fields, with health and welfare showing the most associations (4 loci) and several fields showing 1–3 loci each (Supplementary Table 2a). All significant loci were field specific. Manhattan plots and quantile–quantile plots are shown in Supplementary Figs. 1–20. SNP associations identified for the fields with lower sample sizes (such as natural sciences, mathematics and statistics) are more likely to be false positives.

Liability-scale genome-wide SNP heritability estimates, calculated using linkage disequilibrium (LD) score regression[35], were 7% on average (median 5%) and ranged from 3% (health and welfare) to 14% (natural sciences, mathematics and statistics) (Fig. 1 and Supplementary Table 3). SNP heritability estimates were consistent across cohorts, with genetic correlations generally >0.75 (Supplementary Tables 4 and 5).

### Genetic associations exist net of educational attainment
Two approaches confirmed that genetic associations reflect the field choice itself, not just the educational level. Figure 1 shows that, after controlling for educational attainment (EA) as a covariate, mean SNP heritability dropped from 7% to 4%. GWAS-by-subtraction analyses[49] in Genomic Structural Equation Modeling (SEM; using the largest external GWAS of EA to date[43]) yielded similar results with a median SNP heritability of 3% (Supplementary Table 6). Five SNP associations remained significant after EA adjustment (Supplementary Table 2,b),

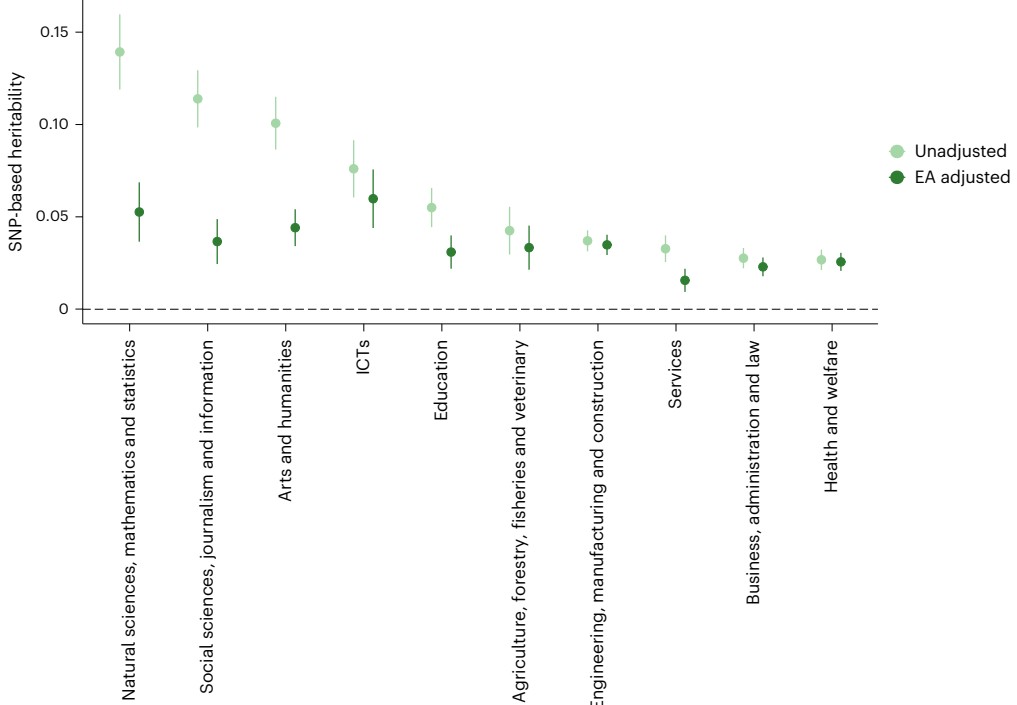

**Fig. 1 | SNP-based heritability estimates for educational fields, before and after adjusting for educational attainment.** Data are presented as a point estimate ± s.e. Statistical analysis used LD score regression with two-sided tests. The total sample size was 463,134, with the number of 'cases' ranging from 10,252 for natural sciences, mathematics and statistics to 102,874 for engineering, manufacturing and construction. The sum of effective sample sizes was 317,209 for engineering, manufacturing and construction, 292,929 for health and welfare, 261,182 for business, administration and law, 168,157 for services, 102,970 for education, 97,262 for arts and humanities, 69,123 for social sciences, journalism and information, 63,834 for agriculture, forestry, fisheries and veterinary, 50,819 for ICTs and 40,072 for natural sciences, mathematics and statistics. Educational attainment was adjusted for as a covariate.

and Genomic SEMs confirmed significant field-specific variance (Supplementary Table 7). Genetic correlations between EA and fields indicated that the adjustment procedure was successful for most fields, although some EA variance remained for natural sciences and social sciences (both with ~0.3 genetic correlation with EA; Supplementary Table 8). Unless otherwise stated, we focused on unadjusted GWAS results. Detailed discussion of causal interrelationships and adjustment methods appears in Supplementary Notes and Supplementary Fig. 21, with field-specific EA distributions shown in Supplementary Figs. 22 and 23 and Supplementary Table 9.

### Genetic associations capture direct genetic effects

Population-level genetic associations with field choices might reflect not only direct genetic effects but also indirect genetic effects, geographical influences and population stratification. Although these include causal environmental effects, they are confounders when estimating direct genetic effects. We used two different approaches to understand the relative contributions of nondirect genetic effects to our main findings.

In the independent Lifelines cohort ($n = 36,501$), 8 of 10 polygenic indices (PGIs) associated with their respective fields at $P < 0.005$ (Supplementary Table 10), although effect sizes were small to negligible. The largest associations were for arts and humanities (change in log(odds) = 0.22, s.e. = 0.03 and $R^2 = 0.00529$, where $R^2$ is the pseudo-coefficient of determination for the logistic model) and natural sciences, mathematics and statistics (change in log(odds) = 0.17, s.e. = 0.04, $R^2 = 0.00283$). Next, in a subsample of 17,705 individuals, we included the imputed sum of the PGIs of their parents as a control variable. This exploited random within-family genetic variation to estimate direct genetic effects without confounding. Direct genetic effects did not differ significantly from population estimates, suggesting no evidence for nondirect genetic effects on population associations

(Fig. 2 and Supplementary Table 11; see the bootstrapping results in Supplementary Table 12). However, power for within-family analyses in the subsample was lower and only two PGI–field associations remained statistically significant at $P < 0.005$. Results were similar when employing linear rather than logistic models (Supplementary Tables 13 and 14).

Furthermore, we explored assortative mating involving educational fields. We tested whether the same PGIs predicted the educational field of one's spouse or partner ($n = 28,581$). For education, arts and humanities and services, the PGI is associated with the educational field of one's spouse or partner at $P < 0.005$ (Supplementary Table 15).

We performed GWASs of educational fields in MoBa controlling for (1) birthplace municipality and (2) birthplace and parents' educational fields. We then calculated SNP heritabilities using the resulting summary statistics to approximate the within-region and within-region-and-family genetic variance. Figure 3 shows that there was little evidence for confounding: heritability estimates were not substantially lower after geographical and parental controls had been added. Modeling heritability ratios in Genomic SEM after ref. 38 showed that none of the adjusted estimates was significantly different from the original estimates, apart from for social sciences, journalism and information. For this field, SNP heritability dropped from 11% to 7% ($P = 0.03$; see Supplementary Tables 16 and 17 for heritability results and ratios and $P$ values, respectively).

Even after environmental confounding has been considered, direct genetic effects are mediated through the environment (see Supplementary Fig. 24 for an explanation of how gene–environment correlation mechanisms apply here).

### Technical–social and practical–abstract components of sorting

We summarized the genetic sorting into educational fields through genetic correlation and principal component analysis (PCA). First,

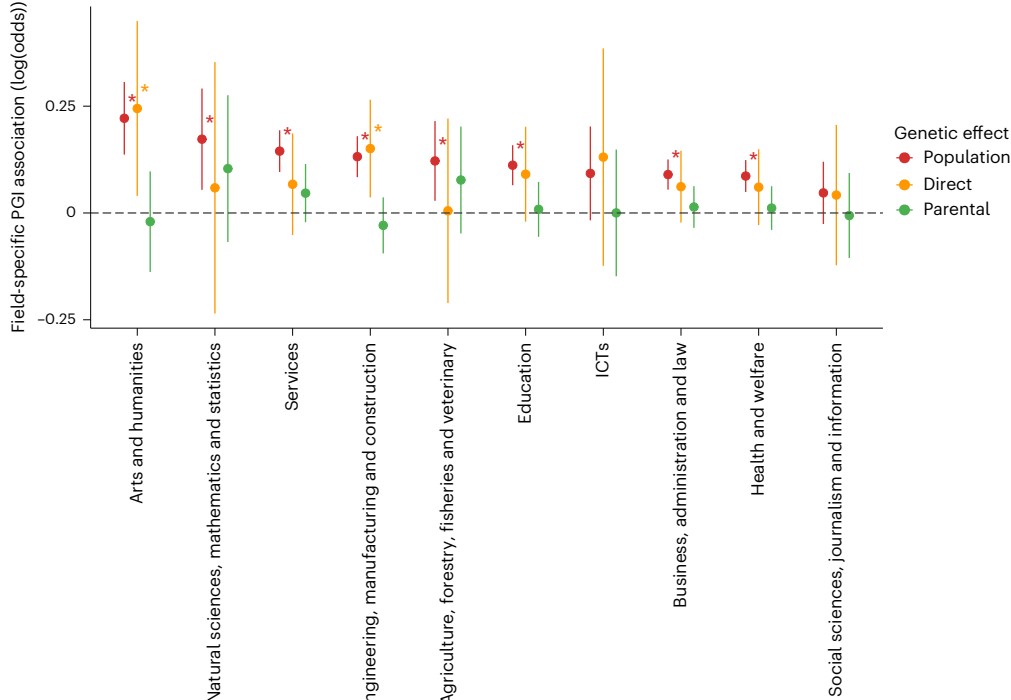

**Fig. 2 | PGI associations with educational fields in an independent Dutch cohort.** Data are presented as effect estimates with 99.5% confidence intervals (CIs). The statistical analysis used logistic regression with two-sided tests. Multiple comparisons were adjusted using Bonferroni's correction ($\alpha = 0.005$ for 10 hypotheses; * denotes statistical significance at the 0.005 level) ($n = 36,501$ for estimating population effects and 17,705 for estimating direct genetic effects and parental indirect genetic effects). Direct genetic effects are causal within-family estimates, whereas parental indirect genetic effects could be biased by population stratification and other gene–environment correlations. Significant within-family PGI associations were observed for arts and humanities (change in log(odds) = 0.245, $P = 0.0008$) and engineering, manufacturing and construction (change in log(odds) = 0.15, $P = 0.0002$).

we estimated pairwise genetic correlations between fields, using the GWAS-by-subtraction results to identify the key components of field qualifications beyond attainment. Figure 4 shows that STEM subjects correlated positively (for example, ICTs and natural sciences (Spearman's rank-order correlation ($r_g$) = 0.51, s.e. = 0.11)), as did arts and humanities with social sciences (see Supplementary Table 18 for genetic correlations among fields). Second, to make the interrelationships more interpretable and identify key patterns of sorting into fields, we applied PCA. The first two axes of variation (PCs) collectively explain 64% of the variance (see Supplementary Table 19 for PCA results). Although parallel analysis indicated that three PCs could be extracted (Supplementary Fig. 25), we focused on the first two PCs for simplicity and interpretability.

Figure 5 plots the contributions of the genetic associations for each field to the two first PCs. The correlation between an educational field and a PC is used as the coordinates of the variable on the PC. PC1 (horizontal axis in Fig. 3b), which we term 'technical versus social', reflects genetic variation correlated with qualifications in engineering, manufacturing and construction and natural sciences, mathematics and statistics versus education and health and welfare. PC2 (vertical axis in Fig. 3b), which we term 'practical versus abstract', reflects genetic-variation-correlated qualifications in services and health and welfare rather than social sciences, journalism and information and arts and humanities. The structure of genetic correlations between fields without controlling for EA is shown in Supplementary Fig. 26.

We performed GWASs of the PCs (following ref. 50). PC1 showed six genome-wide significant SNP associations and PC2 showed none (see Supplementary Table 2c for the lead SNPs, Supplementary Figs. 27 and 28 for Manhattan plots and Supplementary Table 18 for genetic correlations between the PCs and individual educational fields).

## Genetic correlates of the two PCs

Figure 6 presents the genetic correlations with these two PCs and 96 phenome-wide human phenotypes spanning domains, including personality, mental health, substance use, health and fertility (Supplementary Table 20).

A positive genetic correlation with PC1 indicates a link with technical fields, whereas a negative correlation with PC1 indicates a link with social fields. PC1 was negatively genetically correlated with extraversion and agreeableness ($r_g = -0.42$ and $-0.37$, respectively). Significant negative genetic correlations were also observed with trying cannabis ($r_g = -0.23$), alcohol dependence ($r_g = -0.20$) and six psychiatric diagnoses (average $r_g = -0.18$), whereas the genetic correlation with cigarettes per day was positive ($r_g = 0.14$). PC1 was positively genetically correlated with memory and childhood and adulthood IQ ($r_g = 0.37$ and 0.19, respectively), but negatively genetically correlated with income and noncognitive skills ($r_g = -0.13$ and $-0.10$, respectively).

A positive genetic correlation with PC2 indicates a link with practical fields, whereas a negative correlation with PC2 indicates a link with abstract fields of study. PC2 was negatively genetically correlated with open personality, autism, schizophrenia, bipolar disorder and trying cannabis ($r_g = -0.31$, $-0.27$, $-0.16$, $-0.12$ and $-0.13$, respectively). Negative genetic correlations were also observed with two fertility indicators: age at first birth and age at first intercourse ($r_g = -0.19$ and $-0.15$). PC2 was positively genetically correlated with visiting friends or family and family relationship satisfaction ($r_g = 0.24$ and 0.23) and with higher body mass index and waist-to-hip ratio ($r_g = 0.19$ and 0.17).

Although the PC summary statistics were adjusted for educational attainment using GWASs by subtraction, PC2 remains significantly (negatively) genetically correlated with occupational status, educational attainment and childhood IQ ($r_g = -0.26$, $-0.21$, $-0.29$), as well as occupational creativity ($r_g = -0.35$). Notably, PC2 was positively

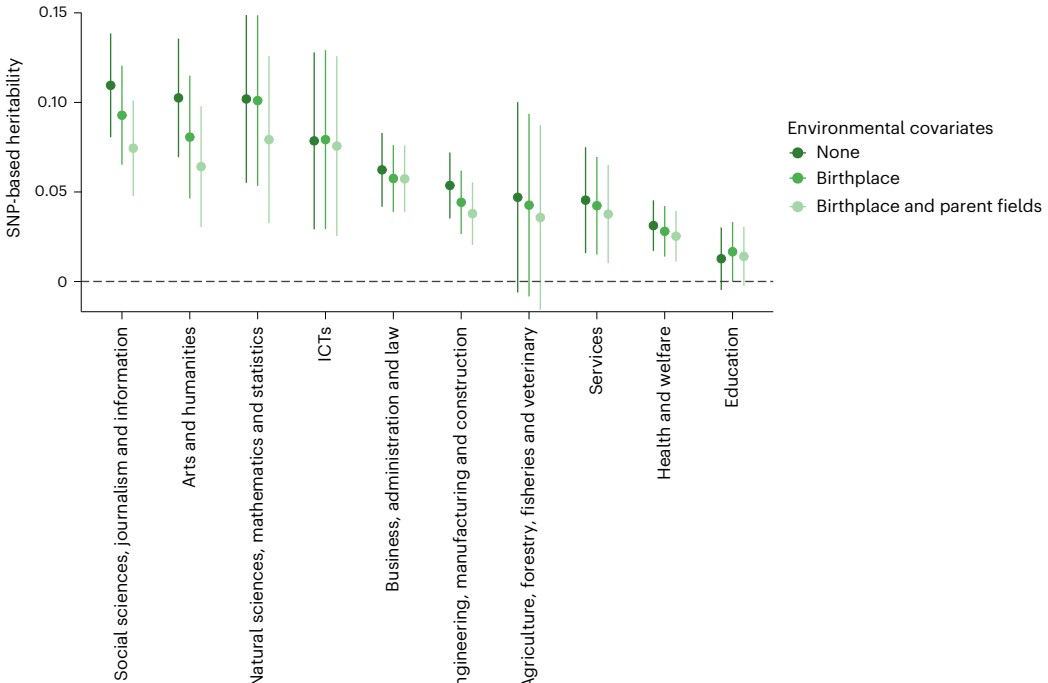

**Fig. 3 | SNP heritability estimates for educational fields controlling for birthplace and parental educational fields.** Data are presented as point estimates with 95% CIs. Statistical analysis used LD score regression with two-sided tests. The sum of effective sample sizes ranged from 40,072 for natural sciences, mathematics and statistics to 317,209 for engineering, manufacturing and construction.

genetically correlated with a latent factor capturing social and economic stability in the UK Biobank ($r_g$ = 0.18; this so-called factor 15 relates to social support networks, loneliness, home ownership, household income and never having been divorced), but negatively correlated with two other factors representing occupation or workplace environment (F5) and educational attainment (F10) ($r_g$ = −0.18 for both).

### Limited evidence for sex differences

The social–technical component (PC1) shows strong sex segregation: 84% of engineering qualifications and 88% of health and welfare qualifications go to men and women, respectively (Supplementary Fig. 29). To investigate the role of sex in the structure of genetic associations with educational fields, we performed two analyses. First, we repeated PCA excluding heavily sex-skewed fields (≥70% one sex: engineering, health and education). Second, we conducted sex-stratified GWASs and repeated PCA. Both analyses showed consistent genetic structure across sexes (Supplementary Figs. 30–32). SNP heritability estimates were similar between men and women, but cross-sex genetic correlations varied widely from 0.17 (engineering) to 0.72 (natural sciences) (Supplementary Table 21). However, several factors limit strong conclusions about sex differences. Sample sizes were low for sex-specific analyses (minimum of 952 ICT cases for women) and broad fields contain heterogeneous subfields with different sex distributions. Low genetic correlations may reflect different heritable traits for choosing subfields (for example, construction versus engineering) rather than fundamental sex differences in genetic associations.

### Discussion

Being sorted into fields of education is important for both individuals and society: it influences health, wellbeing and success, as well as the knowledge and skills available in the labor supply. Using population-wide data from Finland and Norway, we demonstrated genetic associations with field specializations independent of the educational level. Within-family analyses in an independent Dutch cohort suggested that these reflect direct genetic effects rather than confounding. We discovered two key dimensions describing sorting into fields: technical versus social and practical versus abstract. Extensive genetic correlations between these components and personality, fertility, mental health, substance use and socioeconomic status provide a wealth of hypotheses for downstream analysis on vocational interests and horizontal stratification.

We found that SNP heritability estimates for educational field choices were 7%, on average. These are lower bound estimates of the role of genetic factors because our methodology captures only additive effects of common variants tagged by genotyping arrays, rather than the full broad-sense heritability (the missing heritability problem[51]). Across two different approaches, namely within-family PGI associations and GWASs controlling for birthplace and parents' fields, we did not observe evidence for confounding of genetic effects. However, lack of statistical power for within-family analyses precluded us from making strong conclusions and genetic correlation results indicated substantial interplay with socioeconomic factors. Future large-scale, family-based studies should quantify these effects[43,52]. Given our own results and prior sociological evidence that the familial reproduction of field choices is an independent channel of transmission to that of attainment[53], indirect genetic effects on educational field choices may be smaller than those on educational attainment. Nevertheless, parental environmental factors are likely to be key mediators of direct genetic effects.

Genetic analysis revealed relationships that are difficult to study phenotypically, such as overlap between social sciences and arts and humanities. Through dimension reduction of the genetic correlations, we provided new evidence on patterns of sorting into fields. We found two important dimensions: technical versus social (PC1) and practical versus abstract (PC2). PC1 reflects differentiation into fields involving things versus people (for example, engineering versus education), whereas PC2 captures differentiation into hands-on, pragmatic versus theoretical and exploratory activities (for example, services versus social sciences). The PCs corresponded well with major theories in the

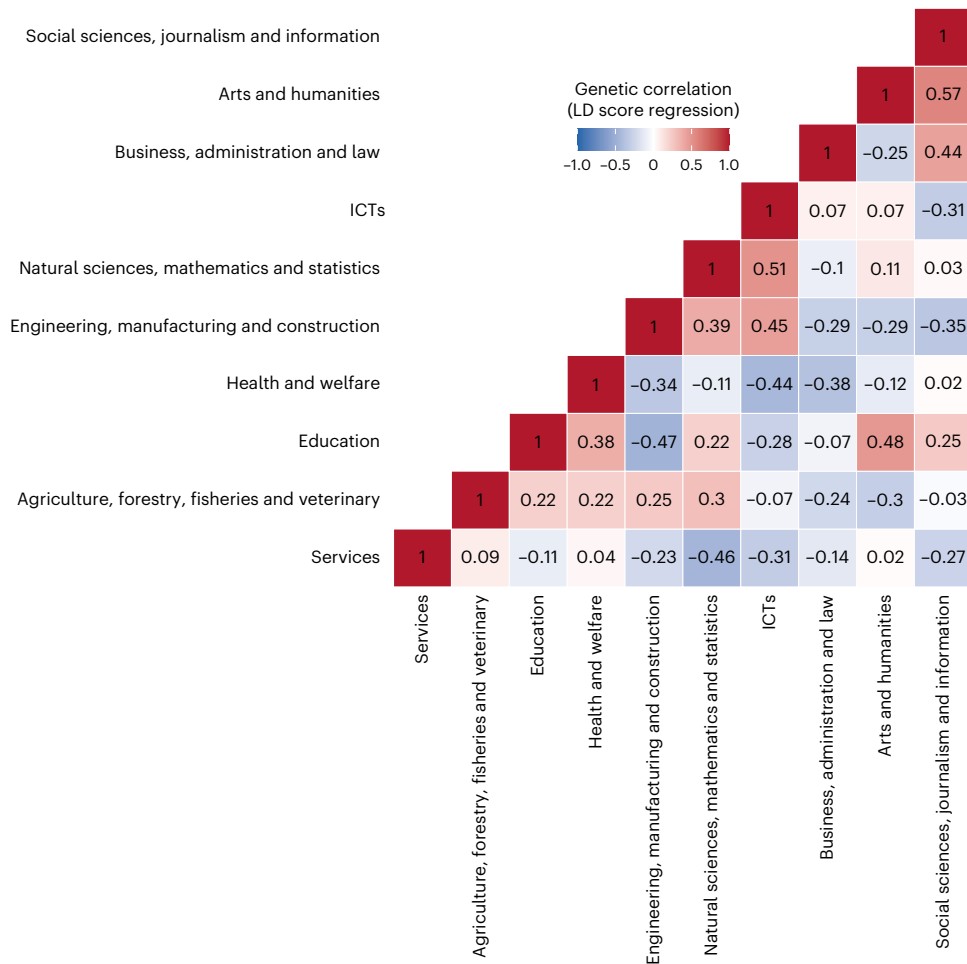

**Fig. 4 | Genetic correlations between educational fields adjusted for educational attainment using GWASs by subtraction.** Data are presented as genetic correlation coefficients. Statistical analysis used LD score regression with two-sided tests.

social sciences. The Realistic, Investigative, Artistic, Social, Enterprising and Conventional model of vocational interests, widely used by careers advisers[54], includes social and realistic interests (like PC1) and investigative or artistic and realistic or conventional interests (like PC2). The PCs also match sociological theory delineating the major educational resources in which individuals invest (communicative, technical, cultural and economic)[9,55]. This convergence between hypothesis-free genetic analysis and social science theory provides new validation of existing frameworks. Identifying patterns of social sorting without the need for theory or direct measurement of preferences demonstrates how genetic approaches may complement social–scientific enquiry (for corresponding results on social and health inequalities, see ref. 56).

In showing how technical–social and practical–abstract qualifications correlate on the genetic level with 96 human phenotypes, we expanded the scope of social science research on educational fields. Although studies have investigated the causes and consequences of interests and qualifications, such as personality[57], earnings[11] and fertility[58], these studies have been limited by the difficulty of measuring phenome-wide outcomes at scale in the same sample. We therefore incorporated new domains such as mental health, substance use, relationship satisfaction and body size.

Many of the genetic correlation results are consistent with the interpretation that the PCs capture individuals' heritable vocational interests. The technical–social component is genetically correlated with early developing social traits such as extraversion, agreeableness and frequency of social visits. The practical–abstract component captures individual tendencies toward open personality and creativity. The

positive genetic relationships with schizophrenia and bipolar disorder align with evidence that relatives of affected individuals are more likely to have creative jobs[59]. Vocational interests and the fit between one's work and interests play a critical role in career choices, productivity and finding meaning in life[60]. Capturing genome-wide associations with interests has not been possible due to insufficiently powered genotyped samples. We provided a new GWAS on a proxy of interest through educational field choice.

Results also capture wider social stratification patterns. The abstract–practical component (PC2) relates to traditional 'vertical' socioeconomic indicators including occupational status. This may partly reflect incomplete removal of educational attainment variance, although the genetic correlation between PC2 and EA remains small and abstract–practical fields are not clearly patterned by educational level (~40% of services and agriculture graduates have undergraduate degrees versus 76% for health and welfare). These findings may therefore reflect how social and economic resources hold greater importance in abstract educational trajectories. Unlike practical fields like education and healthcare oriented toward welfare state jobs, abstract qualifications often lead to elite professions in media, politics, research, law and the arts, which are typically more accessible to advantaged families[61]. It is interesting that our genetic results paint a more nuanced picture than conventional status markers, potentially identifying disadvantages of elite educational paths: propensity toward abstract rather than practical fields relates to socioeconomic instability, including loneliness, divorce, lower relationship satisfaction and higher risks of psychiatric disorders.

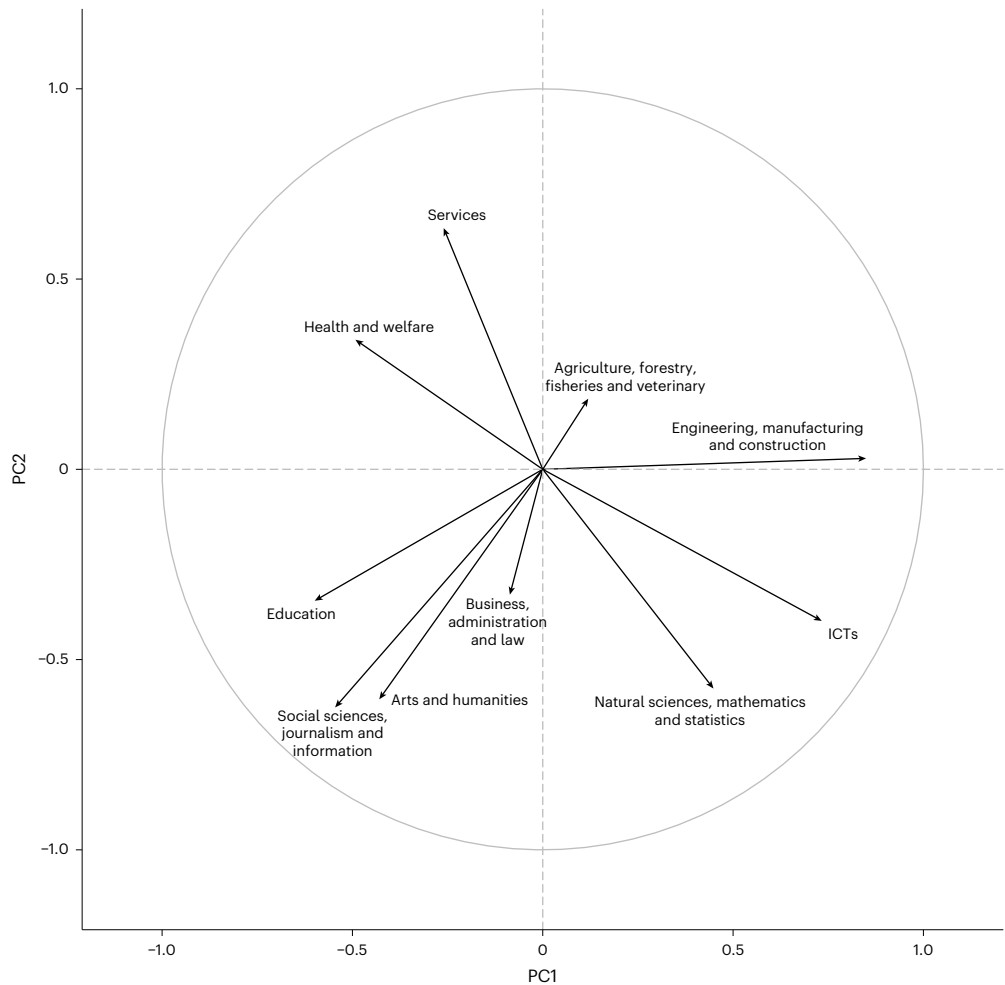

**Fig. 5 | The first two PCs of genetic variation associated with educational fields.** Data are presented as PC loadings. Statistical analysis used PCA. Positively correlated variables are grouped together and negatively correlated variables are positioned on opposite sides of the plot origin; variables away from the origin are well represented. The sum of the effective sample sizes is as follows: for PC1 = 10,413 and for PC2 = 7,353.

Why are genetic variants associated with field of study? In Nordic contexts with free education and strong safety nets, genetic effects are likely to operate through individual preferences and skills rather than resource constraints. However, mechanisms remain fundamentally social—genetic tendencies interact with environments through gene–environment correlations beginning early in life. Gender norms are a key social mediator, with stereotypes that influence choice of field of study beginning early. For instance, both girls and boys tend to be steered away from female-type educational tracks[21] and the gender gap in STEM degrees is partly because boys benefit from teacher biases[62]. Results could also capture downstream effects of educational program prerequisites and pick up dropout due to poor person–environment fit or discrimination. Results reflect probabilistic influences mediated through social contexts, not genetic determinism.

As discussed in our frequently asked questions document, determinist interpretations of genetic associations with complex outcomes like fields of study are wrong. Genetic factors do not determine field specializations but probabilistically influence individuals' tendencies, which, via interaction and mediation through the social and structural context, become correlated with educational outcomes. If the social context were to change, genetic associations might change. Genetic factors correlated with field choices may look different if people were encouraged to explore a wider range of subjects, if the skills involved in certain fields were different or the gender norms or economic returns to fields changed. In countries where social inequality is higher and the socioeconomic consequences of some field choices are riskier

than in Nordic countries, the heritability of field choices might be lower and links with individual interests and preferences might be less prominent.

Our study has several limitations. First, broad field categories may obscure specific genetic signals—for example, engineering and construction differ substantially despite grouping together. By increasing sample sizes in the future, it will become possible to study more homogeneous groups within narrow field categories using GWAS methods. Second, although it is an advantage of the study that we used two strict approaches to control for confounding due to passive gene–environment correlation and population stratification, these analyses were underpowered. Moreover, the SNP heritability estimates adjusted for birthplace and parental fields could still be confounded if parental fields did not have a perfect genetic correlation with offspring fields (for example, cohort differences) or by social influences of other relatives such as aunts, uncles and cousins[63]. Third, results from European populations in egalitarian societies may not generalize to diverse backgrounds or different welfare systems. Fourth, if a SNP is positively associated with being in one field, then it is mechanically negatively associated with being in other fields. Future work should investigate how this issue of seemingly unrelated regression[64] affects the genetic structure of fields, for instance, through multinomial regression.

These findings open new research directions in vocational interests and horizontal stratification. We introduced qualitative educational dimensions to complement quantitative GWAS literature on

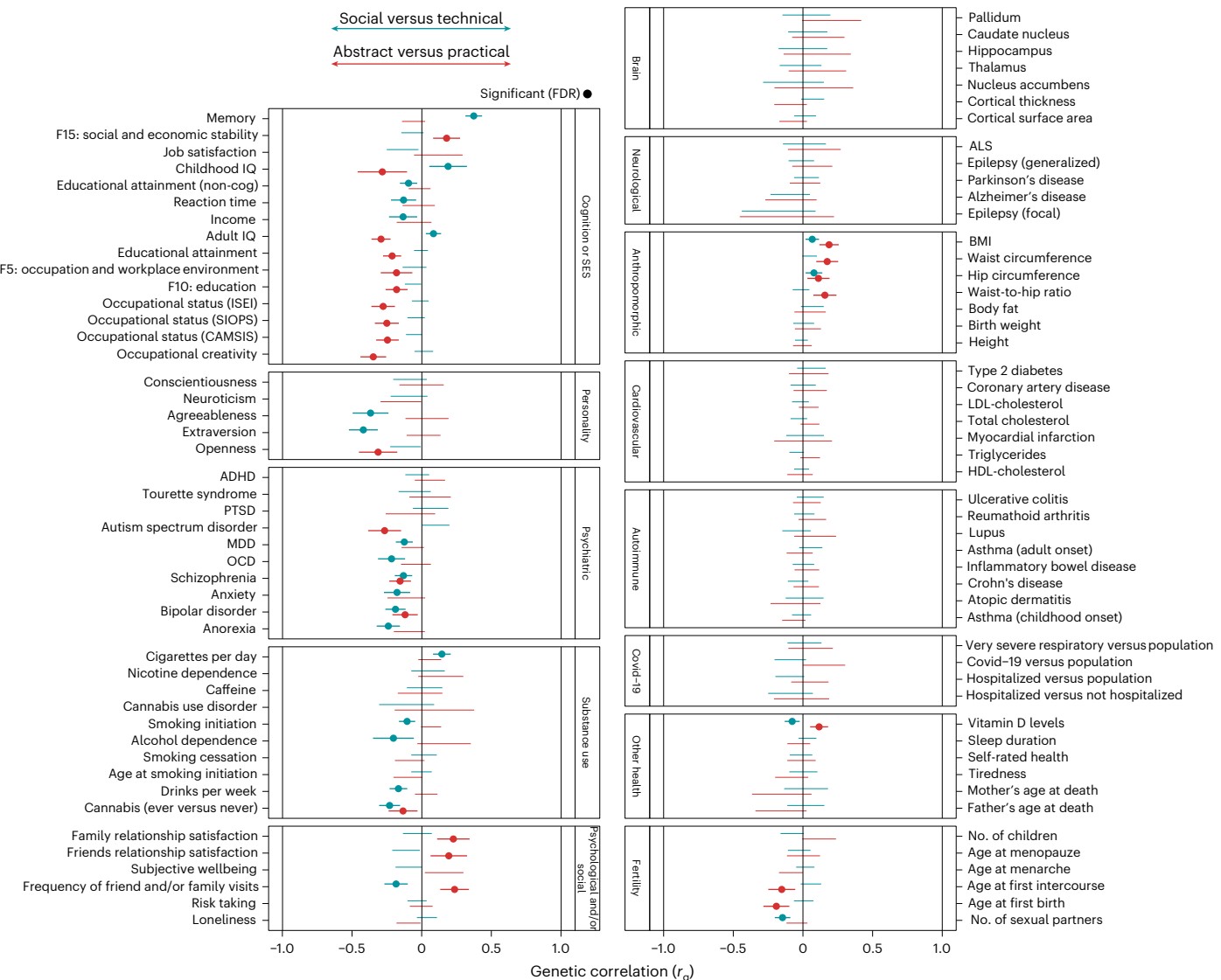

**Fig. 6 | Genetic correlations between two PCs of educational field specializations and 96 human phenotypes.** Data are presented as correlation coefficients with 95% CIs. Statistical analysis used LD score regression with two-sided tests. FDR correction was applied for multiple comparisons across 96 phenotypes. Educational field GWASs were adjusted for educational attainment. ADHD, attention-deficit and/or hyperactivity disorder; ALS, amyotrophic lateral sclerosis; BMI, body mass index; cog, cognitive; FDR, false discovery rate; HDL, high-density lipoprotein; LDL, low-density lipoprotein; MDD, major depressive disorder; OCD, obsessive–compulsive disorder; PTSD, post-traumatic stress disorder; ISEI, International Socio-Economic Index of Occupational Status; SIOPS, Standard International Occupational Prestige Scale; CAMSIS, Cambridge Social Interaction and Stratification Scale.

educational and financial attainments. Our summary statistics enable studies of early interest development, gene–environment interactions and causal effects of field choice on health and income[9]. Progress requires large-scale, family-based methods integrating interdisciplinary perspectives on individual preferences and social norms.

For answers to common questions about interpreting genetic associations with educational fields, see our frequently asked questions document in Supplementary Notes or online at https://www.thehastings center.org/genomic-findings-on-social-and-behavioral-outcomes-faqs/ and https://github.com/rosacheesman/Fields_genetics/wiki/ Frequently-Asked-Questions-(FAQ).

## Online content

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

**FinnGen**

**Andrea Ganna**[3,10]

A full list of members and their affiliations appears in Supplementary Table 22.

## Methods

### Ethics

This study complies with all relevant ethical regulations. The Norwegian Mother, Father and Child Cohort Study (MoBa) was approved by the Regional Committees for Medical and Health Research Ethics (protocol no. 2017/2205) and operates under the Norwegian Health Registry Act, with data handling managed by the University of Oslo under agreements with Statistics Norway. FinnGen received approval from the Coordinating Ethics Committee of the Hospital District of Helsinki and Uusimaa (protocol no. HUS/990/2017), with participants providing informed consent under the Finnish Biobank Act and multiple institutional permits from Finnish health authorities. Lifelines was approved by the University Medical Center Groningen (UMCG) Medical Ethical Committee (2007/152). All participants provided informed consent and data were processed in secure facilities compliant with national data protection regulations. See Supplementary Note for full details of ethical approvals.

### Contexts

Our main analyses were based on data from Finland and Norway, which are both social democratic welfare states[65] that fit the 'Scandinavian model' of education for all[66]. Compared to other high-income countries, economic inequality is low and access to education is less restricted by economic barriers. For example, Norway and Finland have free tuition, affordable loans and generous public subsidies for students. However, despite the reversal of the gender gap in educational attainment, gender-typical segregation into fields of study persists[44,67]. Correspondingly, Nordic labor markets are among the most gender segregated.

We also analyzed a Dutch sample. The Netherlands has been defined as a conservative welfare state[68]. Relative to the social democratic welfare states, social stratification in education is greater, partly due to early educational tracking and tuition fees[69].

### Samples

**FinnGen.** FinnGen (https://www.finngen.fi/en), launched in 2017, is a public–private research project, combining genome and digital healthcare data on about 500,000 Finns. The nationwide research project aims to provide new medically and therapeutically relevant insight into human diseases. FinnGen is a pre-competitive partnership of Finnish biobanks and their background organizations (universities and university hospitals) and international pharmaceutical industry partners and the Finnish Biobank Cooperative (FINBB). All FinnGen partners are listed at https://www.finngen.fi/en/partners. The project utilizes data from the nationwide longitudinal health register collected since 1969 from every resident in Finland. Analyses were conducted on individuals aged >25 years with complete data for genome-wide genotyping and complete educational records.

**The Norwegian Mother, Father and Child Cohort Study.** We studied adults who participated in MoBa, a prospective population-based pregnancy cohort study conducted by the Norwegian Institute of Public Health[46]. Pregnant women were recruited from across Norway from 1999 to 2009. In 41% of the pregnancies, the women consented to initial participation. Of fathers invited to participate, 82.9% consented. The total cohort includes approximately 114,500 children, 95,200 mothers and 75,200 fathers. Analyses were conducted on MoBa parents aged >25 years with complete data for genome-wide genotyping and complete administrative records linked to MoBa through the Norwegian national ID number system ($n = 125,016$).

**Dutch Lifelines.** Lifelines is a multidisciplinary, prospective, population-based cohort study examining, in a unique three-generation design, the health and health-related behaviors of 167,729 people living in the north of the Netherlands[70]. It employs a broad range of investigative procedures in assessing the biomedical, sociodemographic, behavioral, physical and psychological factors that contribute to the health and disease of the general population, with a special focus on multimorbidity and complex genetics. Participants were sampled from the northern population of the Netherlands and the final sample encompasses about 10% of the region's population. Between 2006 and 2013, randomly selected general practitioners invited all their listed patients aged 25–49 years to participate in the study. We restricted our sample to genotyped Lifelines respondents who were ≥25 years ($n = 63,927$). PGIs and the first ten PCs of the genetic data were linked to an administrative data file containing educational fields ('HOOGSTEO-PLTAB 2022, v1'), housed by Statistics Netherlands. Due to missingness in educational fields, in particular for the older generations, the total final sample was $n = 36,501$.

### Genetic data QC

**FinnGen.** FinnGen release 11 contains genotype data for 473,681 individuals after quality control (QC). A total of 387,601 individuals were genotyped with a FinnGen Thermo Fisher Axiom customized array v2. Data on 86,080 additional individuals were derived from legacy collections[47]. Further information is available at https://finngen.gitbook.io/finngen-handbook/finngen-data-specifics/red-library-data-individual-level-data/genotype-data/affymetrix-chip-and-its-design.

**MoBa.** Blood samples were obtained from both parents during pregnancy and from mothers and children (umbilical cord) at birth. Quality-controlled genotyping array data for the full 207,569 unique MoBa participants were recently generated[45]. Phasing and imputation were performed with IMPUTE4.1.2_r300.3, using the publicly available Haplotype Reference Consortium release 1.1 panel as a reference. To identify a subpopulation of European-associated ancestry, PCA was performed with 1,000 Genomes phase 1 after LD pruning. During post-imputation QC, the following thresholds were used for SNP removal: imputation quality (INFO) score ≤0.8; minor allele frequency (MAF) <1%; call rate <95%.

**Dutch Lifelines.** Blood samples were collected from Lifelines participants at the first assessment visit. Genotypes were released as part of two separate cohorts. The CytoSNP cohort was measured on the Illumina CytoSNP-12v2 array, measuring ~300,000 SNPs. The UMCG Genetics Lifelines Initiative (UGLI) cohort was measured on the Infinium Global Screening Array MultiEthnic Disease version, measuring ~700,000 SNPs. Quality-controlled data for both cohorts were released. The QC reports for CytoSNP and UGLI are available at http://wiki.lifelines.nl/doku.php?id=gwas and http://wiki.lifelines.nl/lib/exe/fetch.php?media=qc_report_ugli_r1.pdf, respectively. Before PGI construction, and in each cohort, we dropped multiallelic SNPs, SNPs with MAF < 1%, SNPs with an INFO score <0.8 or SNPs that were not in Hardy–Weinberg equilibrium ($P < 10^{-6}$). We also dropped individuals with homozygosity rates of ±3 s.d. values (removing 655 respondents). We further dropped 1,289 respondents from the CytoSNP cohort who were also available in the UGLI cohort. After all these QC steps were completed, we merged the CytoSNP and UGLI cohorts into a single data file, using only SNPs that both cohorts had in common after QC (~6.4 million SNPs in total).

### Measures

**Broad educational fields.** In all three cohorts, we extracted register data on broad educational field codes representing the field of education of each person's highest qualification completed by the year 2018. We extracted field codes at all highest qualification levels (that is, not just at university level).

To harmonize the data and facilitate future replication studies in other cohorts, we converted broad field codes from national-level coding systems to broad field codes as defined by the ISCED 2013

(https://uis.unesco.org/sites/default/files/documents/international-standard-classification-of-education-fields-of-education-and-training-2013-detailed-field-descriptions-2015-en.pdf).

In FinnGen, we used linked administrative data from Statistics Finland to define individuals' educational qualifications. The Finnish educational field records are described at https://www2.stat.fi/fi/luokitukset/koulutusala/. In MoBa, we used linked administrative data from the Norwegian Standard Classification of Education (NUS2000). The administrative data were of high quality and did not suffer from attrition. More information on the NUS coding and the conversion to ISCED is available at http://www.ssb.no/en/utdanning/norwegian-standard-classification-of-education. In the present study, missing data occur only for individuals whose fields do not map exactly on to the ISCED system, for instance because they are interdisciplinary (for example, 16,000 genotyped individuals in MoBa with qualifications termed 'interdisciplinary programs' and 'qualifications involving health and welfare'). Note that Dutch administrative records on education are incomplete, such that we had educational field measures available only for 56% of the original Lifelines sample ($n = 36,373$).

We created a binary variable for each of the ISCED broad field specialization codes, scoring individuals as 1 if they chose the field and 0 otherwise. The 0 category included people studying generic programs (not a specialization as such). This includes a wide range of qualifications, for example, unspecialized high school diploma and professional development skills training.

We also created harmonized educational attainment variables in all datasets. We took educational level information from the exact variable containing the field code and converted it to the ISCED Edu-Years (years of completed education) categories, as per international GWAS meta-analyses.

**Geographical and parental data in MoBa.** To test whether genetic associations reflected direct effects rather than confounding from familial and geographical factors, we created covariates from Norwegian population and education registers. We generated dummy variables for birthplace municipalities (216 codes representing Norway's lowest administrative level) to control for shared local environments and for parents' educational fields to control for familial transmission of education-specific skills, resources and networks. The genetic–geographical data linkage and structure has been described previously[71]. Although Norway's free tuition and dispersed population with geographical variation in educational opportunities[25] provide a useful context for these analyses, our approach may not fully capture temporally unstable or more localized influences.

**PGIs in Dutch Lifelines.** To validate our GWA analysis results, we tested associations between PGIs for educational field choices and those for actual educational field choices in an independent cohort. PGIs were constructed using the GWAS summary statistics for each respective field (not corrected for EA), using SBayesR within GCTB software v2.05beta_Linux[72]. SBayesR uses Bayesian shrinkage and explicitly models LD to estimate SNP effect sizes in the presence of correlated markers, using LD scores of individuals with European-associated ancestry estimated by the UK Biobank. To test for the direct effect of the PGI, we added the mid-parental PGI as a control variable. The parental PGI was constructed using a combination of observed and imputed genotypes (from parents and siblings), constructed using snipar. Snipar uses sibling data or the data of available parents to impute genotypes for unobserved parents[73]. The parental PGI could be imputed for individuals who had at least one sibling or at least one parent who was also genotyped in Lifelines ($n = 17,705$).

## Analyses

### GWA meta-analyses.
We performed GWA analyses (GWASs) for ten dichotomous broad educational field phenotypes in MoBa and FinnGen, using models developed for resource-efficient analysis of case–control phenotypes in biobank-scale datasets. In MoBa, the FastGWA-GLMM approach (a generalized linear mixed model (GLMM) method for large-scale (GWASs) in GCTA software v1.91.7beta (--fastGWA-mlm-binary))[74] was used. This is a logistic regression model with the added complexity of using a sparse matrix to account for the dense genetic relatedness in MoBa without removing relatives. FastGWA-GLMM also uses the saddle point approximation method to account for inflation in test statistics due to case–control imbalance. In FinnGen, the binary option in REGENIE[75] software was used (v2.2.4). REGENIE is a machine learning method that implements whole-genome ridge regression and uses Firth's logistic regression test to account for case–control imbalance. The two methods are similarly effective in terms of false-positive rates and statistical power.

To enable meta-analysis of MoBa and FinnGen GWAS summary statistics, we performed QC and harmonization. We removed variants with low MAF < 1%, poor imputation quality (INFO < 0.8), multiallelic variants and variants with ambiguous alleles (for example, alleles other than A, C, G or T) and we resolved strand and sign flips. The datasets were harmonized based on chr:pos from genome build 37 as an SNP identifier. Sample-size-weighted meta-analyses of MoBa and FinnGen were then performed using METAL software[48]. We used the SCHEME SAMPLESIZE setting to convert effect sizes to z-scores before meta-analysis.

To identify independent genome-wide significant associations in the meta-analytical results, we performed clumping using standard parameters in FUMA software[76]: leadP = $5 \times 10^{-8}$, gwasP = 0.05, r2 = 0.6, r2_2 = 0.1, refpanel = 1KG/Phase3.

We calculated cohort-specific effective sample sizes following ref. 77 and then summed these to obtain the sum of effective sample sizes for each field-of-study GWAS.

In MoBa, we also used FastGWA-GLMM for the following additional analyses: GWASs of ten fields with controls for educational attainment; GWASs of ten fields controlling for geographical and parental variables; and GWASs of ten fields in men and women separately.

In all GWAS (except sex-stratified) analyses, we controlled for sex, age, PCs of genetic ancestry (20 in MoBa, 10 in FinnGen) and batch identifiers.

### GWAS-by-subtraction analyses controlling for educational attainment.
To identify genetic signals associated with educational fields net of educational level, we used GWASs by subtraction[49] in Genomic SEM[78] (package within R-4.3.2) using GWAS summary statistics for educational fields and educational attainment[43]. This genomic Cholesky's decomposition approach was first applied to create a GWAS of non-cognitive skills by 'subtracting' the genetic component of cognitive performance from the association of each SNP with educational attainment. Here we 'subtracted' the genetic component of educational attainment from the association of each SNP with a given educational field. For each of the ten fields, we fitted an SNP-level Cholesky's model regressing two observed variables (the field, and educational attainment) on two latent variables (a field-specific factor and an educational attainment factor). The covariances between the latent variables were fixed to 0 so that all variance is explained by the latent factors. See Supplementary Fig. 21 and accompanying Supplementary Notes for full model specification details and a model illustration.

### SNP-based heritability analyses.
The overall contribution of common SNPs to field choices was estimated from GWAS summary statistics by LD score regression, via Genomic SEM v0.0.3f in R-4.3.2[35,78]. On average, SNPs with higher LD scores (more correlations with other SNPs) are more likely to be correlated with a true causal variant. As such, when GWA test statistics ($\chi^2$) are regressed on LD scores, the slope provides an estimate of the heritability that can be explained by common SNPs. Heritability represents the percentage of variance explained by common genetic variants.

We estimated SNP-based heritabilities of educational fields after controlling for EA in two ways: through GWASs by subtraction and phenotypic adjustment. We note that it is not recommended to base heritability estimates on the former approach, because environmental variance is undefined. Nevertheless, we reported SNP heritability estimates based on GWASs by subtraction because they are relevant to interpreting downstream PCs and genetic correlation analyses.

**Genetic analyses adjusting for birthplace and parents' fields.** To study how much the genetic associations with broad field choices are mediated through unobserved social factors in the geographical area in which individuals were born, we repeated the GWAS analyses in MoBa only, controlling for municipality codes and parental field codes as dummy variables and testing for attenuated heritability with LD score regression. We did this in an iterative fashion, first controlling for the most distal factor (birthplace municipality), then adding parental fields. Following previous methods applied in UK Biobank[38], we used Genomic SEM to compare the SNP heritability estimates with and without controls while accounting for dependence between estimates. We used the p.adjust function in R with the method 'fdr' to control the false discovery rate (the expected proportion of false discoveries among the rejected hypotheses)[79] and considered results with an adjusted $P < 0.05$ as significant.

**PGI analyses.** In Dutch Lifelines, we tested PGI associations with educational fields using logistic regression, controlling for ten PCs, age polynomials, sex and their interactions. We quantified variance explained using incremental McFadden's pseudo-$R^2$. This is defined as $1 - L_1/L_0$, where $L_1$ is the likelihood of the model and $L_0$ the likelihood of the model with only a fitted intercept. To estimate direct genetic effects, we added controls for imputed parental PGIs and compared within-family versus population-level effect sizes[80], with CIs from 1,000 bootstrap replications and Bonferroni's correction ($P < 0.005$) to correct for testing 10 hypotheses. We also tested PGI associations with spouse's or partner's educational field ($n = 28,581$; logit regression), defining spouses or partners as the first co-parent identified through Dutch population linkage, although these associations likely reflect broader assortative mating patterns rather than direct genetic effects on partner choice.

**Genetic structure of fields: genetic correlations and PCA GWASs.** We explored the structure of the field GWASs (meta-analytic results) by calculating genetic correlations using LD score regression within Genomic SEM v0.0.3f in R-4.3.2[78]. For pairs of traits, the product of the GWA $z$-scores at each SNP can be regressed on the LD score, providing an estimate of the genetic correlation between the two traits.

We explored the dimensionality of the genetic associations with the ten educational fields by applying PCA (using eigen() in R-4.3.2) to the standardized matrix of genetic correlations between fields (after GWAS by subtraction to remove EA genetic variance). To obtain GWAS summary statistics for PC1 and PC2, we then performed PCA GWASs following ref. 50 (more details at https://annafurtjes.github.io/genomicPCA/). This approach adapts a GWA meta-analysis function designed for meta-analysis across multiple traits[81]. Instead of weighting by SNP heritability when averaging across SNP effects, the standardized loadings on the PC of interest provide the weights. The function allows for sample overlap in the GWAS summary statistics by adjusting for the LD score regression intercept. The resulting GWAS summary statistics for the two independent educational field components had effective sample sizes of 10,413 and 7,353, respectively (following ref. 82).

We chose to use PCA rather than confirmatory factor analysis (CFA) for studying the dimensionality of educational fields for several reasons. First, PCA involves fewer assumptions than CFA. Although CFA models latent factors assumed to represent real traits measured by field choices, PCA simply reduces the dimensionality of the data by finding axes that explain maximum variation. In addition, CFA might require the somewhat ad-hoc addition of crossloadings based on modification indices to achieve a good fit, whereas PCA does not involve model-fit considerations. Second, the limited number of possible fields (ten indicators) makes PCA more suitable. CFA is ideally designed for studying latent factors that can be measured by an extensive, potentially infinite, number of indicators, which is not the case here.

**Genetic correlations.** We estimated genetic correlations between latent educational fields factors and 96 human phenotypes using LD score regression. We used publicly available GWAS summary statistics that were well powered and covered a comprehensive range of domains of human variation. See Supplementary Table 21 for the GWAS study reference list with sample sizes.

**Reporting summary**
Further information on research design is available in the Nature Portfolio Reporting Summary linked to this article.

## Data availability
For individual-level data, the Finnish biobank data can be accessed through the Fingenious services (https://site.fingenious.fi/en/) managed by FINBB. The Finnish Health register data can be applied for from Findata (https://findata.fi/en/data/). Instructions for access to MoBa data from the Norwegian Institute of Public Health can be found at https://www.fhi.no/en/studies/moba/for-forskere-artikler/research-and-data-access/. Instructions for access to the Lifelines cohort study can be found at https://www.lifelines-biobank.com/researchers/working-with-us. Administrative data on the Dutch population for the purpose of academic research are housed by Statistics Netherlands. Access is possible only for authorized institutions. Instructions on how to apply for the usage of administrative data can be found at https://www.cbs.nl/en-gb/our-services/customised-services-microdata/microdata-conducting-your-own-research. For summary statistics, the GWAS summary statistics produced from this study are available via Zenodo at https://zenodo.org/records/15584414 (ref. 83). Summary statistics from each data FinnGen release will be made publicly available after a 1-year embargo period and can be accessed freely at http://www.finngen.fi/en/access_results.

## Code availability
The code used in this study is available via GitHub at https://github.com/rosacheesman/Fields_genetics and via Zenodo at https://zenodo.org/records/15584414 (ref. 83).

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

## Acknowledgements

The Research Council of Norway supported this work through project nos. 288083 (R.C. and E.Y.), 325245 (R.C.), 274611, 336085 and 300668 (A.H.) and 336078 and 331640 (E.Y.). R.C. was also supported by the Jacobs Foundation (grant no. 2023-1510-00). S.v.A. was supported by the National Institute on Aging (grant no. R01AG078522) and the Dutch National Science Foundation (grant no. 016.VIDI.185.044). Research reported in this publication was supported by the European Union's (EU's) Horizon Europe research and innovation program under the Marie Skłodowska-Curie grant agreement (no. ESSGN 101073237). E.Y., P.A.D., E.M.E., J.C.E. and Q.Q. were supported by the EU grant agreement (grant no. 101045526, project GeoGen). T.H.L. and A.F. were supported by the European Research Council (ERC) consolidator grant (no. OPENFLUX 818420). H.L. was supported by the ERC (grant no. 101019329). A.A. is supported by the Amsterdam UMC Fellowship. E.Y. is also supported by the ERC (project no. 818425) and the Swedish Research Council (project no. 2024-06499_VR). A.H. was supported by the South-Eastern Norway Health Authority (grant nos. 2020022 and 2019097). Views and opinions expressed are, however, those of the authors only and do not necessarily reflect those of the EU or the ERC Executive Agency. Neither the EU nor the granting authority can be held responsible for them. The Lifelines initiative has been made possible via a subsidy from the Dutch Ministry of Health, Welfare and Sport, the Dutch Ministry of Economic Affairs, the UMCG, Groningen University and the provinces in the north of the Netherlands (Drenthe, Friesland and Groningen). We acknowledge the services of the Lifelines cohort study and thank the contributing research centers delivering data to Lifelines and all the study participants. We thank all the participating families in Norway who have taken part in the on-going MoBa. Resources provided by Sigma2, the national infrastructure for high performance computing and data storage in Norway, was used for analyses (grant no. NS9867S). We acknowledge the work of the data managers C. Timpe and O. van Jole.

## Author contributions

Study conception and design: R.C. and E.Y. Data acquisition and quality control: R.C., V.A., S.v.A., A.A., H.L., A.G., E.Y. and FinnGen. Statistical analysis: R.C., V.A., S.v.A., A.A. and R.P. Data interpretation: R.C., V.A., S.v.A., A.A., R.P., J.C.E., Z.A., P.A.D., E.M.E., A.F., A.H., H.L., T.H.L., Q.Q., A.G. and E.Y. Manuscript drafting: R.C. Manuscript revision: R.C., V.A., S.v.A., A.A., R.P., J.C.E., Z.A., P.A.D., E.M.E., A.F., A.H., H.L., T.H.L., Q.Q., A.G. and E.Y. All authors approved the submitted version, agreed to be personally accountable for their contributions and ensured appropriate investigation and resolution of questions related to the accuracy or integrity of any part of the work.

## Competing interests

A.G. is founder of Real World Genetics Oy. The other authors declare no competing interests.

## Additional information

**Correspondence and requests for materials** should be addressed to Rosa Cheesman.

# Reporting Summary

## Statistics

For all statistical analyses, confirm that the following items are present in the figure legend, table legend, main text, or Methods section.

| n/a | Confirmed | |
|---|---|---|
| ☐ | ☒ | The exact sample size (*n*) for each experimental group/condition, given as a discrete number and unit of measurement |
| ☐ | ☒ | A statement on whether measurements were taken from distinct samples or whether the same sample was measured repeatedly |
| ☐ | ☒ | The statistical test(s) used AND whether they are one- or two-sided<br>*Only common tests should be described solely by name; describe more complex techniques in the Methods section.* |
| ☐ | ☒ | A description of all covariates tested |
| ☐ | ☒ | A description of any assumptions or corrections, such as tests of normality and adjustment for multiple comparisons |
| ☐ | ☒ | A full description of the statistical parameters including central tendency (e.g. means) or other basic estimates (e.g. regression coefficient) AND variation (e.g. standard deviation) or associated estimates of uncertainty (e.g. confidence intervals) |
| ☐ | ☒ | For null hypothesis testing, the test statistic (e.g. *F*, *t*, *r*) with confidence intervals, effect sizes, degrees of freedom and *P* value noted<br>*Give P values as exact values whenever suitable.* |
| ☒ | ☐ | For Bayesian analysis, information on the choice of priors and Markov chain Monte Carlo settings |
| ☒ | ☐ | For hierarchical and complex designs, identification of the appropriate level for tests and full reporting of outcomes |
| ☐ | ☒ | Estimates of effect sizes (e.g. Cohen's *d*, Pearson's *r*), indicating how they were calculated |

*Our web collection on statistics for biologists contains articles on many of the points above.*

## Software and code

Policy information about availability of computer code

| | |
|---|---|
| Data collection | No software was used for data collection. |
| Data analysis | GWA were performed using FastGWA and REGENIE software. METAL was used to perform meta-analyses. To identify independent genome-wide significant associations in the meta-analytic results, we performed clumping using standard parameters in FUMA software. SNP heritability was esitmated using LD score regression software. Genetic corelations and factor analyses were performed in GenomicSEM R package. PGIs were constructed using SBayesR. Versions: SBayesR within GCTB software version 2.05beta_Linux; gcta_1.91.7beta/gcta64; regenie 2.2.4; Genomic SEM v 0.0.3f in R-4.3.2; METAL for Linux current version 2011-03-25. The code used in this study are available at: https://github.com/rosacheesman/Fields_genetics. |

For manuscripts utilizing custom algorithms or software that are central to the research but not yet described in published literature, software must be made available to editors and reviewers. We strongly encourage code deposition in a community repository (e.g. GitHub). See the Nature Portfolio guidelines for submitting code & software for further information.

# Data

Policy information about availability of data

All manuscripts must include a data availability statement. This statement should provide the following information, where applicable:
- Accession codes, unique identifiers, or web links for publicly available datasets
- A description of any restrictions on data availability
- For clinical datasets or third party data, please ensure that the statement adheres to our policy

Individual level data: The Finnish biobank data can be accessed through the Fingenious® services (https://site.fingenious.fi/en/) managed by FINBB. Finnish Health register data can be applied from Findata (https://findata.fi/en/data/). Instructions for access to MoBa data from the Norwegian Institute of Public Health can be found here: https://www.fhi.no/en/studies/moba/for-forskere-artikler/research-and-data-access/. Instructions for access to the Lifelines Cohort Study can be found here: https://www.lifelines-biobank.com/researchers/working-with-us. Administrative data on the Dutch population for the purpose of academic research is housed by Statistics Netherlands. Access is only possible for authorised instutions. Instructions on how to apply for the usage of administrative data can be found here: https://www.cbs.nl/en-gb/our-services/customised-services-microdata/microdata-conducting-your-own-research.

Summary statistics: The GWAS summary statistics produced from this study are available at: https://zenodo.org/records/15584414; DOI: 10.5281/zenodo.15584414. Summary statistics from each data FinnGen release will be made publicly available after a one-year embargo period and can be accessed freely from: www.finngen.fi/en/access_results.

# Research involving human participants, their data, or biological material

Policy information about studies with human participants or human data. See also policy information about sex, gender (identity/presentation), and sexual orientation and race, ethnicity and racism.

| | |
|---|---|
| Reporting on sex and gender | We primarily use the term 'sex' because the data were obtained from Norwegian Medical Birth Register e.g., "we conducted sex-stratified GWAS in MoBa (sex defined from birth register data) ". |
| Reporting on race, ethnicity, or other socially relevant groupings | We primarily use the term 'ancestry' e.g. "European-associated Ancestry individuals " /"To identify a sub-population of European-associated ancestry" since homogenous populations for genetic analysis were identified using allele frequency clustering. We discuss this in the Discussion: "our GWAS were limited to Norwegian and Finnish individuals with European-associated ancestries. It remains unclear how much the results generalise to people of diverse backgrounds. In the future, genetically sensitive studies of educational field choices should include underrepresented groups and countries, taking systematic social differences into account. " |
| Population characteristics | We use population-wide register data on adults >age 25 from Finland and Norway |
| Recruitment | We describe this for all cohorts used in the GWAS:<br>"Pregnant women were recruited from across Norway from 1999 to 2009. " for MoBa<br>"Participants were sampled from the northern population of the Netherlands, which is about 10% of the region's population. Between 2006 and 2013, randomly selected general practitioners invited all their listed patients aged 25-49 years to participate in the study" for LifeLines<br><br>"The project utilises data from the nationwide longitudinal health register collected since 1969 from every resident in Finland." for FinnGen |
| Ethics oversight | The establishment of MoBa and initial data collection was based on a licence from the Norwegian Data Protection Agency and approval from The Regional Committees for Medical and Health Research Ethics. The MoBa cohort is now based on regulations related to the Norwegian Health Registry Act. The current study was approved by The Regional Committees for Medical and Health Research Ethics (project # 2017/2205).<br><br>Study subjects in FinnGen provided informed consent for biobank research, based on the Finnish Biobank Act. Alternatively, separate research cohorts, collected prior the Finnish Biobank Act came into effect (in September 2013) and start of FinnGen (August 2017), were collected based on study-specific consents and later transferred to the Finnish biobanks after approval by Fimea (Finnish Medicines Agency), the National Supervisory Authority for Welfare and Health. Recruitment protocols followed the biobank protocols approved by Fimea. The Coordinating Ethics Committee of the Hospital District of Helsinki and Uusimaa (HUS) statement number for the FinnGen study is Nr HUS/990/2017.<br><br>The FinnGen study is approved by Finnish Institute for Health and Welfare (permit numbers: THL/2031/6.02.00/2017, THL/1101/5.05.00/2017, THL/341/6.02.00/2018, THL/2222/6.02.00/2018, THL/283/6.02.00/2019, THL/1721/5.05.00/2019 and THL/1524/5.05.00/2020), Digital and population data service agency (permit numbers: VRK43431/2017-3, VRK/6909/2018-3, VRK/4415/2019-3), the Social Insurance Institution (permit numbers: KELA 58/522/2017, KELA 131/522/2018, KELA 70/522/2019, KELA 98/522/2019, KELA 134/522/2019, KELA 138/522/2019, KELA 2/522/2020, KELA 16/522/2020), Findata permit numbers THL/2364/14.02/2020, THL/4055/14.06.00/2020, THL/3433/14.06.00/2020, THL/4432/14.06/2020, THL/5189/14.06/2020, THL/5894/14.06.00/2020, THL/6619/14.06.00/2020, THL/209/14.06.00/2021, THL/688/14.06.00/2021, THL/1284/14.06.00/2021, THL/1965/14.06.00/2021, THL/5546/14.02.00/2020, THL/2658/14.06.00/2021, THL/4235/14.06.00/2021, Statistics Finland (permit numbers: TK-53-1041-17 and TK/143/07.03.00/2020 (earlier TK-53-90-20) TK/1735/07.03.00/2021, TK/3112/07.03.00/2021) and Finnish Registry for Kidney Diseases permission/extract from the meeting minutes on 4th July 2019.<br><br>The Biobank Access Decisions for FinnGen samples and data utilized in FinnGen Data Freeze 11 include: THL Biobank |

BB2017_55, BB2017_111, BB2018_19, BB_2018_34, BB_2018_67, BB2018_71, BB2019_7, BB2019_8, BB2019_26, BB2020_1, BB2021_65, Finnish Red Cross Blood Service Biobank 7.12.2017, Helsinki Biobank HUS/359/2017, HUS/248/2020, HUS/430/2021 §28, §29,  HUS/150/2022 §12, §13, §14, §15, §16, §17, §18, §23, §58 and §59, Auria Biobank AB17-5154 and amendment #1 (August 17 2020) and amendments BB_2021-0140, BB_2021-0156 (August 26 2021, Feb 2 2022), BB_2021-0169, BB_2021-0179, BB_2021-0161,  AB20-5926 and amendment #1 (April 23 2020) and it´s modification (Sep 22 2021), BB_2022-0262, BB_2022-0256, Biobank Borealis of Northern Finland_2017_1013, 2021_5010, 2021_5018, 2021_5015, 2021_5015 Amendment, 2021_5023, 2021_5023 Amendment, 2021_5017, 2022_6001, 2022_6006 Amendment,  BB22-0067, 2022_0262,  Biobank of Eastern Finland 1186/2018 and amendment 22§/2020, 53§/2021, 13§/2022, 14§/2022, 15§/2022, 27§/2022, 28§/2022, 29§/2022, 33§/2022, 35§/2022, 36§/2022, 37§/2022, 39§/2022, 7§/2023, Finnish Clinical Biobank Tampere MH0004 and amendments (21.02.2020 & 06.10.2020), 8§/2021, 9§/2021, §9/2022, §10/2022, §12/2022, 13§/2022, §20/2022, §21/2022, §22/2022, §23/2022, 28§/2022, 29§/2022, 30§/2022, 31§/2022, 32§/2022, 38§/2022, 40§/2022, 42§/2022, 1§/2023, Central Finland Biobank 1-2017, BB_2021-0161, BB_2021-0169, BB_2021-0179, BB_2021-0170, BB_2022-0256, and Terveystalo Biobank STB 2018001 and amendment 25th Aug 2020, Finnish Hematological Registry and Clinical Biobank decision 18th June 2021, Arctic biobank P0844: ARC_2021_1001.

The Lifelines protocol was approved by the UMCG Medical ethical committee under number 2007/152.

Note that full information on the approval of the study protocol must also be provided in the manuscript.

# Field-specific reporting

Please select the one below that is the best fit for your research. If you are not sure, read the appropriate sections before making your selection.

☐ Life sciences          ☒ Behavioural & social sciences          ☐ Ecological, evolutionary & environmental sciences

For a reference copy of the document with all sections, see nature.com/documents/nr-reporting-summary-flat.pdf

# Behavioural & social sciences study design

All studies must disclose on these points even when the disclosure is negative.

| | |
|---|---|
| Study description | Quantitative research - GenomeWide Association Meta-analysis of Educational Fields. |
| Research sample | We required the maximum sample of adults (age 25) with educational records AND genotype data.This meant using Finnish and Norwegian data from educational registries linked to FinnGen and MoBa. Administrative data are of high quality, and do not suffer attrition. Our study only loses representativeness by being restricted to genotyped cohort study participants of European ancestry (necessary to avoid population stratification). The sample is wealthier and healthier than the general populations. |
| Sampling strategy | Our strategy was to collect the maximum sample of participants from European-associated ancestries with genotype and educational phenotype information. Our sample size of >460,000 meets power requirements for identifying SNP associations, and Polygenic Index analyses include well over the minimum of several hundred participants. FinnGen used population-wide registers, MoBa convenience sampling of pregnant families, and Lifelines stratified random sampling via general practitioners. |
| Data collection | We used previously collected data, with collection and recruitment described elsewhere. Phenotypes used were recorded through linked administrative data. Educational field data were extracted from administrative registers using national systems—Statistics Finland, Norwegian Standard Classification of Education (NUS2000) in MoBa, and Statistics Netherlands administrative files ("HOOGSTEOPLTAB 2022, V1") in Lifelines—then harmonized to ISCED 2013 broad field categories. Admin records are described here: https://www2.stat.fi/fi/luokitukset/koulutusala/ and: http://www.ssb.no/en/utdanning/norwegian-standard-classification-of-education. Data collection was prior to initiation of this study. The current analysts were not blinded to study hypotheses. |
| Timing | Education register data includes entire populations year-by-year. We used most recently available data available in both (2018) on adults aged >25. Data collection spanned 06/1999-12/2009 (MoBa), 01/2006-12/2013 (Lifelines), and 01/1969-ongoing (FinnGen administrative data, genetic collection from 2017). |
| Data exclusions | We excluded participants for missingness on phenotypic and genomic variables of study. A total of 5, 601, 162 individuals with educational field data were excluded (2, 857, 143 from Norway and 2, 744, 019 from Finland) due to lack of genetic data. |
| Non-participation | Not applicable --- administrative registers include the total populations. However, not all initial MoBa and FinnGen participants consented to genotyping. |
| Randomization | Not applicable, participants were not allocated into experimental groups. |

# Reporting for specific materials, systems and methods

We require information from authors about some types of materials, experimental systems and methods used in many studies. Here, indicate whether each material, system or method listed is relevant to your study. If you are not sure if a list item applies to your research, read the appropriate section before selecting a response.

## Materials & experimental systems

| n/a | Involved in the study |
|-----|-----------------------|
| ☒ ☐ | Antibodies |
| ☒ ☐ | Eukaryotic cell lines |
| ☒ ☐ | Palaeontology and archaeology |
| ☒ ☐ | Animals and other organisms |
| ☒ ☐ | Clinical data |
| ☒ ☐ | Dual use research of concern |
| ☒ ☐ | Plants |

## Methods

| n/a | Involved in the study |
|-----|-----------------------|
| ☒ ☐ | ChIP-seq |
| ☒ ☐ | Flow cytometry |
| ☒ ☐ | MRI-based neuroimaging |

## Plants

| | |
|---|---|
| Seed stocks | *Report on the source of all seed stocks or other plant material used. If applicable, state the seed stock centre and catalogue number. If plant specimens were collected from the field, describe the collection location, date and sampling procedures.* |
| Novel plant genotypes | *Describe the methods by which all novel plant genotypes were produced. This includes those generated by transgenic approaches, gene editing, chemical/radiation-based mutagenesis and hybridization. For transgenic lines, describe the transformation method, the number of independent lines analyzed and the generation upon which experiments were performed. For gene-edited lines, describe the editor used, the endogenous sequence targeted for editing, the targeting guide RNA sequence (if applicable) and how the editor was applied.* |
| Authentication | *Describe any authentication procedures for each seed stock used or novel genotype generated. Describe any experiments used to assess the effect of a mutation and, where applicable, how potential secondary effects (e.g. second site T-DNA insertions, mosiacism, off-target gene editing) were examined.* |

