## [Peer Review File · Nature Genetics]

Genetic associations with educational fields

Corresponding Author: Dr Rosa Cheesman

Version 0:

Reviewer comments:

Reviewer #1

(Remarks to the Author)

REMARKS TO THE AUTHORS

The two reviewers submitting this referee report are a Ph.D. student and his supervisor.

Cheesman et al. perform a genome-wide association study (GWAS) of educational field choice (college major). There are two key results of the paper. The first is the identification of genetic correlations among field choices and the use of principal component analysis to reduce these choices to two dimensions of genetic variation: a “creative versus conformist” and a “technical versus social.” These dimensions are then analyzed by estimating their genetic correlations with other traits. The second key result is the GWAS summary statistics, which will enable further studies analyzing the genetic covariance between traits and education field choice.

Whilst the results are exciting, we would have thought they would be more interesting to a journal with more of a social-science focus. The large sample size and novelty of studying field choice makes the paper strong. There are, however, a few areas where the manuscript can be improved upon. We have organized our suggestions under a few headings.

GWAS

Did the GWAS use a linear model, or non-linear (logistic, probit etc.) model? We cannot see this information in the paper. A nonlinear model seems most appropriate, especially if one or more covariates has a substantial association.

Similarly, the paper does not seem to explicitly state whether linear or logistic regression was used to validate the polygenic scores (“indices”). Here the case for using logistic regression seems even stronger, because the scores and covariates together might have a rather nontrivial (and hence nonlinear) association with the outcome. Furthermore, the pseudo- R^2 from logistic regression may be useful for downstream analysis by others who want to know the power of the scores or adjust the scores for their reliability. The authors report the R^2 and the “additive R^2 ” from their validation models. We do not know the term additive R^2 and presume the authors mean the incremental R^2 . The term additive R^2 should be defined, or more conventional nomenclature should be used.

At what level of education does field of choice correspond to? Is this only for participants who have completed undergraduate study in university or higher, or is field choice at high school also considered? To discover this requires going to supplementary materials. Perhaps this should be stated more clearly in the main text.

Education was treated as a continuous covariate in the GWAS. Perhaps it should be treated as categorical. If either the independent or dependent variable has a strongly nonlinear relationship with a discrete covariate, treating the latter as categorical can lead to more effective statistical control.

Could Figure 1 present SNP heritability for the traits with and without the control for education? The models controlling for education are perhaps more pivotal to the paper, since the authors use these results for Genomic SEM and the authors are concerned with overcoming the “conventional hierarchy of attainment” by controlling for education. So these results should also be presented upfront. We find plausible the explanation by the authors in a supplementary figure that education might be a collider, but since the results controlling for education are often what are used in the paper downstream, it seems to make sense to include them in Figure 1.

Latent traits

We think that doing a GWAS of the PCs rather than the latent factors is more appropriate, for a number of reasons. In order to avoid some theoretical problems with the factor model, a factor must be susceptible to being measured by an infinite number of indicators, which in the limit take the reliability of measurement as close to one as we please (McDonald, 2003). This is an idealization of course, but it is met with reasonable verisimilitude if the number of potential items that can be written to measure a trait is very large. The fields of study analyzed in this paper seem to us closer to the entire universe of possibilities in this domain rather than a sample of an infinite (i.e., very large) indicator pool, rendering factor analysis inapplicable.

The interpretation of PCA is application specific (e.g., McVean, 2009) or indeed can be completely lacking. One can treat PCA as dimension reduction, full stop. Rather than presenting genetic correlations with 10 different correlated traits, one can present genetic correlations with a smaller number of PCs, presenting a simplified and more digestible summary to readers.

Inspired by the PCA, the authors sought a factor model similar to it. The final factor model, however, is rather unsatisfactory because of its complex cross-loadings. These seem to be necessary to obtain an acceptable fit. In a PCA there is no conception of fit and hence no need to worry about tweaking a model. It would have been better to just stop with the PCA.

Another reason to prefer performing a GWAS of the PCs rather than F1 and F2 is that the effective sample size of the Genomic SEM is very low (~10K). Although this figure is somewhat arbitrary since it really depends upon the presumed heritability of the latent factor, it does imply that there is little power to perform a GWAS of the latent factors. By contrast, with a GWAS of the PCs, the analysis may have more power.

The authors suggest that the appropriate number of PCs to retain is two, writing:

To establish the number of components explaining patterns of genetic correlations between fields, we performed a principal components analysis (PCA). Two PCs were satisfactory, given that they collectively explained 69% of the variance, and that the individual fields GWAS showed strong correlations with the top two PCs but not the third (see Supplementary Table 13 for PCA results).

Whilst the choice of a number of latent factors to extract can be quite subjective, the given explanation for two factors seems insufficiently substantiated. That two PCs explain a lot (69%) of the variance does not mean that a third PC does not also capture much important variation.

Information on how to perform GWAS of the PCs can be found in this link:
<https://annafurtjes.github.io/genomicPCA/index.html>

We know that the issue that we are raising here about factor vs. principal component analysis will seem esoteric and even trivial. We realize that the authors will probably resist our suggestion; for this reason we do not insist that they accept our admittedly minority view, since it probably will not make too much of a difference to the message of their paper. But the meaning of factor analysis has long been dogged by confusion and controversy, and we think that the conception of factor analysis that we have cited here stands up to all challenges.

The developers of Genomic SEM seem to take the view that a factor model can be applied to an inherently limited domain of indicators if a common-pathway model of SNPs acting through the factors tends to show excellent fit (Clapp Sullivan et al., 2024). If the authors want to adopt this view, they should report the Q_SNP statistics calculated by Genomic SEM.

In ST14 only the AIC is given for the different models. Other informative fit statistics are given by GenomicSEM, e.g. SRMR. It would be good to give all the fit statistics available. If other statistics suggest the final model is a poor fit, then please explain this in the text of the paper.

Creative vs. conformist

Although labelling components does involve a degree of subjective judgement, we wonder if a better label can be created for the creative versus conformist scale; in the PCA, this is PC2. Firstly, conformist is not clearly the opposite of creative. Indeed, the authors seem to agree that creative and conformism are not opposite ends of a scale, saying:

Indeed, whilst creativity and conformity have been thought to be diametrically opposed, recent work has highlighted that more conforming individuals can enhance the creative process.

More generally, are the terms apt descriptions? Are people interested in health and welfare really conformist? We are not sure. An interest in practical, hands-on work is what farming, health, engineering/construction, education and services all have in common. We think the conformist description can come across as slightly pejorative, which is acceptable if we are sure the label is accurate. Likewise, business, administration and law do not seem especially creative to us.

Correlation between field choice and cognitive performance/educational attainment

Even after controlling for education, variation in field choice is still strongly correlated with cognitive ability and EA. This is because phenotypic education is only a noisy proxy for genetic variation in EA and intelligence. However, the authors state they want to shift from studying the "conventional hierarchy of attainments towards multidimensional tendencies and interests." We certainly think studying interests and variation in cognitive abilities, orthogonal to the general factor of intelligence, is important. However, the GWAS does not entirely succeed in this goal, in light of the genetic correlations

mentioned above.

We have already mentioned one possible attempt at a fix, which is to treat educational level as a categorical rather than continuous covariate in the GWAS.

An interesting analysis would be to use Genomic SEM to implement a model removing EA or CP variation from the field choices, rendering the latter true residuals. Then recalculate the genetic correlations. This method might be more interesting than controlling for phenotypic education in the GWAS, but it would be understandable if the authors left this for others to attempt.

Minor points

We now make some comments tied to specific bits of the paper.

title: We think it is somewhat misleading to suggest a sample size greater than 460,000.

p. 3: "An accurate, holistic, and hypothesis-free approach to the dimensionality of educational fields would be to factor analyse" Some authorities have deprecated exploratory ("hypothesis-free") factor analysis. Instead of "factor analyse," perhaps say "apply dimension-reducing multivariate techniques to." And perhaps remove the word "accurate" because there is some judgment required in determining which eigenvalues are "large."

p. 5: "statistically indistinguishable from 1 apart from for" There is a typo here.

p. 8: What is an FDR-corrected p-value? FDR and p-value have almost exactly opposite meanings. FDR is the proportion of rejected null hypotheses that are in fact true, whereas p-value is the proportion of true null hypotheses that are rejected. That is, FDR is $P(\text{null is true}|\text{rejection})$ and p-value is $P(\text{rejection}|\text{null is true})$. Some software tools call FDR a method for adjusting p-values, but this is misleading. Please use appropriate terminology. Perhaps the simplest solution is to determine the p-value corresponding to an of FDR of 0.05 and call everything below that threshold significant. Also, the set of hypotheses (all traits? just one trait?) to which the FDR applies needs to be specified with precision.

p. 10: "Engineering, manufacturing, and construction ..." How the fields are designated, with capital letters or not, is confusing. Is there a better way to do this? Italics, quote marks?

Figure 5: What FDR threshold is used for statistical significance? The standard 0.05? This needs to be stated somewhere.

p. 15: "We found that SNP heritability estimates for educational field choices were 7% on average. These are likely to be underestimated because our methodology only captures additive effects of common variants tagged by genotyping arrays, rather than the full heritability (the missing heritability problem)" The wording here seems unclear. SNP heritability is an estimate of the variance explained by the common variants, so it is not necessarily an underestimate of what it is trying to measure.

p. 15: "educational field choices may be smaller than and show little overlap with parental genetic effects on educational attainment" What does this mean? This clause makes no sense.

p. 17: "Individuals with more engineering-associated DNA variants 'should' not necessarily choose engineering and are not qualitatively different to people with more art-associated DNA variants." This sentence veers away from the factual and into the normative. (It is also hard to see why the word `_should_` is in quote marks.) Robert Plomin has argued that individuals with polygenic scores indicating promise for a certain pursuit should be encouraged to take up that pursuit. This is the opposite of what the authors are saying here. The conclusion to be drawn from these contrary statements is that they are not scientific and thus do not belong in scientific papers. Moreover, the second clause is just meaningless. Of course people are different; they are genetically different. What does it mean to say that this difference is not "qualitative?"

p. 22: "SBayesR uses Bayesian shrinkage to correct for linkage disequilibrium" This is incorrect. SBayesR and similar polygenic-score constructions methods use Bayesian shrinkage to deal with sampling error.

p. 24 and Supplementary Table 8: "cube in age" It is more standard to say "cubic in age."

Supplementary Figure 17: "we also avoid making assumptions about causal direction of EA<->fields" Is this a typo? What does this mean?

Supplementary Table 5: There is something wrong with this table. Is it supposed to say "controls" rather than "cases" in the lower part.

Supplementary Table 11: How were the SEs and p-values of the ratios calculated?

References

Clapp Sullivan, M.L. et al. (2024). Beyond the factor indeterminacy problem using genome-wide association data. *Nature Human Behaviour*, 8, 202-218.

McDonald, R.P. (2003). Behavior domains in theory and in practice. *Alberta Journal of Educational Research*, 49, 212-230. doi:10.11575/ajer.v49i3.54980

McVean, G. (2009). A genealogical interpretation of principal components analysis. *PLoS Genetics*, 5, e10000686. doi:10.1534/genetics.111.131540

Reviewer #2

(Remarks to the Author)

Summary. This paper reports a GWAS meta-analysis of educational field of study across three relatively large cohorts (FinnGen, MoBa, and Lifelines). The paper is novel in focusing on field of study, rather than educational attainment (total years of schooling), which has been the focus of much prior work. The paper identifies genome-wide significant SNPs, estimates SNP heritabilities, conducts polygenic prediction analyses, tests for robustness using within-family variation, identifies two main axes of genetic influence on field of study, and examines genetic correlations with a wide range of phenotypes.

Evaluation. This is a creative and intriguing paper. Educational field of study is an important phenotype to study, and the results are interesting, most notably the categorization into Technical vs. Social fields and Creative vs. Conformist fields. The paper has two main limitations. First, the sample sizes are small for some of the fields (e.g., according to Supplementary Table 5, the effective sample size for the field "Natural sciences, mathematics and statistics" is only ~40,000 individuals). This raises the concern that some of the GWAS may be underpowered. While the small sample size per se is not a concern for the SNP heritability and genetic correlation analyses (though contributes to wide confidence intervals), it implies that the specific SNPs identified by the GWAS are more likely to be false positives. Second, it is unclear how much of the genetic signal goes beyond the genetic signal for educational attainment. Although the paper controls for educational attainment in many of the analyses, the control is likely incomplete, as discussed further below.

Suggestions. My main suggestions correspond to the two main limitations listed above. First, while the sample size is what it is (unless the authors can get access to additional cohorts to include in the GWAS meta-analysis), the paper should be clearer about the sample sizes in the GWASs and the fact that the relatively small sample size is a limitation. The title and abstract report a total sample size for FinnGen and MoBa of >460,000 individuals, but I could not find effective sample sizes for the GWASs until I dug into the Supplementary Tables and found Supplementary Table 5. This information should be in the main text. The reporting of the number of SNPs identified in each GWAS should be accompanied by a caveat about the low power for several of the fields.

Second, the paper should use a more sophisticated approach to examining genetic influences on fields of study that go beyond genetic influences on educational attainment. Consider the GWAS on fields of study that controls for educational attainment. Both because educational attainment is influenced by non-genetic factors and because educational attainment is a coarse measure (it is discrete with just a few categories, whereas the genetic factor for educational attainment is essentially continuous), controlling for educational attainment in the GWAS is an incomplete control for genetic influences on educational attainment. A similar concern arises in the GWAS that controls for parental educational fields, since that is an imperfect measure of parents' genetic propensity for educational fields.

One approach to control for the genetic influences on educational attainment would be the GWAS-by-subtraction method of Demange et al. (2021, *Nature Genetics*). The identifying assumption—which may not be plausible—would be that while genetic influences on educational attainment affect field of study, genetic influences on field of study do not affect educational attainment. Another approach would be to control for the polygenic index for educational attainment, along with an errors-in-variables bias correction (as in DiPrete et al., 2018, *PNAS*, or Becker et al., 2021, *Nature Human Behaviour*).

Here are some other specific suggestions that I hope will be useful in improving the paper:

1. The paper should mention the issue that the error terms across the GWASs are correlated because (I think) the educational fields are mutually exclusive. Thus, if a SNP is by chance positively associated with being in one field, then it is mechanically negatively associated with being in other fields. This is therefore a case of seemingly unrelated regressions (SUR; Zellner, 1962, *Journal of the American Statistical Association*). I believe this is not a problem for any of the analyses in the paper because SUR estimation gives the same coefficients as OLS when the regressors in each equation are the same (Amemiya, 1985, *Advanced Econometrics*), but this issue deserves a mention.

2. Lines 220-222: "However, power for within-family analyses in the subsample was low, such that only PGI associations for Arts, humanities and languages and Engineering, manufacturing and construction remained significant." – The paper should report confidence intervals for the decreases.

3. Lines 269-273: Why use PCA rather than (orthogonal) factor analysis, which is more standard for identifying latent factors?

4. Lines 383-384: "such that low genetic correlations capture different influences on different entities rather than on the same entity" – What does this mean?

5. Lines 403-404: "[SNP heritabilities] are likely to be underestimated because our methodology only captures additive effects of common variants tagged by genotyping arrays, rather than the full heritability" – While it is true that broad heritability is likely underestimated, this sentence does not make sense regarding SNP heritability. (LDSC may underestimate SNP heritability but for different reasons than mentioned in this sentence.)

6. Lines 412-414: "Nonetheless, our low SNP heritability estimates leave a large role for environmental factors and random chance in educational field choices." – As the paper acknowledges elsewhere, genetic influences can operate through environmental factors, but the sentence as stated incorrectly implies that a large heritability would rule out a large role for environmental factors. Also, what is "random chance" aside from exogenous environmental factors?

Signed,
Dan Benjamin

Reviewer #3

(Remarks to the Author)

Having now read through this paper several times and thought carefully about it, the first thing to say is that the paper is excellent. There's no doubt that the datasets- the uniqueness of the data structure, the sample size, and the cross-country nature when speaking about educational and training qualifications, is perfectly suited for a decomposition at the phenotypic and genetic level of realms/factors of qualifications. The methods are excellent- when I first opened the paper, I thought I might find myself making several comments about the appropriateness of the methods as I usually do as a reviewer. Instead, the methods read almost like a handbook of how I would go about doing such a study if I had access to this data. So instead- I find most of my comments directed toward the social science going on here:

1) First, the paper suffers from a lack of integration of deepness and richness of the sociological exploration education and occupation that has come over the last several hundred years. I understand that this is a Nature Genetics paper, but some of its conclusions will be most meaningful and impactful in the social sciences. I see sociologists and a sociology department listed in the affiliations, but this paper does not really read like it's trying to speak to the fields that will indeed be most curious or excited about they work. And so I would challenge the authors, with of course the editor's permission, to add a section that deepens the articulation of the literature and just why this topic matters so much for life outcomes. Authors- this is call to action to really dig into the work and previous literature- so often when these sorts of studies are published (I'm thinking about all of the iterations of the genomics of educational attainment work), I think we're left with a very straightforward technical manuscript about an outcome that is deeply important to society, but not understood by geneticists at all. So you need to prime the geneticists to tell them why this is important so that this paper doesn't open itself up to the typical criticism, and you need to take good care of the sociologists who have come to think of geneticists studying these outcomes as something of an affront. This can absolutely be done, but it will take some careful work.

2) In the same vein, I finished the paper's discussion and was left with a bit of a "so what" reaction. I felt like the authors crept to the precipice of giving us some important concluding thoughts on their ability to parse outcomes into technical vs. creative domains or to interpret correlations with health outcomes. But the authors didn't move much beyond this. They posit toward potentially meaningful analyses- for example: "GWAS summary statistics for the two factors and for the individual educational fields could be used to further understand the mechanisms involved in, and consequences of, field choices. For example, PGIs could be used to trace how genetic predispositions manifest in young people's early interests, skills and choices, and how social contexts magnify or suppress certain predispositions (gene-environment interaction questions)." Why not actually do some of these analyses to move us toward meaning a bit more? In Carey et al. published earlier this year in Nature Human Behavior, though they are brief, in the section of the paper dealing with SES factors, the authors point us back toward the importance of these findings for the fields of social science dealing with socioeconomic status and our ability to measure and understand it. These are the sorts of conclusions drawn from psychology and sociology that should be demonstrated - perhaps through some of the additional analyses the authors suggest, but also with a concentrated effort to tell us why these findings matter. What exactly is it we learn here with this unique era of genetic and phenotypic data that moves the social sciences forward in a way that we couldn't get to before this type of data was available? I find myself seeing social science revolutionary work here, but as it is currently presented, a standard reader of NG won't see the innovation here.

3) A frequently Asked Questions (FAQ) is necessary, no doubt. But I implore the authors to do something new and innovative here. Why not create and FAQ website where sum stats are made available, but also where an interactive FAQ is available. The big paper on same-sex sexual behavior had such a website- this seems like a great idea here- <https://geneticsexbehavior.info> I'd push the authors to make this something really special and engaging that can draw readers in.

In summary, this paper is important and compelling, but unless the authors can get this to speak to the correct audiences in a more convincing and rich way, I worry that it will be lost amongst other papers published on topics like these and sort of just "added to the pile."

Decision Letter:

30th August 2024

Dear Rosa,

Your Article "Genetic associations with educational fields in >460,000 individuals" has been seen by three referees. You will

see from their comments below that, while they find your work of interest, they have raised several relevant points. We are interested in the possibility of publishing your study in Nature Genetics, but we would like to consider your response to these points in the form of a revised manuscript before we make a final decision on publication.

To guide the scope of the revisions, the editors discuss the referee reports in detail within the team, including with the chief editor, with a view to identifying key priorities that should be addressed in revision, and sometimes overruling referee requests that are deemed beyond the scope of the current study. In this case, we think the reviewers have provided very thoughtful input, and we ask that you carefully address all technical points related to the association analyses and their interpretation, revising terminology where needed and extending the analyses where feasible as proposed by the reviewers. We also encourage you to further place the work in context with the social sciences literature and extend the discussion of the study's implications. We hope you will find this prioritized set of referee points to be useful when revising your study. Please do not hesitate to get in touch if you would like to discuss these issues further.

We therefore invite you to revise your manuscript taking into account all reviewer and editor comments. Please highlight all changes in the manuscript text file. At this stage, we will need you to upload a copy of the manuscript in MS Word .docx or similar editable format.

*2) If you have not done so already, please begin to revise your manuscript so that it conforms to our Article format instructions, available

[here](http://www.nature.com/ng/authors/article_types/index.html).

*3) Include a revised version of any required Reporting Summary: <https://www.nature.com/documents/nr-reporting-summary.pdf>

Link Redacted

We hope to receive your revised manuscript within 8-12 weeks. If you cannot send it within this time, please let us know.

Nature Genetics is committed to improving transparency in authorship. As part of our efforts in this direction, we are now requesting that all authors identified as 'corresponding author' on published papers create and link their Open Researcher and Contributor Identifier (ORCID) with their account on the Manuscript Tracking System (MTS), prior to acceptance. ORCID helps the scientific community achieve unambiguous attribution of all scholarly contributions. You can create and link your ORCID from the home page of the MTS by clicking on 'Modify my Springer Nature account'. For more information, please visit www.springernature.com/orcid.

Sincerely,

Referee expertise:

Referee #1: Genetics, complex traits, human behavior

Referee #2: Genetics, complex traits, human behavior

Referee #3: Genetics, complex traits, human behavior

Reviewers' Comments:

Reviewer #1:

Remarks to the Author:

REMARKS TO THE AUTHORS

The two reviewers submitting this referee report are a Ph.D. student and his supervisor.

Cheesman et al. perform a genome-wide association study (GWAS) of educational field choice (college major). There are two key results of the paper. The first is the identification of genetic correlations among field choices and the use of principal component analysis to reduce these choices to two dimensions of genetic variation: a "creative versus conformist" and a "technical versus social." These dimensions are then analyzed by estimating their genetic correlations with other traits. The second key result is the GWAS summary statistics, which will enable further studies analyzing the genetic covariance between traits and education field choice.

Whilst the results are exciting, we would have thought they would be more interesting to a journal with more of a social-science focus. The large sample size and novelty of studying field choice makes the paper strong. There are, however, a few areas where the manuscript can be improved upon. We have organized our suggestions under a few headings.

GWAS

Did the GWAS use a linear model, or non-linear (logistic, probit etc.) model? We cannot see this information in the paper. A nonlinear model seems most appropriate, especially if one or more covariates has a substantial association.

Similarly, the paper does not seem to explicitly state whether linear or logistic regression was used to validate the polygenic scores ("indices"). Here the case for using logistic regression seems even stronger, because the scores and covariates together might have a rather nontrivial (and hence nonlinear) association with the outcome. Furthermore, the pseudo- R^2 from logistic regression may be useful for downstream analysis by others who want to know the power of the scores or adjust the scores for their reliability. The authors report the R^2 and the "additive R^2 " from their validation models. We do not know the term additive R^2 and presume the authors mean the incremental R^2 . The term additive R^2 should be defined, or more conventional nomenclature should be used.

At what level of education does field of choice correspond to? Is this only for participants who have completed undergraduate study in university or higher, or is field choice at high school also considered? To discover this requires going to supplementary materials. Perhaps this should be stated more clearly in the main text.

Education was treated as a continuous covariate in the GWAS. Perhaps it should be treated as categorical. If either the independent or dependent variable has a strongly nonlinear relationship with a discrete covariate, treating the latter as categorical can lead to more effective statistical control.

Could Figure 1 present SNP heritability for the traits with and without the control for education? The models controlling for education are perhaps more pivotal to the paper, since the authors use these results for Genomic SEM and the authors are concerned with overcoming the "conventional hierarchy of attainment" by controlling for education. So these results should also be presented upfront. We find plausible the explanation by the authors in a supplementary figure that education might be a collider, but since the results controlling for education are often what are used in the paper downstream, it seems to make sense to include them in Figure 1.

Latent traits

We think that doing a GWAS of the PCs rather than the latent factors is more appropriate, for a number of reasons. In order to avoid some theoretical problems with the factor model, a factor must be susceptible to being measured by an infinite number of indicators, which in the limit take the reliability of measurement as close to one as we please (McDonald, 2003). This is an idealization of course, but it is met with reasonable verisimilitude if the number of potential items that can be written to measure a trait is very large. The fields of study analyzed in this paper seem to us closer to the entire universe of possibilities in this domain rather than a sample of an infinite (i.e., very large) indicator pool, rendering factor analysis inapplicable.

The interpretation of PCA is application specific (e.g., McVean, 2009) or indeed can be completely lacking. One can treat PCA as dimension reduction, full stop. Rather than presenting genetic correlations with 10 different correlated traits, one can present genetic correlations with a smaller number of PCs, presenting a simplified and more digestible summary to readers.

Inspired by the PCA, the authors sought a factor model similar to it. The final factor model, however, is rather unsatisfactory because of its complex cross-loadings. These seem to be necessary to obtain an acceptable fit. In a PCA there is no conception of fit and hence no need to worry about tweaking a model. It would have been better to just stop with the PCA.

Another reason to prefer performing a GWAS of the PCs rather than F1 and F2 is that the effective sample size of the Genomic SEM is very low (~10K). Although this figure is somewhat arbitrary since it really depends upon the presumed heritability of the latent factor, it does imply that there is little power to perform a GWAS of the latent factors. By contrast, with a GWAS of the PCs, the analysis may have more power.

The authors suggest that the appropriate number of PCs to retain is two, writing:

To establish the number of components explaining patterns of genetic correlations between fields, we performed a principal components analysis (PCA). Two PCs were satisfactory, given that they collectively explained 69% of the variance, and that the individual fields GWAS showed strong correlations with the top two PCs but not the third (see Supplementary Table 13 for PCA results).

Whilst the choice of a number of latent factors to extract can be quite subjective, the given explanation for two factors seems insufficiently substantiated. That two PCs explain a lot (69%) of the variance does not mean that a third PC does not also capture much important variation.

Information on how to perform GWAS of the PCs can be found in this link:
<https://annafurtjes.github.io/genomicPCA/index.html>

We know that the issue that we are raising here about factor vs. principal component analysis will seem esoteric and even trivial. We realize that the authors will probably resist our suggestion; for this reason we do not insist that they accept our admittedly minority view, since it probably will not make too much of a difference to the message of their paper. But the meaning of factor analysis has long been dogged by confusion and controversy, and we think that the conception of factor analysis that we have cited here stands up to all challenges.

The developers of Genomic SEM seem to take the view that a factor model can be applied to an inherently limited domain of indicators if a common-pathway model of SNPs acting through the factors tends to show excellent fit (Clapp Sullivan et al., 2024). If the authors want to adopt this view, they should report the Q_SNP statistics calculated by Genomic SEM.

In ST14 only the AIC is given for the different models. Other informative fit statistics are given by GenomicSEM, e.g. SRMR. It would be good to give all the fit statistics available. If other statistics suggest the final model is a poor fit, then please explain this in the text of the paper.

Creative vs. conformist

Although labelling components does involve a degree of subjective judgement, we wonder if a better label can be created for the creative versus conformist scale; in the PCA, this is PC2. Firstly, conformist is not clearly the opposite of creative. Indeed, the authors seem to agree that creative and conformism are not opposite ends of a scale, saying:

Indeed, whilst creativity and conformity have been thought to be diametrically opposed, recent work has highlighted that more conforming individuals can enhance the creative process.

More generally, are the terms apt descriptions? Are people interested in health and welfare really conformist? We are not sure. An interest in _practical_, hands-on work is what farming, health, engineering/construction, education and services all have in common. We think the conformist description can come across as slightly pejorative, which is acceptable if we are sure the label is accurate. Likewise, business, administration and law do not seem especially creative to us.

Correlation between field choice and cognitive performance/educational attainment

Even after controlling for education, variation in field choice is still strongly correlated with cognitive ability and EA. This is because phenotypic education is only a noisy proxy for genetic variation in EA and intelligence. However, the authors state they want to shift from studying the “conventional hierarchy of attainments towards multidimensional tendencies and interests.” We certainly think studying interests and variation in cognitive abilities, orthogonal to the general factor of intelligence, is important. However, the GWAS does not entirely succeed in this goal, in light of the genetic correlations mentioned above.

We have already mentioned one possible attempt at a fix, which is to treat educational level as a categorical rather than continuous covariate in the GWAS.

An interesting analysis would be to use Genomic SEM to implement a model removing EA or CP variation from the field choices, rendering the latter true residuals. Then recalculate the genetic correlations. This method might be more interesting than controlling for phenotypic education in the GWAS, but it would be understandable if the authors left this for others to attempt.

Minor points

We now make some comments tied to specific bits of the paper.

title: We think it is somewhat misleading to suggest a sample size greater than 460,000.

p. 3: "An accurate, holistic, and hypothesis-free approach to the dimensionality of educational fields would be to factor analyse" Some authorities have deprecated exploratory ("hypothesis-free") factor analysis. Instead of "factor analyse," perhaps say "apply dimension-reducing multivariate techniques to." And perhaps remove the word "accurate" because there is some judgment required in determining which eigenvalues are "large."

p. 5: "statistically indistinguishable from 1 apart from for" There is a typo here.

p. 8: What is an FDR-corrected p-value? FDR and p-value have almost exactly opposite meanings. FDR is the proportion of rejected null hypotheses that are in fact true, whereas p-value is the proportion of true null hypotheses that are rejected. That is, FDR is $P(\text{null is true}|\text{rejection})$ and p-value is $P(\text{rejection}|\text{null is true})$. Some software tools call FDR a method for adjusting p-values, but this is misleading. Please use appropriate terminology. Perhaps the simplest solution is to determine the p-value corresponding to an of FDR of 0.05 and call everything below that threshold significant. Also, the set of hypotheses (all traits? just one trait?) to which the FDR applies needs to be specified with precision.

p. 10: "Engineering, manufacturing, and construction ..." How the fields are designated, with capital letters or not, is confusing. Is there a better way to do this? Italics, quote marks?

Figure 5: What FDR threshold is used for statistical significance? The standard 0.05? This needs to be stated somewhere.

p. 15: "We found that SNP heritability estimates for educational field choices were 7% on average. These are likely to be underestimated because our methodology only captures additive effects of common variants tagged by genotyping arrays, rather than the full heritability (the missing heritability problem)" The wording here seems unclear. SNP heritability is an estimate of the variance explained by the common variants, so it is not necessarily an underestimate of what it is trying to measure.

p. 15: "educational field choices may be smaller than and show little overlap with parental genetic effects on educational attainment" What does this mean? This clause makes no sense.

p. 17: "Individuals with more engineering-associated DNA variants 'should' not necessarily choose engineering and are not qualitatively different to people with more art-associated DNA variants." This sentence veers away from the factual and into the normative. (It is also hard to see why the word `_should_` is in quote marks.) Robert Plomin has argued that individuals with polygenic scores indicating promise for a certain pursuit should be encouraged to take up that pursuit. This is the opposite of what the authors are saying here. The conclusion to be drawn from these contrary statements is that they are not scientific and thus do not belong in scientific papers. Moreover, the second clause is just meaningless. Of course people are different; they are genetically different. What does it mean to say that this difference is not "qualitative?"

p. 22: "SBayesR uses Bayesian shrinkage to correct for linkage disequilibrium" This is incorrect. SBayesR and similar polygenic-score constructions methods use Bayesian shrinkage to deal with sampling error.

p. 24 and Supplementary Table 8: "cube in age" It is more standard to say "cubic in age."

Supplementary Figure 17: "we also avoid making assumptions about causal direction of EA<>fields" Is this a typo? What does this mean?

Supplementary Table 5: There is something wrong with this table. Is it supposed to say "controls" rather than "cases" in the lower part.

Supplementary Table 11: How were the SEs and p-values of the ratios calculated?

References

Clapp Sullivan, M.L. et al. (2024). Beyond the factor indeterminacy problem using genome-wide association data. *Nature Human Behaviour*, 8, 202-218.

McDonald, R.P. (2003). Behavior domains in theory and in practice. *Alberta Journal of Educational Research*, 49, 212-230. doi:10.11575/ajer.v49i3.54980

McVean, G. (2009). A genealogical interpretation of principal components analysis. *PLoS Genetics*, 5, e1000686. doi:10.1534/genetics.111.131540

Reviewer #2:

Remarks to the Author:

Summary. This paper reports a GWAS meta-analysis of educational field of study across three relatively large cohorts (FinnGen, MoBa, and Lifelines). The paper is novel in focusing on field of study, rather than educational attainment (total

years of schooling), which has been the focus of much prior work. The paper identifies genome-wide significant SNPs, estimates SNP heritabilities, conducts polygenic prediction analyses, tests for robustness using within-family variation, identifies two main axes of genetic influence on field of study, and examines genetic correlations with a wide range of phenotypes.

Evaluation. This is a creative and intriguing paper. Educational field of study is an important phenotype to study, and the results are interesting, most notably the categorization into Technical vs. Social fields and Creative vs. Conformist fields.

The paper has two main limitations. First, the sample sizes are small for some of the fields (e.g., according to Supplementary Table 5, the effective sample size for the field “Natural sciences, mathematics and statistics” is only ~40,000 individuals). This raises the concern that some of the GWAS may be underpowered. While the small sample size per se is not a concern for the SNP heritability and genetic correlation analyses (though contributes to wide confidence intervals), it implies that the specific SNPs identified by the GWAS are more likely to be false positives. Second, it is unclear how much of the genetic signal goes beyond the genetic signal for educational attainment. Although the paper controls for educational attainment in many of the analyses, the control is likely incomplete, as discussed further below.

Suggestions. My main suggestions correspond to the two main limitations listed above. First, while the sample size is what it is (unless the authors can get access to additional cohorts to include in the GWAS meta-analysis), the paper should be clearer about the sample sizes in the GWASs and the fact that the relatively small sample size is a limitation. The title and abstract report a total sample size for FinnGen and MoBa of >460,000 individuals, but I could not find effective sample sizes for the GWASs until I dug into the Supplementary Tables and found Supplementary Table 5. This information should be in the main text. The reporting of the number of SNPs identified in each GWAS should be accompanied by a caveat about the low power for several of the fields.

Second, the paper should use a more sophisticated approach to examining genetic influences on fields of study that go beyond genetic influences on educational attainment. Consider the GWAS on fields of study that controls for educational attainment. Both because educational attainment is influenced by non-genetic factors and because educational attainment is a coarse measure (it is discrete with just a few categories, whereas the genetic factor for educational attainment is essentially continuous), controlling for educational attainment in the GWAS is an incomplete control for genetic influences on educational attainment. A similar concern arises in the GWAS that controls for parental educational fields, since that is an imperfect measure of parents' genetic propensity for educational fields.

One approach to control for the genetic influences on educational attainment would be the GWAS-by-subtraction method of Demange et al. (2021, Nature Genetics). The identifying assumption — which may not be plausible — would be that while genetic influences on educational attainment affect field of study, genetic influences on field of study do not affect educational attainment. Another approach would be to control for the polygenic index for educational attainment, along with an errors-in-variables bias correction (as in DiPrete et al., 2018, PNAS, or Becker et al., 2021, Nature Human Behaviour).

Here are some other specific suggestions that I hope will be useful in improving the paper:

1. The paper should mention the issue that the error terms across the GWASs are correlated because (I think) the educational fields are mutually exclusive. Thus, if a SNP is by chance positively associated with being in one field, then it is mechanically negatively associated with being in other fields. This is therefore a case of seemingly unrelated regressions (SUR; Zellner, 1962, Journal of the American Statistical Association). I believe this is not a problem for any of the analyses in the paper because SUR estimation gives the same coefficients as OLS when the regressors in each equation are the same (Amemiya, 1985, Advanced Econometrics), but this issue deserves a mention.
2. Lines 220-222: “However, power for within-family analyses in the subsample was low, such that only PGI associations for Arts, humanities and languages and Engineering, manufacturing and construction remained significant.” – The paper should report confidence intervals for the decreases.
3. Lines 269-273: Why use PCA rather than (orthogonal) factor analysis, which is more standard for identifying latent factors?
4. Lines 383-384: “such that low genetic correlations capture different influences on different entities rather than on the same entity” – What does this mean?
5. Lines 403-404: “[SNP heritabilities] are likely to be underestimated because our methodology only captures additive effects of common variants tagged by genotyping arrays, rather than the full heritability” – While it is true that broad heritability is likely underestimated, this sentence does not make sense regarding SNP heritability. (LDSC may underestimate SNP heritability but for different reasons than mentioned in this sentence.)
6. Lines 412-414: “Nonetheless, our low SNP heritability estimates leave a large role for environmental factors and random chance in educational field choices.” – As the paper acknowledges elsewhere, genetic influences can operate through environmental factors, but the sentence as stated incorrectly implies that a large heritability would rule out a large role for environmental factors. Also, what is “random chance” aside from exogenous environmental factors?

Signed,
Dan Benjamin

Reviewer #3:

Remarks to the Author:

Having now read through this paper several times and thought carefully about it, the first thing to say is that the paper is excellent. There's no doubt that the datasets - the uniqueness of the data structure, the sample size, and the cross-country nature when speaking about educational and training qualifications, is perfectly suited for a decomposition at the phenotypic and genetic level of realms/factors of qualifications. The methods are excellent - when I first opened the paper, I thought I might find myself making several comments about the appropriateness of the methods as I usually do as a reviewer. Instead, the methods read almost like a handbook of how I would go about doing such a study if I had access to this data. So instead - I find most of my comments directed toward the social science going on here:

1) First, the paper suffers from a lack of integration of deepness and richness of the sociological exploration education and occupation that has come over the last several hundred years. I understand that this is a Nature Genetics paper, but some of its conclusions will be most meaningful and impactful in the social sciences. I see sociologists and a sociology department listed in the affiliations, but this paper does not really read like it's trying to speak to the fields that will indeed be most curious or excited about they work. And so I would challenge the authors, with of course the editor's permission, to add a section that deepens the articulation of the literature and just why this topic matters so much for life outcomes. Authors - this is call to action to really dig into the work and previous literature - so often when these sorts of studies are published (I'm thinking about all of the iterations of the genomics of educational attainment work), I think we're left with a very straightforward technical manuscript about an outcome that is deeply important to society, but not understood by geneticists at all. So you need to prime the geneticists to tell them why this is important so that this paper doesn't open itself up to the typical criticism, and you need to take good care of the sociologists who have come to think of geneticists studying these outcomes as something of an affront. This can absolutely be done, but it will take some careful work.

2) In the same vein, I finished the paper's discussion and was left with a bit of a "so what" reaction. I felt like the authors crept to the precipice of giving us some important concluding thoughts on their ability to parse outcomes into technical vs. creative domains or to interpret correlations with health outcomes. But the authors didn't move much beyond this. They posit toward potentially meaningful analyses - for example: "GWAS summary statistics for the two factors and for the individual educational fields could be used to further understand the mechanisms involved in, and consequences of, field choices. For example, PGIs could be used to trace how genetic predispositions manifest in young people's early interests, skills and choices, and how social contexts magnify or suppress certain predispositions (gene-environment interaction questions)." Why not actually do some of these analyses to move us toward meaning a bit more? In Carey et al. published earlier this year in Nature Human Behavior, though they are brief, in the section of the paper dealing with SES factors, the authors point us back toward the importance of these findings for the fields of social science dealing with socioeconomic status and our ability to measure and understand it. These are the sorts of conclusions drawn from psychology and sociology that should be demonstrated - perhaps through some of the additional analyses the authors suggest, but also with a concentrated effort to tell us why these findings matter. What exactly is it we learn here with this unique era of genetic and phenotypic data that moves the social sciences forward in a way that we couldn't get to before this type of data was available? I find myself seeing social science revolutionary work here, but as it is currently presented, a standard reader of NG won't see the innovation here.

3) A frequently Asked Questions (FAQ) is necessary, no doubt. But I implore the authors to do something new and innovative here. Why not create and FAQ website where sum stats are made available, but also where an interactive FAQ is available. The big paper on same-sex sexual behavior had such a website - this seems like a great idea here - <https://geneticsexbehavior.info> . I'd push the authors to make this something really special and engaging that can draw readers in.

In summary, this paper is important and compelling, but unless the authors can get this to speak to the correct audiences in a more convincing and rich way, I worry that it will be lost amongst other papers published on topics like these and sort of just "added to the pile."

Version 1:

Reviewer comments:

Reviewer #1

(Remarks to the Author)

The two reviewers submitting this referee report are a Ph.D. student and his supervisor.

The paper has been improved for clarity in many areas. The chief methodological changes are the use GWAS-by-Subtraction and genomic PCA. Both of these alterations represent improvements in focusing in on field choice within levels of educational aptitudes. There are a few areas, discussed below, where greater changes could have been made in response to our suggestions. However, these are small issues we are happy to overlook. There are some major issues stopping us from supporting publication.

We have organized our suggestions under a series of headings. Only the major suggestions are mandatory for our approval. The minor ones are up to the discretion of the authors.

MAJOR ISSUES

PC EA correlations

The genetic correlations between PCs 1 and 2 and EA are now -0.006 and -0.21, respectively.

After GWAS-by-subtraction, the correlations between PCs and EA should be exactly zero. Each PC is a linear combination of the field choices. If GWAS-by-Subtraction has been performed successfully, then these field choices should be uncorrelated with EA, and thus any linear combination of them will still be uncorrelated with EA.

A correlation of -0.006 can be explained by slight mathematical imprecision, -0.21 cannot be. There would seem to be an error here, unless EA is taken from a new validation sample.

To debug the issue, you could first check that GWAS-by-Subtraction has been performed successfully on each field choice. After subtraction, does each field choice have a 0 genetic correlation with EA? If it has been success, something perhaps has gone wrong in creating the PCs.

If the GWAS-by-Subtraction failed to remove associations with EA, then the problem might be with GenomicSEM. In this case, we recommend using the GSUB R package. This performs GWAS-by-Subtraction using a closed-form solution rather than producing an SEM for each SNP. As such, it is much faster and much simpler, meaning that there are fewer things that can go wrong.

<https://github.com/qlu-lab/GSUB>

Incorrect Eigenvalues

In spreadsheet ST17 the eigenvalues do not correspond to the loadings. Calculating the sum of squared loadings for the first component we obtain an eigenvalue of 1.86, but the value reported in the spreadsheet is 2.63.

The order of the components appears to be incorrect too. The components are not listed in order of descending eigenvalues. For example, component 3 has an eigenvalue of 1.38, but component 4 has an eigenvalue of 1.65.

We've looked back at the original submission and the eigenvalues were calculated correctly then and they were in descending order. So the error has been added to the paper.

MINOR ISSUES

"We then estimated SNP-based heritability of the residual GWAS summary statistics."

We believe that this sentence added in revisions, and all related material, should be removed. The developers of Genomic SEM have explicitly recommended that summary statistics produced by latent-variable modeling not be used as a basis for heritability estimates (e.g., Mallard et al., 2022). Environmental variance is not modeled in this approach and so therefore heritability (which includes environmental variance in the denominator) is undefined. The "heritability" estimated in this way can also depend on irrelevant choices such as the identification strategy (unit loading vs. unit variance).

It is possible that the authors took this approach after reading our comments in the first round. We are very sorry about this. We should have anticipated this problem. Perhaps the authors can retain their summary statistics with education as a covariate and, whenever necessary, use those to estimate heritabilities.

Mallard, T.T. et al. (2022). Multivariate GWAS of psychiatric disorders and their cardinal symptoms reveal two dimensions of cross-cutting genetic liabilities. *Cell Genomics*, 2, 100140.

Figure 4a)

The figure shows genetic correlations between educational fields adjusted for educational attainment using GWAS-by-subtraction.

It might be nice to report the genetic correlations before GWAS-by-Subtraction, as well as after. This could be done by using one side of the diagonal for 'before' correlations and the other for 'after' correlations. Additionally, reporting the EA correlations in this table would help explain why this approach was felt to be important in the first place.

Citation Format

"We then performed GWAS of the principal components (following) 50"

Some of the text is written as if the citation uses author-date style (e.g. APA), like in the above example. These ought to be fixed at least in the proofing stage.

Fertility Indicators

“Negative genetic correlations were also observed with two fertility indicators ($r_g = -0.19$ and -0.15).”

Which fertility indicators do these correspond to?

Figure design

Figure 5

Variable names overlap which is a bit ugly.

Figures sometimes appear to have low resolution. Saving the images from R in a vector format (e.g. pdf) should help with this issue.

Levels of Education

The paper has been altered to make clear that field choice is also studied prior to university. This is very helpful, but a statistic might make it even clearer what sort of education is being studied. E.g. X% of participants had undergraduate degrees versus higher levels of education. The supplementary materials give years of education by field choice, so it still requires a bit of investigation and calculation to work out whether this is a study mostly of high school or university choices.

Justification of factor analysis

If it is all the same to the authors, we suggest amendments of one paragraph to produce the following:

“We chose to use PCA rather than confirmatory factor analysis (CFA) for studying the dimensionality of educational fields for several reasons. First, PCA involves fewer assumptions than CFA. While CFA models latent factors assumed to represent real traits measured by field choices, PCA simply reduces the dimensionality of the data by finding axes that explain maximum variation. Additionally, CFA might require the somewhat ad hoc addition of cross-loadings on the basis of modification indices in order to achieve a good fit, whereas PCA does not involve model-fit considerations. Second, the limited number of possible fields (10 indicators) makes PCA more suitable. CFA is ideally designed for studying latent factors that can be measured by an extensive, potentially infinite, number of indicators, which is not the case here.”

If the authors have some reason for preferring their original wording, then we will not wrangle with them.

False discovery rates

We do not understand the authors' revision. We are beginning to fear that the authors do not understand what FDR means; we are discovering that this is a common gap in the knowledge of our students and colleagues. We apologize for sounding patronizing, but nevertheless feel compelled to recommend the relevant material in the book Efron and Hastie (2016) and references therein. Once they understand what FDR is, the authors should address our comments. If they already understand what FDR is, they need to strive for greater clarity.

Efron, B., & Hastie, T. (2016). Computer age statistical inference: Algorithms, evidence, and data science. Cambridge University Press.

**“social science context to the foreground”*

Although prompted by another reviewer, this expansion of “social science context” is dubious. Why do the authors fail to mention the possibility (nay, near certainty) of innate evolved differences between men and women in educational and occupational preferences? This failure will suggest to the discerning reader that this is a slanted presentation of the issues. Unless the authors can be more thorough (with the downside of being more prolix), we suggest pruning this part back.

How many PCs?

In response to our criticism that the extraction of two factors is insufficiently justified, all justification is removed altogether. We were hoping that perhaps more factors might be extracted and analyzed, even if only in the supplementary materials. But removing any explanation for the choice altogether is a deterioration rather than an improvement. The explanation for the author's choice seems to be simplicity — two PCs means fewer things to focus on. This explanation would be good enough.

**“Horizontal stratification”*

This phrase, although perhaps inserted in response to reviewer comments, is more than a bit awkward. “Stratification” is derived from “stratum,” which according to the New Oxford American Dictionary means “a layer or series of layers of rock in the ground.” Strata are thus always stacked vertically.

(Remarks on code availability)

Reviewer #2

(Remarks to the Author)

I liked the originally submitted manuscript, and I think the revised manuscript is even stronger. The authors have been responsive to my main concerns. Moreover, I think the addition of an FAQ, which the authors have disseminated widely, was a great idea.

I have only a few remaining, minor comments:

Lines 247 and 402: The paper refers to the coefficients on the parents' PGIs as estimates of indirect genetic effects, but unlike the coefficients on the proband's PGI, the parental coefficient is biased by population stratification and other gene-environment correlation. I suggest clarifying/rephrasing.

Line 285: Typo: "used applied"

FAQ lines 273-275: "It has not been possible to examine how fields of study hang together since each person usually only studies one field. Genetic methods overcome this problem." – I don't think this will make sense to a reader (even an expert). I know it's hard to explain to a non-expert, but I think it needs some work.

FAQ lines 517-518: I like the sentiment, but these lines come out of nowhere, which risks making them seem just like an effort at political correctness. I suggest adding a transition.

Signed,
Dan Benjamin

(Remarks on code availability)

I did not attempt to run the code, but I did look it over. For the most part, it looks clear and well commented. However, files 4 and 5 under "meta" are missing.

Reviewer #3

(Remarks to the Author)

The authors have done an extraordinary job incorporating sociological explanations and addressing the ethical considerations that I raised, as well as a fantastic job addressing the comments of the other 2 reviewers. This paper is now in line with the best practices in the field of sociogenomics/behavioral genetics, and it reads very well for both a social science and genetics audience. It is imaginative, rigorous, and original in demonstrating the value added from incorporating genetics into the social sciences and vice versa. I congratulate the authors and fully support the publication of this paper in Nature Genetics.

(Remarks on code availability)

I assessed all code posted on GitHub. It all seems satisfactory!

Decision Letter:

17th January 2025

Dear Rosa,

Your revised Article "Genetic associations with educational fields" has been seen by the original referees. You will see from their comments below that, while Reviewers #2 and #3 are satisfied with the revision pending minor clarifications, Reviewer #1 has a few remaining concerns. We remain interested in the possibility of publishing your study in Nature Genetics, but we would like to consider your response to these concerns in the form of a further revision before we make a final decision on publication.

As before, to guide the scope of the revisions, the editors discuss the referee reports in detail within the team, including with the chief editor, with a view to identifying key priorities that should be addressed in revision. In this case, we ask that you address all remaining technical points, performing additional analyses where needed, and revise the text and display items accordingly taking into account all referee comments and suggestions. We again hope you will find this prioritized set of referee points to be useful when revising your study. Please do not hesitate to get in touch if you would like to discuss these issues further.

We therefore invite you to revise your manuscript taking into account all reviewer and editor comments. Please highlight all changes in the manuscript text file. At this stage, we will need you to upload a copy of the manuscript in MS Word .docx or similar editable format.

*2) If you have not done so already, please begin to revise your manuscript so that it conforms to our Article format instructions, available

http://www.nature.com/ng/authors/article_types/index.html here.

*3) Include a revised version of any required Reporting Summary: <https://www.nature.com/documents/nr-reporting-summary.pdf>

Please be aware of our <https://www.nature.com/nature-research/editorial-policies/image-integrity> guidelines on digital image standards.

EXTENDED DATA FIGURES

Link Redacted

We hope to receive your revised manuscript within 4-8 weeks. If you cannot send it within this time, please let us know.

Nature Genetics is committed to improving transparency in authorship. As part of our efforts in this direction, we are now requesting that all authors identified as 'corresponding author' on published papers create and link their Open Researcher and Contributor Identifier (ORCID) with their account on the Manuscript Tracking System (MTS), prior to acceptance. ORCID helps the scientific community achieve unambiguous attribution of all scholarly contributions. You can create and link your ORCID from the home page of the MTS by clicking on 'Modify my Springer Nature account'. For more information, please visit www.springernature.com/orcid.

Sincerely,

Referee expertise:

Referee #1: Genetics, complex traits, human behavior

Referee #2: Genetics, complex traits, human behavior

Referee #3: Genetics, complex traits, human behavior

Reviewers' Comments:

Reviewer #1 (Remarks to the Author):

The two reviewers submitting this referee report are a Ph.D. student and his supervisor.

The paper has been improved for clarity in many areas. The chief methodological changes are the use GWAS-by-Subtraction and genomic PCA. Both of these alterations represent improvements in focusing in on field choice within levels of educational aptitudes. There are a few areas, discussed below, where greater changes could have been made in response to our suggestions. However, these are small issues we are happy to overlook. There are some major issues stopping us from supporting publication.

We have organized our suggestions under a series of headings. Only the major suggestions are mandatory for our approval. The minor ones are up to the discretion of the authors.

MAJOR ISSUES

PC EA correlations

The genetic correlations between PCs 1 and 2 and EA are now -0.006 and -0.21, respectively.

After GWAS-by-Subtraction, the correlations between PCs and EA should be exactly zero. Each PC is a linear combination of the field choices. If GWAS-by-Subtraction has been performed successfully, then these field choices should be uncorrelated with EA, and thus any linear combination of them will still be uncorrelated with EA.

A correlation of -0.006 can be explained by slight mathematical imprecision, -0.21 cannot be. There would seem to be an error here, unless EA is taken from a new validation sample.

To debug the issue, you could first check that GWAS-by-Subtraction has been performed successfully on each field choice. After subtraction, does each field choice have a 0 genetic correlation with EA? If it has been successful, something perhaps has gone wrong in creating the PCs.

If the GWAS-by-Subtraction failed to remove associations with EA, then the problem might be with Genomic SEM. In this case, we recommend using the GSUB R package. This performs GWAS-by-Subtraction using a closed-form solution rather than producing an SEM for each SNP. As such, it is much faster and much simpler, meaning that there are fewer things that can go wrong.

<https://github.com/qlu-lab/GSUB>

Incorrect Eigenvalues

In spreadsheet ST17, the eigenvalues do not correspond to the loadings. Calculating the sum of squared loadings for the first component, we obtain an eigenvalue of 1.86, but the value reported in the spreadsheet is 2.63.

The order of the components appears to be incorrect too. The components are not listed in order of descending eigenvalues. For example, component 3 has an eigenvalue of 1.38, but component 4 has an eigenvalue of 1.65.

We've looked back at the original submission and the eigenvalues were calculated correctly then and they were in descending order. So the error has been added to the paper.

MINOR ISSUES

***"We then estimated SNP-based heritability of the residual GWAS summary statistics."**

We believe that this sentence added in revisions, and all related material, should be removed. The developers of Genomic SEM have explicitly recommended that summary statistics produced by latent-variable modeling not be used as a basis for heritability estimates (e.g., Mallard et al., 2022). Environmental variance is not modeled in this approach and so therefore heritability (which includes environmental variance in the denominator) is undefined. The "heritability" estimated in this way can also depend on irrelevant choices such as the identification strategy (unit loading vs. unit variance).

It is possible that the authors took this approach after reading our comments in the first round. We are very sorry about this. We should have anticipated this problem. Perhaps the authors can retain their summary statistics with education as a covariate and, whenever necessary, use those to estimate heritabilities.

Mallard, T.T. et al. (2022). Multivariate GWAS of psychiatric disorders and their cardinal symptoms reveal two dimensions of cross-cutting genetic liabilities. *Cell Genomics*, 2, 100140.

Figure 4a)

The figure shows genetic correlations between educational fields adjusted for educational attainment using GWAS-by-Subtraction.

It might be nice to report the genetic correlations before GWAS-by-Subtraction, as well as after. This could be done by using one side of the diagonal for 'before' correlations and the other for 'after' correlations. Additionally, reporting the EA correlations in this table would help explain why this approach was felt to be important in the first place.

Citation Format

"We then performed GWAS of the principal components (following) 50"

Some of the text is written as if the citation uses author-date style (e.g. APA), like in the above example. These ought to be fixed at least in the proofing stage.

Fertility Indicators

"Negative genetic correlations were also observed with two fertility indicators ($r_g = -0.19$ and -0.15)."

Which fertility indicators do these correspond to?

Figure design

Figure 5

Variable names overlap which is a bit ugly.

Figures sometimes appear to have low resolution. Saving the images from R in a vector format (e.g. pdf) should help with this issue.

Levels of Education

The paper has been altered to make clear that field choice is also studied prior to university. This is very helpful, but a statistic might make it even clearer what sort of education is being studied, e.g., X% of participants had undergraduate degrees versus higher levels of education. The supplementary materials give years of education by field choice, so it still requires a bit of investigation and calculation to work out whether this is a study mostly of high school or university choices.

Justification of factor analysis

If it is all the same to the authors, we suggest amendments of one paragraph to produce the following:

"We chose to use PCA rather than confirmatory factor analysis (CFA) for studying the dimensionality of educational fields for several reasons. First, PCA involves fewer assumptions than CFA. While CFA models latent factors assumed to represent real traits measured by field choices, PCA simply reduces the dimensionality of the data by finding axes that explain maximum variation. Additionally, CFA might require the somewhat ad hoc addition of cross-loadings on the basis of modification indices in order to achieve a good fit, whereas PCA does not involve model-fit considerations. Second, the limited number of possible fields (10 indicators) makes PCA more suitable. CFA is ideally designed for studying latent factors that can be measured by an extensive, potentially infinite, number of indicators, which is not the case here."

If the authors have some reason for preferring their original wording, then we will not wrangle with them.

False discovery rates

We do not understand the authors' revision. We are beginning to fear that the authors do not understand what FDR means; we are discovering that this is a common gap in the knowledge of our students and colleagues. We apologize for sounding patronizing, but nevertheless feel compelled to recommend the relevant material in the book Efron and Hastie (2016) and references therein. Once they understand what FDR is, the authors should address our comments. If they already understand what FDR is, they need to strive for greater clarity.

Efron, B., & Hastie, T. (2016). Computer age statistical inference: Algorithms, evidence, and data science. Cambridge University Press.

**"social science context to the foreground"

Although prompted by another reviewer, this expansion of "social science context" is dubious. Why do the authors fail to mention the possibility (nay, near certainty) of innate evolved differences between men and women in educational and occupational preferences? This failure will suggest to the discerning reader that this is a slanted presentation of the issues. Unless the authors can be more thorough (with the downside of being more prolix), we suggest pruning this part back.

How many PCs?

In response to our criticism that the extraction of two factors is insufficiently justified, all justification is removed altogether. We were hoping that perhaps more factors might be extracted and analyzed, even if only in the supplementary materials. But removing any explanation for the choice altogether is a deterioration rather than an improvement. The explanation for the

author's choice seems to be simplicity — two PCs means fewer things to focus on. This explanation would be good enough.

"Horizontal stratification"

This phrase, although perhaps inserted in response to reviewer comments, is more than a bit awkward. "Stratification" is derived from "stratum," which according to the New Oxford American Dictionary means "a layer or series of layers of rock in the ground." Strata are thus always stacked vertically.

Reviewer #2 (Remarks to the Author):

I liked the originally submitted manuscript, and I think the revised manuscript is even stronger. The authors have been responsive to my main concerns. Moreover, I think the addition of an FAQ, which the authors have disseminated widely, was a great idea.

I have only a few remaining, minor comments:

Lines 247 and 402: The paper refers to the coefficients on the parents' PGIs as estimates of indirect genetic effects, but unlike the coefficients on the proband's PGI, the parental coefficient is biased by population stratification and other gene-environment correlation. I suggest clarifying/rephrasing.

Line 285: Typo: "used applied"

FAQ lines 273-275: "It has not been possible to examine how fields of study hang together since each person usually only studies one field. Genetic methods overcome this problem." – I don't think this will make sense to a reader (even an expert). I know it's hard to explain to a non-expert, but I think it needs some work.

FAQ lines 517-518: I like the sentiment, but these lines come out of nowhere, which risks making them seem just like an effort at political correctness. I suggest adding a transition.

Signed,
Dan Benjamin

Reviewer #2 (Remarks on code availability):

I did not attempt to run the code, but I did look it over. For the most part, it looks clear and well commented. However, files 4 and 5 under "meta" are missing.

Reviewer #3 (Remarks to the Author):

The authors have done an extraordinary job incorporating sociological explanations and addressing the ethical considerations that I raised, as well as a fantastic job addressing the comments of the other 2 reviewers. This paper is now in line with the best practices in the field of sociogenomics/behavioral genetics, and it reads very well for both a social science and genetics audience. It is imaginative, rigorous, and original in demonstrating the value added from incorporating genetics into the social sciences and vice versa. I congratulate the authors and fully support the publication of this paper in Nature Genetics.

Reviewer #3 (Remarks on code availability):

I assessed all code posted on GitHub. It all seems satisfactory!

Version 2:

Reviewer comments:

Reviewer #1

(Remarks to the Author)

We thank the authors for going above and beyond in response to our review. We are especially thankful for their thorough investigation into the curious GWAS-by-Subtraction results. The manuscript is now excellent, and it should be published.

(Remarks on code availability)

Decision Letter:

Our ref: NG-A66123R1

24th March 2025

Dear Rosa,

Your revised manuscript "Genetic associations with educational fields" (NG-A66123R1) has been seen by Reviewer #1. As you will see from the comments below, Reviewer #1 is satisfied with the revision and has no remaining requests, and therefore we will be happy in principle to publish your study in Nature Genetics as an Article pending final revisions to comply with our editorial and formatting guidelines.

We are now performing detailed checks on your paper, and we will send you a checklist detailing our editorial and formatting requirements soon. Please do not upload the final materials or make any revisions until you receive this additional information from us.

Thank you again for your interest in Nature Genetics. Please do not hesitate to contact me if you have any questions.

Sincerely,

Reviewer #1 (Remarks to the Author):

We thank the authors for going above and beyond in response to our review. We are especially thankful for their thorough investigation into the curious GWAS-by-Subtraction results. The manuscript is now excellent, and it should be published.

Version 3:

Decision Letter:

In reply please quote: NG-A66123R2 Cheesman

1st October 2025

Dear Rosa,

I am delighted to say that your manuscript "Genetic associations with educational fields" has been accepted for publication in an upcoming issue of Nature Genetics.

Over the next few weeks, your paper will be copyedited to ensure that it conforms to Nature Genetics style. Once your paper is typeset, you will receive an email with a link to choose the appropriate publishing options for your paper, and our Author Services team will be in touch regarding any additional information that may be required.

Your paper will be published online after we receive your corrections and will appear in print in the next available issue. You can find out your date of online publication by contacting the Nature Press Office (press@nature.com) after sending your e-proof corrections.

You may wish to make your media relations office aware of your accepted publication, in case they consider it appropriate to organize some internal or external publicity. Once your paper has been scheduled, you will receive an email confirming the publication details. This is normally 3-4 working days in advance of publication. If you need additional notice of the date and time of publication, please let the production team know when you receive the proof of your article to ensure there is sufficient time to coordinate. Further information on our embargo policies can be found here: <https://www.nature.com/authors/policies/embargo.html>

Before your paper is published online, we will distribute a press release to news organizations worldwide, which may very well include details of your work. We are happy for your institution or funding agency to prepare its own press release, but it must mention the embargo date and Nature Genetics. Our Press Office may contact you closer to the time of publication, but if you or your Press Office have any enquiries in the meantime, please contact press@nature.com.

Authors may need to take specific actions to achieve compliance with funder and institutional open access

mandates. If your research is supported by a funder that requires immediate open access (e.g. according to [Plan S principles](https://www.springernature.com/gp/open-science/plan-s-compliance) or the [NIH public access policy](https://www.springernature.com/gp/open-science/us-federal-agency-compliance)), then you should select the gold OA route, and we will direct you to the compliant route where possible. Because authors warrant under our subscription licensing terms that they haven't committed to licensing any version of their article under a license inconsistent with the terms of our agreement – including the applicable embargo period – publication under the subscription model isn't suitable for authors whose funders require no embargo.

If you have not already done so, we recommend that you upload the step-by-step protocols used in this manuscript to protocols.io. protocols.io is an open online resource that allows researchers to share their detailed experimental know-how. All uploaded protocols are made freely available and are assigned DOIs for ease of citation. Protocols can be linked to any publications in which they are used and will be linked to from your article. You can also establish a dedicated workspace to collect all your lab Protocols. By uploading your Protocols to protocols.io, you are enabling researchers to more readily reproduce or adapt the methodology you use, as well as increasing the visibility of your protocols and papers. Upload your Protocols at <https://protocols.io>. Further information can be found at <https://www.protocols.io/help/publish-articles>.

Sincerely,

Click here if you would like to recommend Nature Genetics to your librarian
<http://www.nature.com/subscriptions/recommend.html#forms>

**Visit the Springer Nature Editorial and Publishing website at http://editorial-jobs.springernature.com?utm_source=ejP_NGen_email&utm_medium=ejP_NGen_email&utm_campaign=ejp_NGen for more information about our career opportunities. If you have any questions, please click [here](mailto:editorial.publishing.jobs@springernature.com).

Response to referees

Reviewers' Comments:

Reviewer #1:

Remarks to the Author:

REMARKS TO THE AUTHORS

The two reviewers submitting this referee report are a Ph.D. student and his supervisor.

Cheesman et al. perform a genome-wide association study (GWAS) of educational field choice (college major). There are two key results of the paper. The first is the identification of genetic correlations among field choices and the use of principal component analysis to reduce these choices to two dimensions of genetic variation: a “creative versus conformist” and a “technical versus social.” These dimensions are then analyzed by estimating their genetic correlations with other traits. The second key result is the GWAS summary statistics, which will enable further studies analyzing the genetic covariance between traits and education field choice.

Whilst the results are exciting, we would have thought they would be more interesting to a journal with more of a social-science focus. The large sample size and novelty of studying field choice makes the paper strong. There are, however, a few areas where the manuscript can be improved upon. We have organized our suggestions under a few headings.

GWAS

Did the GWAS use a linear model, or non-linear (logistic, probit etc.) model? We cannot see this information in the paper. A nonlinear model seems most appropriate, especially if one or more covariates has a substantial association.

Thank you for raising this important point. We have expanded the Methods section to explain our choice of logistic GWA methods.

“We performed genome-wide association analyses (GWA) for 10 dichotomous broad educational field phenotypes in MoBa and FinnGen, using models developed for resource-efficient analysis of case-control phenotypes in biobank-scale datasets. In MoBa, the FastGWA-GLMM approach in GCTA software (--fastGWA-mlm-binary)⁷² was used. This is a logistic regression model with the added complexity of using a sparse matrix to account for the dense genetic relatedness in MoBa. FastGWA-GLMM also uses the saddle point approximation (SPA) method to account for inflation in test statistics due to case-control imbalance. In FinnGen, the binary option in REGENIE⁷³ software was used. REGENIE is a machine learning method that implements whole-genome ridge regression and uses the Firth logistic regression test to account for case-control imbalance. The two methods are similarly effective in terms of false positive rates and statistical power.”

Similarly, the paper does not seem to explicitly state whether linear or logistic regression was used to validate the polygenic scores (“indices”). Here the case for using logistic regression seems even stronger, because the scores and covariates together might have a rather nontrivial (and hence nonlinear) association with the outcome. Furthermore, the pseudo- R^2 from logistic regression may be useful for downstream analysis by others who want to know the power of the scores or adjust the scores for their reliability. The authors report the R^2 and the “additive R^2 ” from their validation models. We do not know the term additive R^2 and presume the authors mean the incremental R^2 . The term additive R^2 should be defined, or more conventional nomenclature should be used. We now report results from logistic regressions in the main text and provide pseudo \$R^2\$. We deleted the unclear term additive \$R^2\$. The logistic results are the same as the linear results in terms of the number of PGIs that are significantly associated with field qualifications in the independent LifeLines cohort.

The reason why we focus on linear results previously is that the approach of imputing and controlling for parental genotypes using *snipar* (Young et al. 2022 Nature Genetics) has only been validated with a linear approach to date. However this is likely to be more of a theoretical than a practical point.

The Methods section now states:

“In Dutch Lifelines, we tested whether each of the 10 PGIs was predictive for its respective educational field using genetic data at the population level, using logistic regression.

To quantify the variance in the field explained by each score, we report the incremental pseudo-R², which was quantified as the difference of the pseudo-R² in the regression with the controls and the PGI, and the pseudo-R² in a regression with only the control variables. Here, we used McFadden’s pseudo-R², which is the default in STATA, and is defined as $1 - L_1/L_0$, with L_1 the likelihood of the model, and L_0 the likelihood of the model with only a fitted intercept.”

The Results now state: “On the population level, 8 out of 10 PGIs are associated with their respective educational field at $p < 0.005$ ($N = 36,501$; see Figure 2 and Supplementary Table 8), though effect sizes were small to negligible. The largest associations were for *Arts and humanities* (change in log-odds = 0.22, $SE = 0.03$, $R^2 = 0.00529$) and *Natural sciences, mathematics and statistics* (change in log-odds = 0.17, $SE = 0.04$, $R^2 = 0.00283$).”

At what level of education does field of choice correspond to? Is this only for participants who have completed undergraduate study in university or higher, or is field choice at high school also considered? To discover this requires going to supplementary materials. Perhaps this should be stated more clearly in the main text.

We state in the Results “We harmonised population-wide administrative data from Norwegian and Finnish education registers on adults’ aged >25 to capture 10 broad fields of education as defined by the International Standard Classification of Education (ISCED) (<https://uis.unesco.org/sites/default/files/documents/international-standard-classification-of-education-fields-of-education-and-training-2013-detailed-field-descriptions-2015-en.pdf>). **We extracted data on individuals’ highest completed qualification by the year 2018, including qualifications at all levels (i.e., not just university level).**” More detailed information on phenotype definitions per cohort is available in the Methods.

Education was treated as a continuous covariate in the GWAS. Perhaps it should be treated as categorical. If either the independent or dependent variable has a strongly nonlinear relationship with a discrete covariate, treating the latter as categorical can lead to more effective statistical control. We point the reviewer to their below suggestion in the section “Correlation between field choice and cognitive performance/educational attainment”. We decided not to run analyses treating EA as discrete. However we conducted major new analyses to address their concern about the effectiveness of our approach to control for EA, by performing GWAS-by-subtraction in Genomic SEM (as Reviewer 2 also suggested).

Could Figure 1 present SNP heritability for the traits with and without the control for education? The models controlling for education are perhaps more pivotal to the paper, since the authors use these results for Genomic SEM and the authors are concerned with overcoming the “conventional hierarchy of attainment” by controlling for education. So these results should also be presented upfront. We find plausible the explanation by the authors in a supplementary figure that education might be a collider, but since the results controlling for education are often what are used in the paper downstream, it seems to make sense to include them in Figure 1.

We agree that it is important to control for educational attainment as effectively as possible, and that doing so bolsters the argument that our study captures ‘horizontal’ rather than ‘vertical’ educational stratification. Figure 1 now includes SNP heritability estimates with and without the control for education. The latter estimates are based on GWAS-by-subtraction.

Latent traits

We think that doing a GWAS of the PCs rather than the latent factors is more appropriate, for a number of reasons. In order to avoid some theoretical problems with the factor model, a factor must be susceptible to being measured by an infinite number of indicators, which in the limit take the reliability of measurement as close to one as we please (McDonald, 2003). This is an idealization of course, but it is met with reasonable verisimilitude if the number of potential items that can be written to measure a trait is very large. The fields of study analyzed in this paper seem to us closer to the entire universe of possibilities in this domain rather than a sample of an infinite (i.e., very large)

indicator pool, rendering factor analysis inapplicable.

The interpretation of PCA is application specific (e.g., McVean, 2009) or indeed can be completely lacking. One can treat PCA as dimension reduction, full stop. Rather than presenting genetic correlations with 10 different correlated traits, one can present genetic correlations with a smaller number of PCs, presenting a simplified and more digestible summary to readers.

Inspired by the PCA, the authors sought a factor model similar to it. The final factor model, however, is rather unsatisfactory because of its complex cross-loadings. These seem to be necessary to obtain an acceptable fit. In a PCA there is no conception of fit and hence no need to worry about tweaking a model. It would have been better to just stop with the PCA.

Another reason to prefer performing a GWAS of the PCs rather than F1 and F2 is that the effective sample size of the Genomic SEM is very low (~10K). Although this figure is somewhat arbitrary since it really depends upon the presumed heritability of the latent factor, it does imply that there is little power to perform a GWAS of the latent factors. By contrast, with a GWAS of the PCs, the analysis may have more power.

We thank the reviewers for raising the important question of the appropriateness of PCA versus CFA. We reflected on this and were convinced by the reviewers' arguments favouring PCA-GWAS. We reanalysed the data using PCA GWAS.

Our Methods section now explains why we chose PCA and how this differs from CFA:

"We explored the dimensionality of the genetic influences on the 10 educational fields by applying principal component analysis (PCA) to the matrix of genetic correlations between fields (after GWAS-by-subtraction to remove EA genetic variance) and then observing variance explained and loadings, and plotting the coordinates of PCs 1 and 2. To obtain GWAS summary statistics for PCs 1 and 2, we then performed GWAS following (Furtjes 2023 Human Brain Mapp)

<https://annafurtjes.github.io/genomicPCA/index.html>. This approach adapts a GWAMA function designed for meta-analysis across multiple traits. Instead of weighting by SNP heritability when averaging across SNP effects, the standardised loadings on the principal component of interest provide the weights. The function allows for sample overlap in the GWAS summary statistics by adjusting for the LD score regression intercept.

We chose to use PCA rather than confirmatory factor analysis (CFA) for studying the dimensionality of educational fields for several reasons. First, PCA involves fewer assumptions than CFA. While CFA models latent factors assumed to represent real traits influencing field choices, PCA simply reduces the dimensionality of the data by finding axes that explain maximum variation. Additionally, CFA requires testing model specifications with different patterns of cross-loadings, whereas PCA does not involve model fit considerations. Second, the limited number of possible fields (10 indicators) makes PCA more suitable. CFA is technically designed for studying latent factors that can be measured by an extensive, potentially infinite, number of indicators, which is not the case here.

"

We also adjusted the text throughout to ensure that we were not reifying the principal components as real traits.

The change from factor analysis to PCA did not affect the downstream results of the study. For example, the factors and PCs have almost identical genetic correlations with personality traits. PC1 was negatively genetically correlated with Extraversion and Agreeableness ($r_g = -0.42$ and -0.37), whilst estimates for F1 were -0.42 and -0.39 , respectively.

The authors suggest that the appropriate number of PCs to retain is two, writing:

__To establish the number of components explaining patterns of genetic correlations between fields, we performed a principal components analysis (PCA). Two PCs were satisfactory, given that they collectively explained 69% of the variance, and that the individual fields GWAS showed strong correlations with the top two PCs but not the third (see Supplementary Table 13 for PCA results).__

Whilst the choice of a number of latent factors to extract can be quite subjective, the given

explanation for two factors seems insufficiently substantiated. That two PCs explain a lot (69%) of the variance does not mean that a third PC does not also capture much important variation.

We agree and have removed the aforementioned text. We now simply describe results for PCs 1 and 2 without claiming that two dimensions are sufficient to explain variation in the genetics of field choices.

Information on how to perform GWAS of the PCs can be found in this link: <https://annafurtjes.github.io/genomicPCA/index.html>

We followed this pipeline—thanks to the reviewer for sharing this.

We know that the issue that we are raising here about factor vs. principal component analysis will seem esoteric and even trivial. We realize that the authors will probably resist our suggestion; for this reason we do not insist that they accept our admittedly minority view, since it probably will not make too much of a difference to the message of their paper. But the meaning of factor analysis has long been dogged by confusion and controversy, and we think that the conception of factor analysis that we have cited here stands up to all challenges.

The developers of Genomic SEM seem to take the view that a factor model can be applied to an inherently limited domain of indicators if a common-pathway model of SNPs acting through the factors tends to show excellent fit (Clapp Sullivan et al., 2024). If the authors want to adopt this view, they should report the Q_SNP statistics calculated by Genomic SEM.

Whilst this is an interesting and useful approach, we chose to explore Genomic PCA. The common pathway model is less applicable to our research question because we have two factors and indicators with complex cross loadings.

In ST14 only the AIC is given for the different models. Other informative fit statistics are given by GenomicSEM, e.g. SRMR. It would be good to give all the fit statistics available. If other statistics suggest the final model is a poor fit, then please explain this in the text of the paper.

Since we now focus on the PCA results and avoid emphasising model assumptions and fit statistics, we decided not to expand the CFA results.

Creative vs. conformist

Although labelling components does involve a degree of subjective judgement, we wonder if a better label can be created for the creative versus conformist scale; in the PCA, this is PC2. Firstly, conformist is not clearly the opposite of creative. Indeed, the authors seem to agree that creative and conformism are not opposite ends of a scale, saying:

__Indeed, whilst creativity and conformity have been thought to be diametrically opposed, recent work has highlighted that more conforming individuals can enhance the creative process.__

More generally, are the terms apt descriptions? Are people interested in health and welfare really conformist? We are not sure. An interest in _practical_, hands-on work is what farming, health, engineering/construction, education and services all have in common. We think the conformist description can come across as slightly pejorative, which is acceptable if we are sure the label is accurate. Likewise, business, administration and law do not seem especially creative to us.

We thank the reviewers for sharing their interpretations of PC2. We agree that creativity and conformity are not opposite ends of a scale. Whilst we chose 'Conformist' to capture an orientation towards secure public sector roles, this is too easily interpreted as pejorative. We have changed the PC2 label to 'Practical versus Abstract'. This has the advantage of focusing on labelling the activities rather than the person, and being a more obvious distinction between hands-on versus conceptual work.

Correlation between field choice and cognitive performance/educational attainment

Even after controlling for education, variation in field choice is still strongly correlated with cognitive ability and EA. This is because phenotypic education is only a noisy proxy for genetic variation in EA and intelligence. However, the authors state they want to shift from studying the "conventional hierarchy of attainments towards multidimensional tendencies and interests." We certainly think studying interests and variation in cognitive abilities, orthogonal to the general factor of intelligence, is

important. However, the GWAS does not entirely succeed in this goal, in light of the genetic correlations mentioned above.

We have already mentioned one possible attempt at a fix, which is to treat educational level as a categorical rather than continuous covariate in the GWAS.

An interesting analysis would be to use Genomic SEM to implement a model removing EA or CP variation from the field choices, rendering the latter true residuals. Then recalculate the genetic correlations. This method might be more interesting than controlling for phenotypic education in the GWAS, but it would be understandable if the authors left this for others to attempt. We conducted GWAS-by-subtraction, which we agree is a more sophisticated approach to controlling for EA that allows us to use the full power of the latest GWAS of EA. As shown in Supplementary Table 6, this approach led to similar conclusions as the phenotypic adjustment method: SNP heritability estimates for all 10 fields remained significant. However, the GWAS-by-subtraction based estimates were slightly lower for some fields, suggesting that this method was more effective at controlling for EA.

The genetic correlations between PCs 1 and 2 and EA are now -0.006 and -0.21, respectively. In the Discussion, we speculate that the residual genetic correlation with EA could arise due to parental indirect genetic effects. It is also possible that EA genetic variance specific to the Finnish and Norwegian datasets has not been successfully adjusted for with the external international GWAS.

Minor points

We now make some comments tied to specific bits of the paper.

title: We think it is somewhat misleading to suggest a sample size greater than 460,000.

We agreed and removed the 460,000 number from the title. We also added the sum of effective sample sizes for each field to the abstract and the first paragraph of the Results.

p. 3: "An accurate, holistic, and hypothesis-free approach to the dimensionality of educational fields would be to factor analyse" Some authorities have deprecated exploratory ("hypothesis-free") factor analysis. Instead of "factor analyse," perhaps say "apply dimension-reducing multivariate techniques to." And perhaps remove the word "accurate" because there is some judgment required in determining which eigenvalues are "large."

We changed this to "A holistic data-driven approach to understand the structure of educational fields would be to apply dimension-reducing multivariate techniques to actual field choices."

p. 5: "statistically indistinguishable from 1 apart from for" There is a typo here.

We corrected this.

p. 8: What is an FDR-corrected p-value? FDR and p-value have almost exactly opposite meanings. FDR is the proportion of rejected null hypotheses that are in fact true, whereas p-value is the proportion of true null hypotheses that are rejected. That is, FDR is $P(\text{null is true}|\text{rejection})$ and p-value is $P(\text{rejection}|\text{null is true})$. Some software tools call FDR a method for adjusting p-values, but this is misleading. Please use appropriate terminology. Perhaps the simplest solution is to determine the p-value corresponding to an of FDR of 0.05 and call everything below that threshold significant. Also, the set of hypotheses (all traits? just one trait?) to which the FDR applies needs to be specified with precision.

Thank you for identifying this issue. Our Methods now reads:

"To study how much the genetic associations with broad field choices are mediated through unobserved social factors in the geographical area individuals were born in, we repeat the GWAS analyses in MoBa only, controlling for municipality codes and parental field codes as dummy variables and test for attenuated heritability with LD Score regression. We did this in an iterative fashion, first controlling for the most distal factor (birthplace municipality) then adding parental fields. Following previous methods applied in UK Biobank ⁴⁰, we used Genomic SEM to compare the SNP-heritability estimates with and without controls whilst accounting for dependence between estimates. We calculated p-values corresponding to a FDR of 0.05 using the `p.adjust` function in R with the method "fdr"."

p. 10: "Engineering, manufacturing, and construction ..." How the fields are designated, with capital letters or not, is confusing. Is there a better way to do this? Italics, quote marks?
We have followed the formatting given in the official international classification system (ISCED). To improve readability, we now put the fields in italics.

Figure 5: What FDR threshold is used for statistical significance? The standard 0.05? This needs to be stated somewhere.
We now state that we used the threshold 0.05.

p. 15: "We found that SNP heritability estimates for educational field choices were 7% on average. These are likely to be underestimated because our methodology only captures additive effects of common variants tagged by genotyping arrays, rather than the full heritability (the missing heritability problem)" The wording here seems unclear. SNP heritability is an estimate of the variance explained by the common variants, so it is not necessarily an underestimate of what it is trying to measure.
Thanks for this point. We have changed the sentence as follows: "We found that SNP heritability estimates for educational field choices were 7% on average. These are likely to be lower bound estimates of the total role of genetic factors because our methodology only captures additive effects of common variants tagged by genotyping arrays, rather than the full broad-sense heritability (the missing heritability problem)."

p. 15: "educational field choices may be smaller than and show little overlap with parental genetic effects on educational attainment" What does this mean? This clause makes no sense.
Thank you for this good catch. We have added the missing text as follows: "parental genetic effects on educational field choices may be smaller than parental genetic effects on educational attainment."

p. 17: "Individuals with more engineering-associated DNA variants 'should' not necessarily choose engineering and are not qualitatively different to people with more art-associated DNA variants." This sentence veers away from the factual and into the normative. (It is also hard to see why the word `_should_` is in quote marks.) Robert Plomin has argued that individuals with polygenic scores indicating promise for a certain pursuit should be encouraged to take up that pursuit. This is the opposite of what the authors are saying here. The conclusion to be drawn from these contrary statements is that they are not scientific and thus do not belong in scientific papers. Moreover, the second clause is just meaningless. Of course people are different; they are genetically different. What does it mean to say that this difference is not "qualitative?"
Thank you for highlighting this. The sentence did not communicate the intended point. We were trying to communicate that these results do not have any necessary implications for what people should choose to study, since this is a personal judgement. We removed this section from the Discussion and include the following edited paragraph in our FAQ document:
"Many of us would be keen to know any information that helps match us with an educational programme.

...
However, PGIs cannot capture important individual and contextual information on actual interests, skills, and opportunities. A person with a high polygenic index for Technical fields might not prefer to study a technical subject, have the option to pursue it, or be most successful or happy in studying it."

p. 22: "SBayesR uses Bayesian shrinkage to correct for linkage disequilibrium" This is incorrect. SBayesR and similar polygenic-score constructions methods use Bayesian shrinkage to deal with sampling error.
Thanks for this. We corrected it.

p. 24 and Supplementary Table 8: "cube in age" It is more standard to say "cubic in age."
We corrected it.

Supplementary Figure 17: "we also avoid making assumptions about causal direction of EA<>fields" Is this a typo? What does this mean?
We replaced this with "By not controlling for EA, we also avoid making assumptions about causal direction of effects between educational attainment and educational fields."

Supplementary Table 5: There is something wrong with this table. Is it supposed to say “controls” rather than “cases” in the lower part.

We cannot find anything wrong with the table. We are showing the number of individuals observed per educational field (cases) in the GWAS samples and in the populations from which the GWAS samples were drawn.

Supplementary Table 11: How were the SEs and p-values of the ratios calculated?

Please see the Methods section we pasted above. We followed the Abdellaoui et al. 2022 Nature Genetics article "Gene-environment correlations across geographic regions affect genome-wide association studies".

References

Clapp Sullivan, M.L. et al. (2024). Beyond the factor indeterminacy problem using genome-wide association data. *Nature Human Behaviour*, 8, 202-218.

McDonald, R.P. (2003). Behavior domains in theory and in practice. *Alberta Journal of Educational Research*, 49, 212-230. doi:10.11575/ajer.v49i3.54980

McVean, G. (2009). A genealogical interpretation of principal components analysis. *PLoS Genetics*, 5, e10000686. doi:10.1534/genetics.111.131540

Reviewer #2:

Remarks to the Author:

Summary. This paper reports a GWAS meta-analysis of educational field of study across three relatively large cohorts (FinnGen, MoBa, and Lifelines). The paper is novel in focusing on field of study, rather than educational attainment (total years of schooling), which has been the focus of much prior work. The paper identifies genome-wide significant SNPs, estimates SNP heritabilities, conducts polygenic prediction analyses, tests for robustness using within-family variation, identifies two main axes of genetic influence on field of study, and examines genetic correlations with a wide range of phenotypes.

Evaluation. This is a creative and intriguing paper. Educational field of study is an important phenotype to study, and the results are interesting, most notably the categorization into Technical vs. Social fields and Creative vs. Conformist fields.

The paper has two main limitations. First, the sample sizes are small for some of the fields (e.g., according to Supplementary Table 5, the effective sample size for the field “Natural sciences, mathematics and statistics” is only ~40,000 individuals). This raises the concern that some of the GWAS may be underpowered. While the small sample size per se is not a concern for the SNP heritability and genetic correlation analyses (though contributes to wide confidence intervals), it implies that the specific SNPs identified by the GWAS are more likely to be false positives. Second, it is unclear how much of the genetic signal goes beyond the genetic signal for educational attainment. Although the paper controls for educational attainment in many of the analyses, the control is likely incomplete, as discussed further below.

Suggestions. My main suggestions correspond to the two main limitations listed above. First, while the sample size is what it is (unless the authors can get access to additional cohorts to include in the GWAS meta-analysis), the paper should be clearer about the sample sizes in the GWASs and the fact that the relatively small sample size is a limitation. The title and abstract report a total sample size for FinnGen and MoBa of >460,000 individuals, but I could not find effective sample sizes for the GWASs until I dug into the Supplementary Tables and found Supplementary Table 5. This information should be in the main text. The reporting of the number of SNPs identified in each GWAS should be accompanied by a caveat about the low power for several of the fields.

We thank the reviewer for the positive comments.

We agree that the effective sample sizes should be reported more prominently.

In the Abstract, we now report the sum of effective sample sizes: “total sample size=463,134; sum of effective sample sizes ranging from 40,072 for Natural sciences, mathematics and statistics to 317,209 for Engineering, manufacturing and construction”.

The first paragraph of the Results now says “The sum of effective sample sizes was 317209 for Engineering, manufacturing and construction, 292929 for Health and welfare, 261182 for Business, administration and law, 168157 for Services, 102970 for Education, 97262 for Arts and humanities, 69123 for Social sciences, journalism and information, 63834 for Agriculture, forestry, fisheries and veterinary, 50819 for Information and Communication Technologies (ICTs), and 40072 for Natural sciences, mathematics and statistics.”

The Results also mentions: “Note that SNP associations identified for the fields with lower sample sizes (such as Natural sciences, mathematics and statistics) are more likely to be false positives.”

The legend for Figure 1 (SNP-based heritability estimates) also contains the sum of effective sample sizes.

Second, the paper should use a more sophisticated approach to examining genetic influences on fields of study that go beyond genetic influences on educational attainment. Consider the GWAS on fields of study that controls for educational attainment. Both because educational attainment is influenced by non-genetic factors and because educational attainment is a coarse measure (it is discrete with just a few categories, whereas the genetic factor for educational attainment is essentially continuous), controlling for educational attainment in the GWAS is an incomplete control for genetic influences on educational attainment. A similar concern arises in the GWAS that controls for parental educational fields, since that is an imperfect measure of parents' genetic propensity for educational fields.

One approach to control for the genetic influences on educational attainment would be the GWAS-by-subtraction method of Demange et al. (2021, Nature Genetics). The identifying assumption — which may not be plausible — would be that while genetic influences on educational attainment affect field of study, genetic influences on field of study do not affect educational attainment. Another approach would be to control for the polygenic index for educational attainment, along with an errors-in-variables bias correction (as in DiPrete et al., 2018, PNAS, or Becker et al., 2021, Nature Human Behaviour).

We conducted GWAS-by-subtraction, which we agree is a more sophisticated approach to controlling for EA that allows us to use the full power of the EA4 GWAS. As shown in Supplementary Table 6, this approach led to similar conclusions as the phenotypic adjustment method: SNP heritability estimates for all 10 fields remained significant. However, the GWAS-by-subtraction based estimates were slightly lower for some fields, suggesting that this method was more effective at controlling for EA.

We also developed our discussion of the causal relationships between educational fields and level (Supplementary Note on “Causal relationships between educational fields and educational attainment”). We include a path diagram and an explanation of the Genomic SEM model approach and assumptions :

“The GWAS-by-subtraction model for the example of social science, with path estimates for a single SNP, is shown in the Supplementary Figure above. SNP, and educational attainment and Social science are observed variables based on GWAS summary statistics. The genetic covariance between these is estimated based on their GWAS summary statistics. The model is fitted to a 3×3 observed variance–covariance matrix (that is, SNP, educational attainment, social science). EA and SOC are latent (unobserved) variables. The covariances between the educational attainment and social science variables are fixed to 0. The variance of the SNP is fixed to the value of $2pq$ (p = reference allele frequency, q = alternative allele frequency, based on 1000 Genomes Project phase 3). The residual variances of EA and SOC are fixed to 0, so that all variance is explained by the latent factors. The variances of the latent factors are fixed to 1. The observed variables were regressed on the latent variables, resulting in the estimates for the path loadings: $\lambda_{\text{Edu-EA}}=0.37$; $\lambda_{\text{SocSci-EA}}= 0.28$; $\lambda_{\text{SocSci-SOC}}=0.18$. Though not fully realistic, we assume that EA causes field choice and that there is no path from the latent SOC factor to observed educational attainment. See Supplementary Table 7

for path loadings for the other fields. The latent variables were then regressed on each SNP that met quality control criteria.

We also compare the GWAS-by-subtraction approach to the approach of simply controlling for EA phenotypically. The advantage of the phenotypic approach is that we control for EA in the specific Norwegian and Finnish cohorts, whereas the EA-GWAS might not be fully overlapping in genetic influences. However, because educational attainment is influenced by non-genetic factors and because educational attainment is a coarse measure (it is discrete with just a few categories, whereas the genetic factor for educational attainment is essentially continuous), controlling for educational attainment using Genomic SEM and the >1m person EA-GWAS offers a more comprehensive control.”

Here are some other specific suggestions that I hope will be useful in improving the paper:

1. The paper should mention the issue that the error terms across the GWASs are correlated because (I think) the educational fields are mutually exclusive. Thus, if a SNP is by chance positively associated with being in one field, then it is mechanically negatively associated with being in other fields. This is therefore a case of seemingly unrelated regressions (SUR; Zellner, 1962, Journal of the American Statistical Association). I believe this is not a problem for any of the analyses in the paper because SUR estimation gives the same coefficients as OLS when the regressors in each equation are the same (Amemiya, 1985, Advanced Econometrics), but this issue deserves a mention.

Thank you for making us aware of the issue. We have mentioned this in the Limitations section of the Discussion “we note that if a SNP is by chance positively associated with being in one field, then it is mechanically negatively associated with being in other fields. Future work should investigate how this issue of Seemingly Unrelated Regression (SUR) ⁷³ affects the genetic structure of fields, for instance through multinomial regression”.

2. Lines 220-222: “However, power for within-family analyses in the subsample was low, such that only PGI associations for Arts, humanities and languages and Engineering, manufacturing and construction remained significant.” – The paper should report confidence intervals for the decreases. Thank you for the good catch. We now report the confidence intervals for the decreases in Supplementary Table 9. We found that none of the decreases were significant.

3. Lines 269-273: Why use PCA rather than (orthogonal) factor analysis, which is more standard for

identifying latent factors?

Please see the response to Reviewer 1 where we provide discussion on the appropriateness of PCA versus FA. In short, the PCA approach allows us to avoid making unrealistic assumptions and decisions about how to handle complex cross-loadings.

4. Lines 383-384: “such that low genetic correlations capture different influences on different entities rather than on the same entity” – What does this mean?

We hope the following revised wording is clearer: “there are sex differences in qualifications within broad fields (e.g., within Engineering, manufacturing and construction, males are more likely than females to study construction). A low genetic correlation for Engineering, manufacturing and construction between males and females could thus reflect that different heritable traits are involved in choosing construction over engineering, rather than anything about sex differences per se.”

5. Lines 403-404: “[SNP heritabilities] are likely to be underestimated because our methodology only captures additive effects of common variants tagged by genotyping arrays, rather than the full heritability” – While it is true that broad heritability is likely underestimated, this sentence does not make sense regarding SNP heritability. (LDSC may underestimate SNP heritability but for different reasons than mentioned in this sentence.)

Thanks for this point. We have changed the sentence as follows: “We found that SNP heritability estimates for educational field choices were 7% on average. These are lower bound estimates of the role of genetic factors because our methodology only captures additive effects of common variants tagged by genotyping arrays, rather than the full broad-sense heritability (the missing heritability problem).”

6. Lines 412-414: “Nonetheless, our low SNP heritability estimates leave a large role for environmental factors and random chance in educational field choices.” – As the paper acknowledges elsewhere, genetic influences can operate through environmental factors, but the sentence as stated incorrectly implies that a large heritability would rule out a large role for environmental factors. Also, what is “random chance” aside from exogenous environmental factors?

Thank you for this good catch. We have changed the paragraph as follows: “Our results suggest that parental genetic effects on educational field choices may be smaller than parental genetic effects on educational attainment. Nonetheless, parental environmental factors are likely to be key mediators of direct genetic effects.”

Signed,
Dan Benjamin

Reviewer #3:
Remarks to the Author:

Having now read through this paper several times and thought carefully about it, the first thing to say is that the paper is excellent. There's no doubt that the datasets - the uniqueness of the data structure, the sample size, and the cross-country nature when speaking about educational and training qualifications, is perfectly suited for a decomposition at the phenotypic and genetic level of realms/factors of qualifications. The methods are excellent - when I first opened the paper, I thought I might find myself making several comments about the appropriateness of the methods as I usually do as a reviewer. Instead, the methods read almost like a handbook of how I would go about doing such a study if I had access to this data. So instead - I find most of my comments directed toward the social science going on here:

1) First, the paper suffers from a lack of integration of deepness and richness of the sociological exploration education and occupation that has come over the last several hundred years. I understand that this is a Nature Genetics paper, but some of its conclusions will be most meaningful and impactful in the social sciences. I see sociologists and a sociology department listed in the affiliations, but this paper does not really read like it's trying to speak to the fields that will indeed be most curious or excited about they work. And so I would challenge the authors, with of course the editor's permission, to add a section that deepens the articulation of the literature and just why this topic matters so much for life outcomes. Authors - this is call to action to really dig into the work and

previous literature - so often when these sorts of studies are published (I'm thinking about all of the iterations of the genomics of educational attainment work), I think we're left with a very straightforward technical manuscript about an outcome that is deeply important to society, but not understood by geneticists at all. So you need to prime the geneticists to tell them why this is important so that this paper doesn't open itself up to the typical criticism, and you need to take good care of the sociologists who have come to think of geneticists studying these outcomes as something of an affront. This can absolutely be done, but it will take some careful work.

We are grateful to the reviewer for commending our work and encouraging us to clarify its significance and context. We have seized this opportunity to communicate to social scientists how genetic approaches can bring relevant novel findings, and to communicate to geneticists about relevant aspects of the social context that are already deeply understood.

We heavily edited the article throughout to bring the social science context to the foreground.

Please see the text below for key examples of changes to our Introduction. We made a particular effort to add an introductory section priming the audience on highly influential research on horizontal stratification in Sociology, Economics, and Education sciences:

“Education is fundamental to the economies, cultures and social stratification systems of modern societies ¹⁻³. As a result, it is one of the most frequently studied aspects of human behaviour by social scientists. Extensive research has linked education to myriad outcomes across multiple domains, including occupational and labour market outcomes ⁴, culture ⁵, and health ⁶.

Fields of education, ranging from fine arts to finance, are extremely diverse. They involve varying degrees of cultural, economic, technical, and communicative skills ⁷. Choosing a field of education sets the course for one's occupational opportunities. Even when controlling for educational level, economic returns differ greatly across fields⁸, with engineering, business, science, or mathematics students usually earning more than those with humanities, education, and social science qualifications ⁸. Field specialisations do not only have economic effects, but also impact on attitudes, leisure activities, and fertility ^{9,10}. Moreover, field of study effects can span generations, by determining social networks and marriage markets ¹¹. ‘Vertical’ stratification in educational attainment (defined simply as the number of years a person spends in education; EA), tells only part of the story of how education affects our lives. It is increasingly important to understand patterns of ‘horizontal’ stratification of educational outcomes (types or quality of education).

The mass expansion of educational systems is one of the most important social transformations of the twentieth century, democratising access to educational opportunities, increasing the supply of skilled labour, and contributing to increased social mobility ¹². The emergence of mass education in industrialized countries makes horizontal stratification relatively more salient ¹³. The importance of one's field of study has increased as the signalling value of educational level has weakened ⁸.”

We briefly review the sociological literature so that geneticists are aware of the structural forces at play for these outcomes.

“Educational choices are structured along multiple social dimensions. Educational choices, and consequently also occupational choice, are highly gender-typed ¹⁴. Women are overrepresented in caring fields such as nursing and social work, while men are overrepresented in technical fields such as engineering and finance ¹⁵. Generally, male-dominated occupations tend to come with higher wages ¹⁶⁻¹⁸. These patterns are likely suggested to be maintained due to socialization, priming, status, gendered norms and cultural stereotypes ¹⁹⁻²¹. Parents' vertical and horizontal educational positions are correlated with offspring field choices ^{22,23}, with students from more educated backgrounds choosing more financially risky careers ²⁴. Geographical factors such as urban-rural disparities also affect choice norms and access to certain fields ²⁵.”

From there, we take the reader through the topics of personality and vocational interests. In doing so, social science readers are eased gradually into the understanding that genetic associations with fields occur because people are different.

We also raise the gaps left by social science research on fields of study.

“Psychology research shows that educational choices are correlated with individuals’ vocational interests and systematic tendencies for behaviour 28,29. For example, more extraverted people sort into fields which provide opportunities for social contact

...

Efforts have been made to establish the overarching mechanisms of choice by measuring individuals’ preferences (e.g., for fields with intrinsic versus extrinsic rewards, or for entrepreneurial versus bureaucratic characteristics 31). However, preference measures are usually only available in small sample sizes, each explaining a small fraction of the choice process, and studies rarely include data on individuals’ actual choices.

...

Genetic associations with various fields of study are unlikely to be completely accounted for by the known genetic correlates of socioeconomic status, and thus may lead to novel insights on how individual and contextual factors combine to influence life chances.

”

2) In the same vein, I finished the paper's discussion and was left with a bit of a "so what" reaction. I felt like the authors crept to the precipice of giving us some important concluding thoughts on their ability to parse outcomes into technical vs. creative domains or to interpret correlations with health outcomes. But the authors didn't move much beyond this. They posit toward potentially meaningful analyses - for example: "GWAS summary statistics for the two factors and for the individual educational fields could be used to further understand the mechanisms involved in, and consequences of, field choices. For example, PGIs could be used to trace how genetic predispositions manifest in young people's early interests, skills and choices, and how social contexts magnify or suppress certain predispositions (gene-environment interaction questions)." Why not actually do some of these analyses to move us toward meaning a bit more? In Carey et al. published earlier this year in Nature Human Behavior, though they are brief, in the section of the paper dealing with SES factors, the authors point us back toward the importance of these findings for the fields of social science dealing with socioeconomic status and our ability to measure and understand it. These are the sorts of conclusions drawn from psychology and sociology that should be demonstrated - perhaps through some of the additional analyses the authors suggest, but also with a concentrated effort to tell us why these findings matter. What exactly is it we learn here with this unique era of genetic and phenotypic data that moves the social sciences forward in a way that we couldn't get to before this type of data was available? I find myself seeing social science revolutionary work here, but as it is currently presented, a standard reader of NG won't see the innovation here.

Thanks to the reviewer, we worked hard to better articulate how our findings contribute to social science. The Carey et al. article provided a great deal of inspiration here as the articles are somewhat similar in revealing previously unknown and unknowable social structures through big data research.

The importance of our findings are stated more clearly throughout our Discussion. Please see below for some of the new text:

“Genetic analyses of field specialisations revealed relationships that are difficult to study on the phenotypic level, such as the strong overlap between Social sciences, journalism and information and Arts and humanities. Through dimension reduction of the genetic correlations, we provide novel evidence on patterns of sorting into fields.

...

The PCs correspond well with major theories in the social sciences. The RIASEC model of vocational interests, widely used by careers advisers ⁵⁴, includes Social and Realistic interests (like PC1) and Investigative/Artistic and Realistic/Conventional interests (like PC2). The PCs also match sociological theory delineating the major educational resources that individuals invest in (Communicative, Technical, Cultural, and Economic) ^{9,55}. **This striking alignment between social science theory and hypothesis-free analyses of big genetic data triangulates the validity of this theory from a novel perspective. More generally, identifying patterns of social sorting without the need for theory or direct measurement of preferences demonstrates how genetic approaches may complement social-scientific enquiry (for corresponding results on social and health inequalities, see ⁵⁶).**

In showing how Technical-Social and Practical-Abstract qualifications correlate on the genetic level with 96 human phenotypes, **we expand the scope of social science research on educational fields**. Although studies have investigated the causes and consequences of interests and qualifications, such as personality ⁶², earnings ¹³ and fertility ⁶³, these studies have been limited by the difficulty of measuring phenome-wide outcomes at scale in the same sample. We **incorporate new domains such as mental health, substance use, relationship satisfaction and body size**.

...

Capturing genome-wide associations with interests has not been possible due to insufficiently powered genotyped samples. We provide a novel GWAS on a proxy of interest through educational field choice.

...

Interestingly, our genetic results paint a **more nuanced picture of social and economic variation in the population than is possible when only looking at conventional status markers**. It appears we may also be identifying some disadvantages of elite educational paths. For example, propensity to study Abstract rather than Practical educational fields is related to socioeconomic instability such as more loneliness and divorce, lower relationship satisfaction, lower vitamin D levels, and higher risks of psychiatric disorders.

Overall, these findings offer a new direction for genetically informed research on vocational interests and horizontal social stratification.”

We believe that the numerous analyses contained in the study already provide plenty of novel findings for the social sciences and open up multiple new pathways for research.

To further highlight how our genetic results tell us about horizontal social stratification, we added an analysis to explore the role of educational fields in mate choice: “Further, we explored assortative mating involving educational fields. We tested whether the same PGIs predicted the educational field of one’s spouse (N=28,581). For Education, Arts and humanities, and Services, the PGI is associated with the educational field of one’s spouse at $p < 0.005$ (Supplementary Table 13).” This shows how important educational fields are for partnership, which is as a key driver of social stratification.

3) A frequently Asked Questions (FAQ) is necessary, no doubt. But I implore the authors to do something new and innovative here. Why not create and FAQ website where sum stats are made available, but also where an interactive FAQ is available. The big paper on same-sex sexual behavior had such a website - this seems like a great idea here - <https://geneticsexbehavior.info> . I'd push the authors to make this something really special and engaging that can draw readers in.

We thank the reviewer for this great suggestion. We created an FAQ with the following headings: What was the motivation for the study? What did we do? What did we find? Who did we study and why does that matter? Why are genetic factors associated with fields of education? Are there any practical uses of genetic results for individuals and policymakers?

Using inspiration from the same-sex sexual behaviour FAQ, we published it on a new GitHub page with neat drop-down headings [https://github.com/rosacheesman/Fields_genetics/wiki/Frequently-Asked-Questions-\(FAQ\)](https://github.com/rosacheesman/Fields_genetics/wiki/Frequently-Asked-Questions-(FAQ)). Full code is also available on Github.

To expand the reach of the FAQ, we have sought out other publication avenues. We additionally submitted the FAQ to the editorial team of the Hastings Centre’s “FAQs on Human Genomic Studies” <https://www.thehastingscenter.org/faqs-on-human-genomic-studies-submission-guidelines/>. We also include the FAQ directly within the paper’s Supplementary Information file.

We have built a page to host summary statistics on Zenodo (built and operated by CERN and OpenAIRE) to be released upon publication of the article.

In summary, this paper is important and compelling, but unless the authors can get this to speak to

the correct audiences in a more convincing and rich way, I worry that it will be lost amongst other papers published on topics like these and sort of just "added to the pile."

Reviewers' Comments:

Reviewer #1 (Remarks to the Author):

The two reviewers submitting this referee report are a Ph.D. student and his supervisor.

The paper has been improved for clarity in many areas. The chief methodological changes are the use GWAS-by-Subtraction and genomic PCA. Both of these alterations represent improvements in focusing in on field choice within levels of educational aptitudes. There are a few areas, discussed below, where greater changes could have been made in response to our suggestions. However, these are small issues we are happy to overlook. There are some major issues stopping us from supporting publication.

We have organized our suggestions under a series of headings. Only the major suggestions are mandatory for our approval. The minor ones are up to the discretion of the authors.

We thank the reviewers for reading the revised article so thoroughly and providing insightful comments. We have incorporated the feedback to the best of our abilities and believe that the article has again substantially improved thanks to the reviewers' input.

MAJOR ISSUES

PC EA correlations

__The genetic correlations between PCs 1 and 2 and EA are now -0.006 and -0.21, respectively.__

After GWAS-by-Subtraction, the correlations between PCs and EA should be exactly zero. Each PC is a linear combination of the field choices. If GWAS-by-Subtraction has been performed successfully, then these field choices should be uncorrelated with EA, and thus any linear combination of them will still be uncorrelated with EA.

A correlation of -0.006 can be explained by slight mathematical imprecision, -0.21 cannot be. There would seem to be an error here, unless EA is taken from a new validation sample.

To debug the issue, you could first check that GWAS-by-Subtraction has been performed successfully on each field choice. After subtraction, does each field choice have a 0 genetic correlation with EA? If it has been successful, something perhaps has gone wrong in creating the PCs.

If the GWAS-by-Subtraction failed to remove associations with EA, then the problem might be with Genomic SEM. In this case, we recommend using the GSUB R package. This performs GWAS-by-Subtraction using a closed-form solution rather than producing an SEM for each SNP. As such, it is much faster and much simpler, meaning that there are fewer things that can go wrong.

<https://github.com/qlu-lab/GSUB>

We thank the reviewers for encouraging us to further investigate the genetic correlations with EA. We conducted several additional GWA and genetic correlation analyses and had a discussion with the Genomic SEM developers. We found that there has not been an error with PCA-GWAS or with our specific study variables.

The GWAS-by-subtraction approach appears not to successfully ‘subtract’ all EA genetic variance for all educational fields. As the reviewers suggested, we checked the genetic correlations with EA before and after GWAS-by-subtraction. The degree of genetic correlation with EA following GWAS-by-subtraction varies across fields (average 0.11, minimum = 0.0009 for *Health*, maximum= 0.3 for *Social sciences*; see Supplementary Table 8). Note however that these large genetic correlations reflect that the genetic covariances are relatively large compared to the small genetic variances (e.g., the genetic covariance of SocialSci-minus-EA4 with EA4 is 0.02, while the variances are .04 and .14, respectively).

Following the reviewer’s suggestion, we also investigated whether the closed-form approach of GSUB would be more effective. However, the results were the same (e.g., the GSUB summary statistics for *Social sciences* had a genetic correlation with EA4 of 0.3).

This imperfect subtraction was already observed and highlighted in the original GWAS-by-subtraction case of non-cognitive and cognitive skills (Demange et al. 2021). Indeed, when extracting GWAS summary statistics for non-cognitive and cognitive skills factors and applying LD score regression—their genetic correlation is between 0.15 (Malanchini et al. 2024) and 0.30 (Demange et al. 2022).

The Genomic SEM developers advised us against interpreting the genetic correlations with EA because it is circular to use EA summary statistics in the model, export ‘subtracted’ summary statistics and then recalculate genetic correlations. We are unable to test genetic correlations between ‘subtracted’ educational fields and an independent GWAS of EA because we lack summary statistics that are independent of EA4 for the samples included in our model. Regardless, we do not think that the genetic correlations would be zero, because the non-cognitive skills genetic factor shows non-zero correlations with IQ even when completely independent IQ GWAS samples are used (Demange et al. 2022).

Overall, this evidence makes us reluctant to over-emphasise or over-interpret the genetic correlations between fields and EA in the article, but focus on the fact that factors for educational fields have more than one dimension, and none of them are collinear with EA.

Moving forward, we believe that we have thoroughly explored the topic of why and how to control for EA and have interpreted the adjusted results with adequate caution. We have implemented three statistical techniques to try to capture the uniquely field-specific variance (phenotypic adjustment, GWAS-by-subtraction, and GSUB). We emphasise results based on GWAS-by-subtraction when calculating PCs, since this is the most effective approach available for capturing subject interests over and above educational level.

As shown below, we have carried out major changes to the text and added supplementary content to discuss the details of whether and how to correct for EA, and to acknowledge that the correction is imperfect.

Results: “Genetic correlations between EA and fields, before and after GWAS-by-subtraction, indicate that the procedure was successful for most fields, yet did not completely eliminate EA variance for some fields (e.g., *Natural sciences* and *Social sciences* were genetically correlated at ~0.3 with EA (Supplementary Table 8).”

...

“Although the PC summary statistics were adjusted for educational attainment using GWAS-by-subtraction, PC2 remains significantly (negatively) genetically correlated with occupational status, educational attainment, and childhood IQ ($r_g = -0.26, -0.21, -0.29$)”

Discussion: “The Abstract-Practical component (PC2) is related to traditional ‘vertical’ socioeconomic indicators including occupational status. This may be partly because the GWAS-by-subtraction did not fully remove genetic overlap with educational attainment (EA) for all fields. However, the genetic correlation between PC2 and EA is small, and Abstract/Practical fields are not clearly patterned by EA (e.g., among Practical fields, ~40% of those with Services and Agriculture qualifications have an undergraduate degree, while degree attainment is over 76% for Health and welfare). These findings may therefore also reflect how social and economic resources hold greater importance in Abstract educational and career trajectories.”

Our **Supplementary Information** section contains a 2-page long discussion of possible causal relationships between educational fields and educational attainment, reasons for controlling for EA (or not), and advantages and limitations of different methods. Two example paragraphs are shown below:

“Notably, we found that the GWAS-by-subtraction did not fully remove genetic overlap with educational attainment (EA) for all fields, as indicated by non-zero genetic correlations between several residual field factors and EA, as well as between PC2 and EA (Supplementary Table 8). The degree of genetic correlation with EA varies across fields (average 0.11, minimum = 0.0009 for Health, maximum = 0.3 for Social sciences). It may be problematic and rather circular to use EA summary statistics in the model, export ‘subtracted’ summary statistics and recalculate genetic correlations. Multiple issues with estimating SNP-level effects such as sample overlap and population stratification can compound and be reintroduced when re-estimating genetic correlations. This makes it difficult to interpret genetic correlations between EA and fields after GWAS-by-subtraction. Even if we do take the genetic correlations with EA to be accurate, they are considerably attenuated after GWAS-by-subtraction, and the genetic correlation with PC2 of 0.21 is not so large as to imply that PC2 simply captures EA.”

“It is artificial to separate preference for a field and preference for a duration as the education system combines these two. When a person chooses a study programme, they are choosing a field and level. Medical doctors usually have long education, and so attainment is an intrinsic part of the choice. Selection into different combinations of levels and fields – such as nursing being a Tertiary-level degree within the health field – will vary across educational programs available in the system. They will also vary across countries and birth cohorts as educational systems and opportunities change over time. It may therefore be unrealistic that any given statistical procedure will remove all variance that is due to the level of qualification people have. Our separation of these is an abstraction that we impose on the “choice architecture” embedded in the educational system.”

References:

Demange, P.A., *et al.* Investigating the genetic architecture of noncognitive skills using GWAS-by-subtraction. *Nat Genet* 53, 35–44 (2021). <https://doi.org/10.1038/s41588-020-00754-2>

Malanchini, M *et al.* Genetic associations between non-cognitive skills and academic achievement over development. *Nat Hum Behav* 8, 2034–2046 (2024). <https://doi.org/10.1038/s41562-024-01967-9>

Incorrect Eigenvalues

In spreadsheet ST17, the eigenvalues do not correspond to the loadings. Calculating the sum of squared loadings for the first component, we obtain an eigenvalue of 1.86, but the value reported in the spreadsheet is 2.63.

The order of the components appears to be incorrect too. The components are not listed in order of descending eigenvalues. For example, component 3 has an eigenvalue of 1.38, but component 4 has an eigenvalue of 1.65.

We've looked back at the original submission and the eigenvalues were calculated correctly then and they were in descending order. So the error has been added to the paper.

We thank the reviewer for noticing this error. We have corrected the PCA results table (please see the subset of Supplementary Table 19 below).

Variable coordinates.

Field	PC1	PC2	PC3
Education	-0.7226861	0.01746635	-0.62286938
		-	
Arts and humanities	-0.5256941	0.49960861	-0.36648456
Social sciences, journalism and information	-0.6485770	0.62243905	-0.17482912
		-	
Business, administration and law	-0.1080261	0.68905917	0.43318173
Natural sciences, mathematics and statistics	0.6646720	0.19789913	-0.68887867
Information and Communication Technologies (ICTs)	0.8770234	0.35222824	-0.08090977
Engineering, manufacturing and construction	0.9534441	0.13672376	-0.07005359
Agriculture, forestry, fisheries and veterinary	0.2245368	0.73412636	-0.35649753
Health and welfare	-0.5997727	0.61774398	-0.33020359
Services	-0.3874606	0.54817228	0.59815884

Eigenvalues.

PC	eigenvalue	% of variance	cumulative %
1	3.91123548	39.1123548	39.11235
	5	5	
2	2.51511089	25.1511089	64.26346
	5	5	

MINOR ISSUES

“We then estimated SNP-based heritability of the residual GWAS summary statistics.”

We believe that this sentence added in revisions, and all related material, should be removed. The developers of Genomic SEM have explicitly recommended that summary statistics

produced by latent-variable modeling not be used as a basis for heritability estimates (e.g., Mallard et al., 2022). Environmental variance is not modeled in this approach and so therefore heritability (which includes environmental variance in the denominator) is undefined. The “heritability” estimated in this way can also depend on irrelevant choices such as the identification strategy (unit loading vs. unit variance).

It is possible that the authors took this approach after reading our comments in the first round. We are very sorry about this. We should have anticipated this problem. Perhaps the authors can retain their summary statistics with education as a covariate and, whenever necessary, use those to estimate heritabilities.

Mallard, T.T. et al. (2022). Multivariate GWAS of psychiatric disorders and their cardinal symptoms reveal two dimensions of cross-cutting genetic liabilities. *Cell Genomics*, 2, 100140. We thank the reviewers for pointing out this advice from the Genomic SEM authors. We indeed took this approach after multiple reviewers provided a strong rationale for using GWAS-by-subtraction to control for EA.

As the reviewers suggested, we have retained the results based on using EA as a covariate and we now report these results up-front in Figure 1. We considered completely removing the GWAS-by-subtraction results but were concerned that it would be harder for the reader to follow the link to the next stages of analysis whereby the GWAS-by-subtraction results were used. Hence, we transparently present results from both GWAS-by-subtraction and for education as a covariate and discuss why we emphasise the former for downstream PCA-GWAS and genetic correlation analyses but the latter for SNP heritability analyses.

Methods: “We estimate SNP-based heritability of educational fields after controlling for EA in two ways: through GWAS-by-subtraction and through phenotypic adjustment. We note that it is not recommended to base heritability estimates on the former approach, since environmental variance is undefined. Nonetheless, we report SNP heritability estimates based on GWAS-by-subtraction because they are relevant to interpreting downstream principal components and genetic correlation analyses.”

Figure 4a)

The figure shows genetic correlations between educational fields adjusted for educational attainment using GWAS-by-Subtraction.

It might be nice to report the genetic correlations before GWAS-by-Subtraction, as well as after. This could be done by using one side of the diagonal for ‘before’ correlations and the other for ‘after’ correlations. Additionally, reporting the EA correlations in this table would help explain why this approach was felt to be important in the first place.

We added a table to show the ‘before and after’ genetic correlations with EA4, and a summary of the estimates in the Results: “Genetic correlations between EA and fields, before and after GWAS-by-subtraction, indicate that the procedure was successful for most fields, yet did not completely eliminate EA variance for some fields (e.g., Natural sciences and Social sciences were genetically correlated at ~0.3 with EA (Supplementary Table 8).”

We decided not to add more data to Figure 4a, as there is already a high level of information content. Moreover, it should be clear that genetic correlations between the 10 variables shown in 4a were the input for 4b (which shows the coordinates of PCs 1 and 2).

	Before EA-adjustment	After GWAS-by-subtraction
Education	0.63	0.13
Arts and humanities	0.73	0.17
Social sciences, journalism and information	0.84	0.30
Business, administration and law	0.35	0.01
Natural sciences, mathematics and statistics	0.80	0.34
Information and Communication Technologies (ICTs)	0.44	0.11
Engineering, manufacturing and construction	-0.19	0.01
Agriculture, forestry, fisheries and veterinary	0.44	0.10
Health and welfare	-0.03	0.00
Services	-0.71	-0.08

Citation Format

“We then performed GWAS of the principal components (following) 50”

Some of the text is written as if the citation uses author-date style (e.g. APA), like in the above example. These ought to be fixed at least in the proofing stage.

We corrected this.

Fertility Indicators

“Negative genetic correlations were also observed with two fertility indicators ($r_g = -0.19$ and -0.15).”

Which fertility indicators do these correspond to?

We now specify ‘age at first birth and age at first intercourse (\$r_g = -0.19\$ and \$-0.15\$ )’.

Figure design

Figure 5

Variable names overlap which is a bit ugly.

We recreated all the figures to remove any overlapping labels.

Figures sometimes appear to have low resolution. Saving the images from R in a vector format (e.g. pdf) should help with this issue.

We have exported high resolution figures in pdf format.

Levels of Education

The paper has been altered to make clear that field choice is also studied prior to university. This is very helpful, but a statistic might make it even clearer what sort of education is being studied, e.g., X% of participants had undergraduate degrees versus higher levels of education. The supplementary materials give years of education by field choice, so it still requires a bit of investigation and calculation to work out whether this is a study mostly of high school or university choices.

Thanks for this useful idea. We have added a Supplementary Table documenting the statistics requested by the reviewers (see below) and have summarised this in the text.

Broad field	% with bachelor level or higher qualification
Engineering, manufacturing and construction	31.44293
Services	37.65680
Agriculture, forestry, fisheries and veterinary	44.26460
Business, administration and law	61.79054
Arts and humanities	70.27897
Health and welfare	75.94613
Information and Communication Technologies (ICTs)	94.35185
Natural sciences, mathematics and statistics	97.67577
Social sciences, journalism and information	99.47631
Education	99.90102

Justification of factor analysis

If it is all the same to the authors, we suggest amendments of one paragraph to produce the following:

“We chose to use PCA rather than confirmatory factor analysis (CFA) for studying the dimensionality of educational fields for several reasons. First, PCA involves fewer assumptions than CFA. While CFA models latent factors assumed to represent real traits measured by field choices, PCA simply reduces the dimensionality of the data by finding axes that explain maximum variation. Additionally, CFA might require the somewhat ad hoc addition of cross-loadings on the basis of modification indices in order to achieve a good fit, whereas PCA does not involve model-fit considerations. Second, the limited number of possible fields (10 indicators) makes PCA more suitable. CFA is ideally designed for studying latent factors that can be measured by an extensive, potentially infinite, number of indicators, which is not the case here.”

If the authors have some reason for preferring their original wording, then we will not wrangle with them.

Thank you for these improvements. We incorporated all the suggestions.

False discovery rates

We do not understand the authors’ revision. We are beginning to fear that the authors do not understand what FDR means; we are discovering that this is a common gap in the knowledge of our students and colleagues. We apologize for sounding patronizing, but nevertheless feel compelled to recommend the relevant material in the book Efron and Hastie (2016) and references therein. Once they understand what FDR is, the authors should address our comments. If they already understand what FDR is, they need to strive for greater clarity.

Efron, B., & Hastie, T. (2016). Computer age statistical inference: Algorithms, evidence, and data science. Cambridge University Press.

Our Methods now reads “We used the `p.adjust` function in R with the method “fdr” to control the false discovery rate (FDR; the expected proportion of false discoveries amongst the rejected hypotheses)” and considered results with an adjusted p-value < 0.05 as significant..

References:

<https://www.rdocumentation.org/packages/stats/versions/3.6.2/topics/p.adjust>

Benjamini, Y., and Yekutieli, D. (2001). The control of the false discovery rate in multiple testing under dependency. *Annals of Statistics*, 29, 1165--1188. 10.1214/aos/1013699998.

“social science context to the foreground”

Although prompted by another reviewer, this expansion of “social science context” is dubious. Why do the authors fail to mention the possibility (nay, near certainty) of innate evolved differences between men and women in educational and occupational preferences? This failure will suggest to the discerning reader that this is a slanted presentation of the issues. Unless the authors can be more thorough (with the downside of being more prolix), we suggest pruning this part back.

We think that the amount of social science content is appropriate for the topic, given the extensive sociological literature on fields of study. Our article emphasises that several factors prevent us from drawing strong conclusions regarding the role of sex in educational field choices, including low sample sizes for most sex-specific fields.

How many PCs?:

In response to our criticism that the extraction of two factors is insufficiently justified, all justification is removed altogether. We were hoping that perhaps more factors might be extracted and analyzed, even if only in the supplementary materials. But removing any explanation for the choice altogether is a deterioration rather than an improvement. The explanation for the author’s choice seems to be simplicity — two PCs means fewer things to focus on. This explanation would be good enough.

We performed parallel analysis to address the reviewers’ concern about the arbitrariness of our selection of PCs 1 and 2. We added the following text in the Results section to summarise our choice: ‘Although parallel analysis indicated that three PCs could be extracted (see Supplementary Figure 25), we focus on the first two PCs for simplicity and interpretability.’

“Horizontal stratification”

This phrase, although perhaps inserted in response to reviewer comments, is more than a bit awkward. “Stratification” is derived from “stratum,” which according to the New Oxford American Dictionary means “a layer or series of layers of rock in the ground.” Strata are thus always stacked vertically.

This is a good point. However, we are retaining the term Horizontal Stratification as it is a major topic in sociology (see e.g., articles in the Annual Review of Sociology) and thus makes the article more searchable and accessible to a broader audience.

Reviewer #1 (Remarks on code availability):

Code for data analysis

The journal portal asks the reviewers to indicate whether they have inspected the code. We have done so cursorily and make the following observations.

- 1) Code for the PGI and genetic correlations are missing, presumably because they were done by co-authors who are not now available.
- 2) Other code for figures seems to be missing. In the Figures folder, code files for stage 2, 4, and 5 are there but not 1 or 3. (Which I think would have created the plots of SNP heritability)
- 3) The code cannot be run. For example, `3_genomicSEM_GWAS.bash` begins in R then switches to bash. This isn't because R is being run from the terminal or bash from R, but because the code files are just long collections of different codes that must have been run interactively. The scripts do contain further logs of instructions, e.g. "# using this pipeline for splitting up sumstats etc.:

```
# https://github.com/sarahcolbert/gsemGWAS/blob/master/README.md
# -- moved pipeline to folder
# -- edited config file then activated it"
```

The code is out of date. We can still see code for the CFA but not for Genomic PCA.

We have updated all the code to include PGI, genetic correlation, PCA analyses, and plotting. We also tidied it up and numbered the scripts to ensure readers can easily follow the steps (e.g., splitting up bash and R scripts).

`linprob` is an argument in the `GenomicSEM` function `sumstats`. The authors set this argument to true, although it is meant to be set to true when a linear probability model was used for the GWAS. The authors write in the code that they use a linear probability model, but in the paper they say logistic. Regardless, the setting should be set to false because they have meta-analyzed the results across samples first. We are not sure whether setting the argument incorrectly matters, because we do not know precisely what these functions are doing under the hood.

The correct setting in our case is `linprob=T`. As the `GenomicSEM` documentation says, `linprob` should be set to TRUE if “the phenotype was a dichotomous outcome for which there are only Z-statistics in the summary statistics file. In this case, a form of standardization is performed where logistic betas are backed out from a signed effect column.” We originally had the same idea as the reviewers that this should be set to False because we used a logistic model, but this does not work because the software cannot compute standard errors.

We want to emphasize that we do not view these seeming irregularities as a bar to publication. But if it is true that the journal requires analysis code to be posted, the authors might want to do what they can to clean it up.

Reviewer #2 (Remarks to the Author):

I liked the originally submitted manuscript, and I think the revised manuscript is even stronger. The authors have been responsive to my main concerns. Moreover, I think the addition of an FAQ, which the authors have disseminated widely, was a great idea.

I have only a few remaining, minor comments:

Lines 247 and 402: The paper refers to the coefficients on the parents' PGIs as estimates of indirect genetic effects, but unlike the coefficients on the proband's PGI, the parental coefficient is biased by population stratification and other gene-environment correlation. I suggest clarifying/rephrasing.

Thank you for this helpful point. We modified these lines to make it clear that the parental coefficients are likely to be biased.

Line 285: Typo: "used applied"

We corrected this.

FAQ lines 273-275: "It has not been possible to examine how fields of study hang together since each person usually only studies one field. Genetic methods overcome this problem." – I don't think this will make sense to a reader (even an expert). I know it's hard to explain to a non-expert, but I think it needs some work.

We changed this to "(We) identify clusters of fields that share a genetic basis. Individuals tend to only specialize in one field, making it difficult to understand similarities across fields. For example, we don't know whether there are any overlapping influences on choosing STEM versus Arts subjects. We use a new *genetic* method that can measure the similarity of two traits even if they are measured in two different groups."

FAQ lines 517-518: I like the sentiment, but these lines come out of nowhere, which risks making them seem just like an effort at political correctness. I suggest adding a transition.

We changed this to "In light of these technical and societal issues, it is difficult and inadvisable to draw conclusions for any individual or for policy based on our genetic study of educational fields."

Signed,
Dan Benjamin

Reviewer #2 (Remarks on code availability):

I did not attempt to run the code, but I did look it over. For the most part, it looks clear and well commented. However, files 4 and 5 under "meta" are missing.

We have ensured that all code is presented and up to date.

Reviewer #3 (Remarks to the Author):

The authors have done an extraordinary job incorporating sociological explanations and addressing the ethical considerations that I raised, as well as a fantastic job addressing the comments of the other 2 reviewers. This paper is now in line with the best practices in the field of sociogenomics/behavioral genetics, and it reads very well for both a social science and genetics audience. It is imaginative, rigorous, and original in demonstrating the value added from incorporating genetics into the social sciences and vice versa. I congratulate the authors and fully support the publication of this paper in Nature Genetics.

Reviewer #3 (Remarks on code availability):

I assessed all code posted on GitHub. It all seems satisfactory!

Thank you very much for the encouraging feedback!